_Article_

# The NLRP6 inflammasome is activated by sterile or pathogen-induced endolysosomal damage

Alexandra Boegli (ID), Elliott M Bernard (ID), Louise Lacante (ID), Gaël Majeux, Ella Hartenian (ID), Vanessa Mack (ID) & Petr Broz (ID) ✉

## Abstract

The cytosolic innate immune sensor NLRP6 controls host defense against bacteria and viruses in the gastrointestinal tract, but the underlying mechanism is poorly understood. Here, we report that NLRP6 forms an inflammasome following endolysosomal damage caused by sterile triggers or bacterial pathogens such as _Listeria monocytogenes_ in human intestinal epithelial cells (IECs). NLRP6 activation requires Listeriolysin O-dependent cytosolic invasion of _L. monocytogenes_ and triggers IEC pyroptosis and IL-1β release via ASC/caspase-1-mediated GSDMD cleavage. NLRP6 activation requires its NACHT domain and ATP binding, whereas inflammasome formation is independent of bacterial pathogen-associated molecular patterns (PAMPs), such as lipoteichoic acid or dsRNA, which were previously reported to activate NLRP6. _L. monocytogenes_ mutants deficient in cell-to-cell spread or escape from secondary vacuoles induce lower levels of cell death, linking bacteria-induced endolysosomal damage to NLRP6 activation. Finally, sterile endolysosomal damage recapitulates pathogen-induced NLRP6 activation and induces IEC pyroptosis. In summary, our study reveals that NLRP6 enables intestinal epithelial cells to detect endolysosomal damage, thereby mediating their response not only to pathogens but more generally to wide-ranging sources of pathological endolysosomal damage.

**Keywords** Inflammasome; NLRP6; Listeria; Endolysosomal Damage; Pyroptosis
**Subject Categories** Autophagy & Cell Death; Immunology; Microbiology; Virology & Host Pathogen Interaction

See also: ANR Weber & P Rosenstiel

## Introduction

The human body constantly encounters pathogens at barrier tissues such as the skin, lung, and gastrointestinal tract. First line defenses like mucus secretion and antimicrobial peptides, but also specialized immune cells, like macrophages, protect barrier tissues from being invaded by these pathogens. To avoid these extracellular defenses, many pathogens have developed strategies to invade and replicate within epithelial cell layers, and even to spread intracellularly from cell to cell. One such pathogen is _L. monocytogenes_, a foodborne, Gram-positive bacterium that can cause severe diseases such as meningitis and sepsis in susceptible patients (Lomonaco et al, 2015; Posfay-Barbe and Wald, 2009). _Listeria_ enters enterocytes in the gastrointestinal tract and escapes from the endolysosomal system using Listeriolysin O, a pore-forming toxin, to damage the endolysosomal membrane (Seveau, 2014). In the host cell cytosol, _Listeria_ replicates and uses the effector ActA, which recruits host Arp2/3, to polymerize actin to move within the infected cell and to spread to neighboring cells (Travier and Lecuit, 2014). This strategy allows the bacteria to avoid resurfacing to the hostile extracellular compartment and can even promote the dissemination to distal organs via migrating cells such as macrophages (Drevets et al, 2004). Other enteric bacterial pathogens like _Shigella flexneri_ also use actin-based motility to move from cell to cell and can cause high morbidity and mortality in risk groups (Choe and Welch, 2016; Khalil et al, 2018).

To sense intracellular pathogens, host cells express pattern recognition receptors (PRRs) that detect conserved microbial molecules, such as modified nucleic acids, cell wall components, or a disruption of cellular homeostasis caused by toxins or other virulence factors. A subset of cytosolic PRRs, namely PYRIN, AIM2, and members of the NOD-like receptor family (NLRs), induce the formation of multi-protein signaling complexes, known as inflammasomes. Inflammasomes recruit and activate the protease Caspase-1, either directly or via the adapter ASC, which then cleaves the pro-forms of IL-1β/-18, to produce the bioactive, mature forms of these cytokines, and induces pyroptosis via the cleavage and activation of the pore-forming cell death executioner GSDMD (Broz and Dixit, 2016). Inflammasome-induced cell death is particularly important in barrier tissues, as it can, for example, promote the extrusion of infected cells into the intestinal lumen thereby reducing bacterial loads in the intestinal epithelium, as shown for NLRC4-mediated restriction of the enteric pathogens _S. flexneri_ or _Salmonella enterica_ serovar Typhimurium (Rauch et al, 2017; Roncaioli et al, 2023; Sellin et al, 2014).

Another NLR family member that has been proposed to confer antimicrobial immunity in intestinal epithelial cells is NLRP6. Although NLRP6 mRNA can be found in different tissues, human NLRP6 protein is exclusively expressed in the small intestine, while murine NLRP6 is expressed in the small intestine and colon (Bracey et al, 2021). NLRP6 has been shown to play an essential role in regulating susceptibility to

Department of Immunobiology, University of Lausanne, Epalinges, Switzerland. ✉E-mail: petr.broz@unil.ch

bacterial and viral infections (Anand et al, 2012; Birchenough et al, 2016; Ghimire et al, 2018; Hara et al, 2018; Levy et al, 2015; Volk et al, 2019; Wang et al, 2015; Wlodarska et al, 2014). Furthermore, NLRP6 is linked to colorectal cancer and inflammatory bowel disease, as well as neuroinflammation (Angosto-Bazarra et al, 2022; Bracey et al, 2021). While some of its functions were proposed to depend on regulating NFκB or type-I-interferon (IFN) signaling (Anand et al, 2012; Wang et al, 2015), NLRP6 has also been reported to form an inflammasome and thereby regulate IL-18 release in vivo (Elinav et al, 2011; Hara et al, 2018; Levy et al, 2015; Shen et al, 2021). The NLRP6 inflammasome has also been suggested to regulate autophagy and mucus secretion by Goblet cells, although others have not found a role in baseline mucus layer formation (Volk et al, 2019; Wlodarska et al, 2014). Several chemically and structurally different ligands were proposed to activate NLRP6. Among these are bacterial metabolites, double-stranded viral RNA (dsRNA), and lipoteichoic acid (LTA), a cell wall component of Gram-positive bacteria such as *L. monocytogenes* and *Staphylococcus aureus* (Hara et al, 2018; Levy et al, 2015; Shen et al, 2021).

In this study, we show that human and mouse NLRP6 form an inflammasome in response to *L. monocytogenes* infection, and that this requires Listeriolysin O-dependent bacterial entry into the host cell cytosol. In *Listeria*-infected human intestinal epithelial cells (IECs), NLRP6 activation resulted in the formation of ASC specks that induced Caspase-1-dependent, but Caspase-4-independent, GSDMD cleavage and pyroptotic cell death. Functional characterization of NLRP6 identified critical roles for a positively charged region within the FISNA domain and the Walker A and B motifs in the NACHT domain of NLRP6. The generation of chimeric receptors between NLRP6 and the closely related NLRP3 showed that *Listeria* recognition was mediated via the NLRP6 NACHT domain. Further investigating NLRP6 signal recognition, we found that none of the previously reported NLRP6-activating pathogen-associated molecular patterns (PAMPs) (Hara et al, 2018; Shen et al, 2021) activated NLRP6 in our system. Indeed, even transfections of whole bacterial lysates, composed of a mixture of diverse PAMPs, failed to elicit NLRP6 activation. In contrast, NLRP6 activation required the presence of live, replicating bacteria in the cytosol, suggesting that NLRP6 does not sense PAMPs but pathogen-induced alterations to cellular homeostasis. We tested the importance of cell-to-cell spread and found that both blocking actin-based motility and escape from double-membrane vacuoles after cell-to-cell spread strongly reduced NLRP6 activation, indicating that vacuolar escape during secondary infections and the resulting endolysosomal damage is a prerequisite for NLRP6 activation. Confirming that NLRP6 acts as a sensor for endolysosomal integrity, we finally found that inducing sterile endolysosomal damage also led to NLRP6 activation and Caspase-1/GSDMD-dependent IEC death, completely independent of any bacterial PAMP. In summary, these data show that NLRP6 does not get activated by pathogen-derived danger signals, as previously published, but responds to the loss of integrity of endolysosomal pathways.

## Results

### NLRP6 detects the entry of *L. monocytogenes* into the host cell cytosol

NLRP6 is predominantly expressed in primary IECs, with no expression found in mouse bone marrow-derived macrophages, even after priming with TLR ligands or interferon-γ (IFNγ) (Bracey

et al, 2021; Shen et al, 2021) (Appendix Fig. S1A). We tested multiple human (HIEC-6, Mino, SW1463, HT-29, HCT116, and Caco-2) and murine cell lines (YAMC, mlCcl2) for expression of NLRP6, either motivated by their intestinal origin, previous use in publications on NLRP6 or RNA expression profiles in the human protein atlas. None of the cell lines tested expressed NLRP6 at the protein level, even after priming (Appendix Fig. S1). Therefore, we developed an NLRP6 inflammasome reconstitution assay in HEK293T cells based on the inflammasome adaptor protein ASC that forms macromolecular inflammasome assemblies referred to as ASC specks. For this assay, HEK293T cells stably expressing GFP- or mCherry-tagged ASC (GFP-/mCherry-ASC$^{tg\ (transgenic)}$) were transiently transfected with an NLRP6 expression plasmid. Since NLRs are prone to auto-activate when overexpressed, we first determined the dynamic range of the assay by transfecting increasing amounts of plasmid DNA encoding human NLRP6. We found that with increasing amounts of transfected plasmid, the percentage of ASC speck-positive cells reached a plateau at 50–60% of the total population, as determined by flow cytometry-based quantification of ASC speck formation (Sester et al, 2015) (Fig. EV1A,B).

To probe for signal-specific NLRP6 activation, we chose a concentration of human NLRP6 vector DNA that caused only basal ASC speck formation upon transfection into GFP-ASC$^{tg}$ HEK293T (80 ng/300,000 cells). These cells were infected with two strains of the Gram-positive bacteria *L. monocytogenes*, as well as *S. aureus*, which were proposed to induce NLRP6 activation, and the Gram-negative bacterium *S.* Typhimurium as a control (Anand et al, 2012; Ghimire et al, 2018; Hara et al, 2018). While *S. aureus* and *S.* Typhimurium did not cause elevated levels of ASC speck formation, *L. monocytogenes* infection resulted in up to 50% of ASC speck-positive cells (Figs. 1A,B and EV1C), indicating that *Listeria* infection triggered NLRP6 activation. To validate the specificity of this response, we expressed NLRP6$^{W53E}$, which is defective for ASC recruitment (Shen et al, 2019), and found that the mutant protein did not allow ASC speck formation after *Listeria* infection (Fig. 1B). In addition, we also repeated the assays with murine NLRP6, which shares 71.14% amino acid identity with human NLRP6 (BLASTP 2.16.0 + ), and infected these cells with *L. monocytogenes*. This also induced ASC speck formation, indicating that NLRP6 activation upon *Listeria* infection is conserved between mouse and human homologs (Figs. 1C,D and EV1D).

*L. monocytogenes* employs the pore-forming toxin Listeriolysin O to escape from the endolysosomal compartment into the cytosol (Seveau, 2014). We found that wild-type bacteria caused progressive NLRP6-dependent ASC speck formation over time, while the Δ*hly* strain, which lacks Listeriolysin O, caused little NLRP6 activation (Fig. 1A,B). Optimal activity of Listeriolysin O requires the low pH of the phagolysosome, allowing the bacteria to escape this compartment without causing excessive damage to other host cell membranes in the process (Beauregard et al, 1997; Podobnik et al, 2015; Schuerch et al, 2005). Consistently, preventing the acidification of endolysosomes with bafilomycin A1 efficiently reduced both the escape of wild-type *Listeria* into the cytosol, as judged by reduced numbers of actin-positive bacteria, and NLRP6-dependent ASC speck formation (Figs. 1E,F and EV1E). We also tested if Listeriolysin O itself could trigger NLRP6 activation. At the concentrations used, purified Listeriolysin O caused plasma

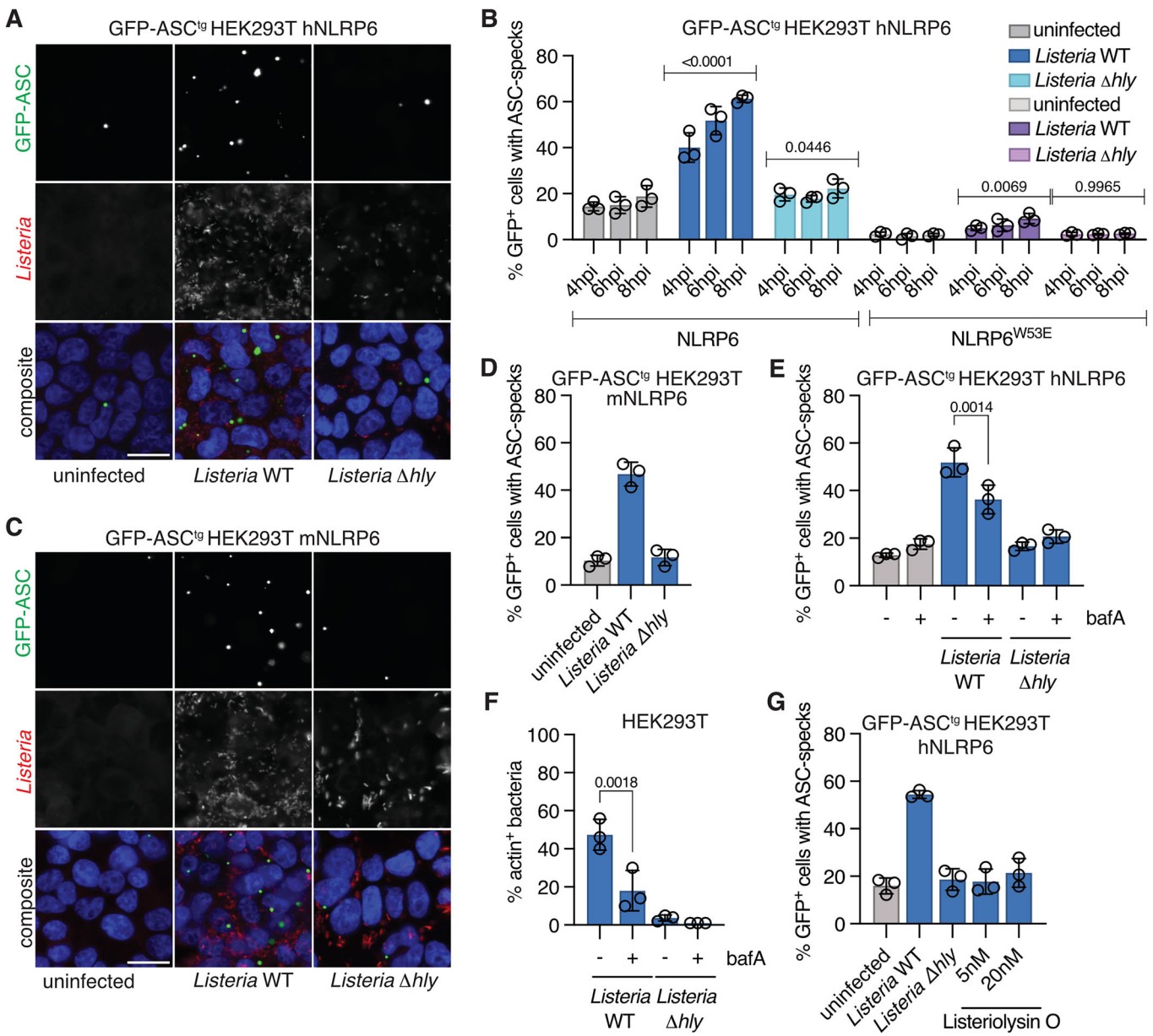

**Figure 1. NLRP6 detects the entry of *Listeria* into the host cell cytosol.**

(A) Representative micrographs of hNLRP6-expressing GFP-ASC$^{tg}$ HEK293T cells infected with WT or Δ*hly* *L. monocytogenes* EGD for 6 h at MOI 20. (B) Flow cytometry-based quantification of ASC speck formation in hNLRP6 or hNLRP6$^{W53E}$-expressing GFP-ASC$^{tg}$ HEK293T cells infected with WT or Δ*hly* *L. monocytogenes* EGD for 4–8 h. hpi: hours post infection. *P* values determined by comparison with the respective uninfected control. (C, D) Representative micrographs and quantification of ASC speck formation by flow cytometry in mNLRP6-expressing GFP-ASC$^{tg}$ HEK293T cells infected with WT or Δ*hly* *L. monocytogenes* EGD for 6 h (E). ASC speck formation in hNLRP6-expressing GFP-ASC$^{tg}$ HEK293T cells left untreated or treated with bafilomycin A1 (bafA) for 2 h and infected with WT or Δ*hly* *L. monocytogenes* EGD for 6 h (F). Percentage of actin-positive, intracellular *Listeria* after 1 h of infection in HEK293T cells treated as in (E). (G) ASC speck formation of hNLRP6-expressing GFP-ASC$^{tg}$ HEK293T cells infected with WT or Δ*hly* *L. monocytogenes* EGD for 6 h at MOI 20 or treated with 5 or 20 nM purified Listeriolysin O for 6 h. Graphs show mean ± SD from three pooled independent experiments. Quantification in panel F was performed on maximum projection confocal micrographs (Fig. EV1F) of ten fields of view and at least 600 bacteria per experiment and condition. Each data point represents the mean of one experiment. Images are representative of at least three independent experiments. Scale bar represents 20 μm. Statistics used: two-way ANOVA with Šídák's multiple comparisons (B), one-way ANOVA with Dunnett's multiple comparisons test (E, F). Source data are available online for this figure.

membrane permeabilization (Fig. EV1F) but did not trigger NLRP6-dependent ASC-speck formation (Fig. 1G), indicating that Listeriolysin O promoted NLRP6 activation indirectly by allowing cytosolic entry of *Listeria*. Furthermore, while *Listeria* infection caused NLRP6 activation, it did not induce host cell

permeabilization, thus excluding plasma membrane permeabilization as a potential trigger for NLRP6 activation (Fig. EV1G). In summary, these experiments showed that the entry of *L. monocytogenes* into the host cell cytosol is required for the activation of NLRP6 in reconstituted HEK293T cells.

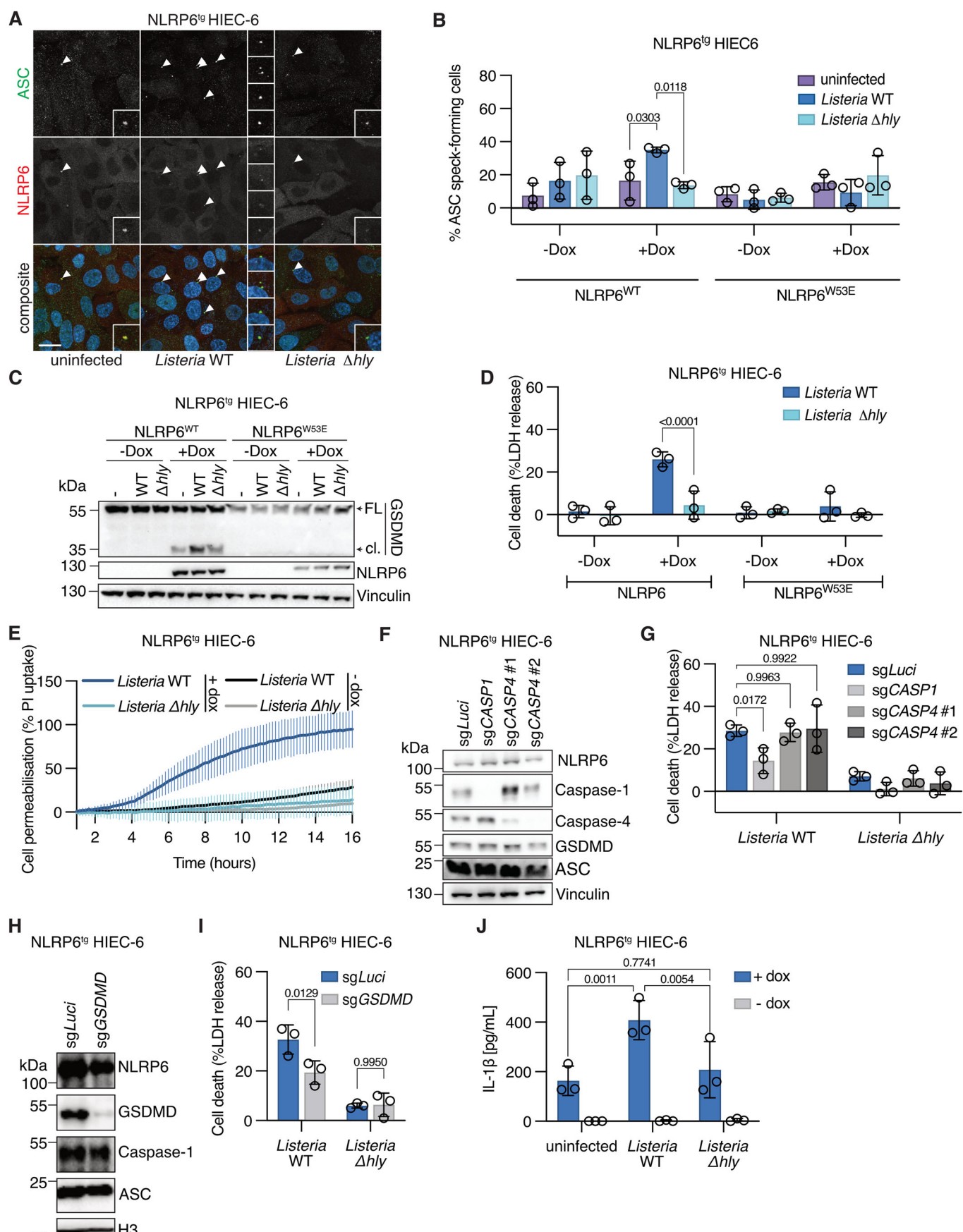

**Figure 2. NLRP6 induces Caspase-1-dependent GSDMD cleavage and pyroptosis in *Listeria*-infected human IECs.**

(A, B) Representative maximum projection confocal micrographs and ASC speck quantification in NLRP6-WT$^{tg}$ or NLRP6-W53E$^{tg}$ HIEC-6 cells induced or not with 1 µg/ml doxycycline overnight and infected with WT or Δ*hly L. monocytogenes* EGD for 8 h. Arrowheads indicate regions in insets. (C, D) GSDMD processing and LDH release from NLRP6-WT$^{tg}$ or NLRP6-W53E$^{tg}$ HIEC-6 cells induced or not with 1 µg/ml doxycycline overnight and infected with WT or Δ*hly L. monocytogenes* EGD for 8 h. (E) Time course of propidium iodide (PI) uptake by NLRP6-WT$^{tg}$ HIEC-6 cells induced or not with 1 µg/ml doxycycline overnight and infected with WT or Δ*hly L. monocytogenes* EGD. (F, G) Protein expression and LDH release from NLRP6-WT$^{tg}$ HIEC-6 control cells (sg*Luci*) or polyclonal populations lacking Caspase-1 (sg*CASP1*) or Caspase-4 (sg*CASP4* #1 and #2) induced with 1 µg/ml doxycycline overnight and infected with WT or Δ*hly L. monocytogenes* EGD for 8 h. (H, I) Protein expression and LDH release from NLRP6-WT$^{tg}$ HIEC-6 control cells (sg*Luci*) or polyclonal populations lacking GSDMD (sg*GSDMD*) induced with 1 µg/ml doxycycline overnight and infected with WT or Δ*hly L. monocytogenes* EGD for 8 h. (J) IL-1β release from NLRP6-WT$^{tg}$ HIEC-6 cells induced or not with 1 µg/ml doxycycline overnight and infected with WT or Δ*hly L. monocytogenes* EGD for 8 h. (B) Shows mean ± SD from three fields of view each for three pooled experiments with at least 200 cells analyzed per condition per experiment. Each data point represents the mean of one experiment. (D, E, G) Show mean ± SD from three pooled experiments with three technical replicates each. Each data point represents the mean of one experiment. Images and blots are representative of three independent experiments. FL full length, cl. cleaved. Scale bar represents 20 µm. Statistics used: two-way ANOVA with Tukey's (B), Dunnett's (G), or Šídák's multiple comparisons (D, I, J). Source data are available online for this figure.

## NLRP6 induces Caspase-1-dependent GSDMD cleavage and pyroptosis in infected human IECs

We next wanted to study NLRP6 activation by *Listeria* infection in a more physiological setting. As HEK293T cells lack critical inflammasome components such as Caspase-1 and GSDMD, we turned to an intestinal epithelial cell line to determine whether NLRP6 engages the full inflammasome pathway upon activation. HIEC-6 (human intestinal epithelial cell-6) is a non-immortalized cell line derived from human fetal small intestine (Perreault and Beaulieu, 1996). Immunoblotting confirmed that HIEC-6 express the inflammasome components ASC, Caspase-1, Caspase-4, and GSDMD, but do not endogenously express NLRP6 (Appendix Fig. S2A) or NLRP3 (Appendix Fig. S2B). We therefore reconstituted these cells with doxycycline-inducible wild-type NLRP6 or the NLRP6$^{W53E}$ mutant coupled with an IRES-GFP for selection (Appendix Fig. S2A) before infecting them with *L. monocytogenes*. Consistent with a specific activation of NLRP6, we observed that wild-type *L. monocytogenes* caused significantly elevated levels of ASC speck formation, GSDMD processing, and LDH release (a measure for lytic cell death) in HIEC-6 treated with doxycycline to induce NLRP6$^{WT}$ expression, but not in non-doxycycline-treated, and thus non-NLRP6-expressing, controls (Fig. 2A–D; Appendix Fig. S2C). Of note, doxycycline treatment induced some background inflammasome activation, potentially due to NLRP6 autoactivation, as visible by background ASC-speck formation and GSDMD processing. Consistent with data obtained from HEK293T cells, we found that the interaction of NLRP6$^{PYD}$ with ASC$^{PYD}$ was required for inflammasome assembly in HIEC-6 cells, since expression of NLRP6$^{W53E}$ did not allow ASC speck formation, GSDMD cleavage or LDH release after wild-type *L. monocytogenes* infection (Fig. 2A–D). In addition, we observed that NLRP6 was recruited to ASC specks, further indicating that NLRP6 is the driver of inflammasome responses in these cells (Fig. 2A). As observed in HEK293T cells, the comparison of wild-type and Δ*hly L. monocytogenes* infections showed that cytosolic entry was required for NLRP6-dependent ASC speck formation, GSDMD cleavage and cell death in NLRP6$^{tg}$ HIEC-6 cells (Fig. 2A–D). To determine the kinetics of *Listeria*-induced NLRP6 activation in HIEC-6 cells, we next assayed propidium iodide (PI) uptake, a measure of plasma membrane permeabilization by GSDMD pores, in cells infected with wild-type and Δ*hly L. monocytogenes*. This analysis showed that the PI signal started to increase in cells infected with WT bacteria compared to cells

infected with Δ*hly* bacteria or cells not expressing NLRP6 at approximately 3 h post infection (Fig. 2E), indicating that NLRP6 activation occurs within a few hours after bacterial escape into the cytosol. Even though we did not find expression of NLRP3 in the HIEC-6 cells at baseline (Appendix Fig. S2B), we additionally treated the cells with the NLRP3 inhibitor MCC950 (Coll et al, 2019) before and during infection. We found that this treatment did not affect cell death upon NLRP6 activation, excluding a contribution of NLRP3 and confirming previous results that MCC950 does not inhibit NLRP6 (Tapia-Abellan et al, 2021) (Appendix Fig. S2D).

A previous study proposed that, in mouse macrophages, transfection of LTA from *S. aureus* induces NLRP6- and ASC-dependent Caspase-11 activation, which then promotes Caspase-1 activation and interleukin-1β (IL-1β)/IL-18 maturation without the induction of pyroptosis (Hara et al, 2018). To study the contribution of Caspase-1 and Caspase-4 (the human ortholog of murine Caspase-11), we used CRISPR/Cas9 to inactivate these genes in NLRP6$^{tg}$ HIEC-6 cells (Fig. 2F) and infected the cells with *L. monocytogenes*. While deletion of Caspase-1 significantly reduced pyroptotic cell death as measured by LDH release, Caspase-4 deficiency had no impact on NLRP6-induced death (Fig. 2G). Similarly, GSDMD knockout HIEC-6 expressing NLRP6 WT also showed reduced pyroptosis during *L. monocytogenes* infection (Fig. 2H,I). Interestingly, neither Caspase-1 nor GSDMD knockout completely ablated cell death. This could be due to residual protein expression in the polyclonal pools or the activation of apoptosis, as has previously been reported to occur downstream of inflammasome activation in Caspase-1 or GSDMD-deficient cells (Heilig et al, 2020; Lee et al, 2018; Pierini et al, 2012; Tsuchiya et al, 2019). Finally, we assayed cytokine secretion in HIEC-6 cells expressing NLRP6 and found that infection with wild-type *L. monocytogenes* elicited increased IL-1β secretion, while the levels of IL-1β release upon infection with Δ*hly L. monocytogenes* were comparable to uninfected controls (Fig. 2J). In summary, our data shows that NLRP6 is activated upon cytosolic entry of *L. monocytogenes* and consequently polymerizes ASC, leading to Caspase-1 recruitment, GSDMD cleavage, cytokine processing and secretion, as well as pyroptotic cell death in human IECs without any contribution of the non-canonical Caspase-4 inflammasome pathway. Furthermore, the results suggest that the Listeriolysin O-dependent escape of *Listeria* from the endolysosomal compartment is an essential prerequisite for NLRP6 activation in human epithelial cells.

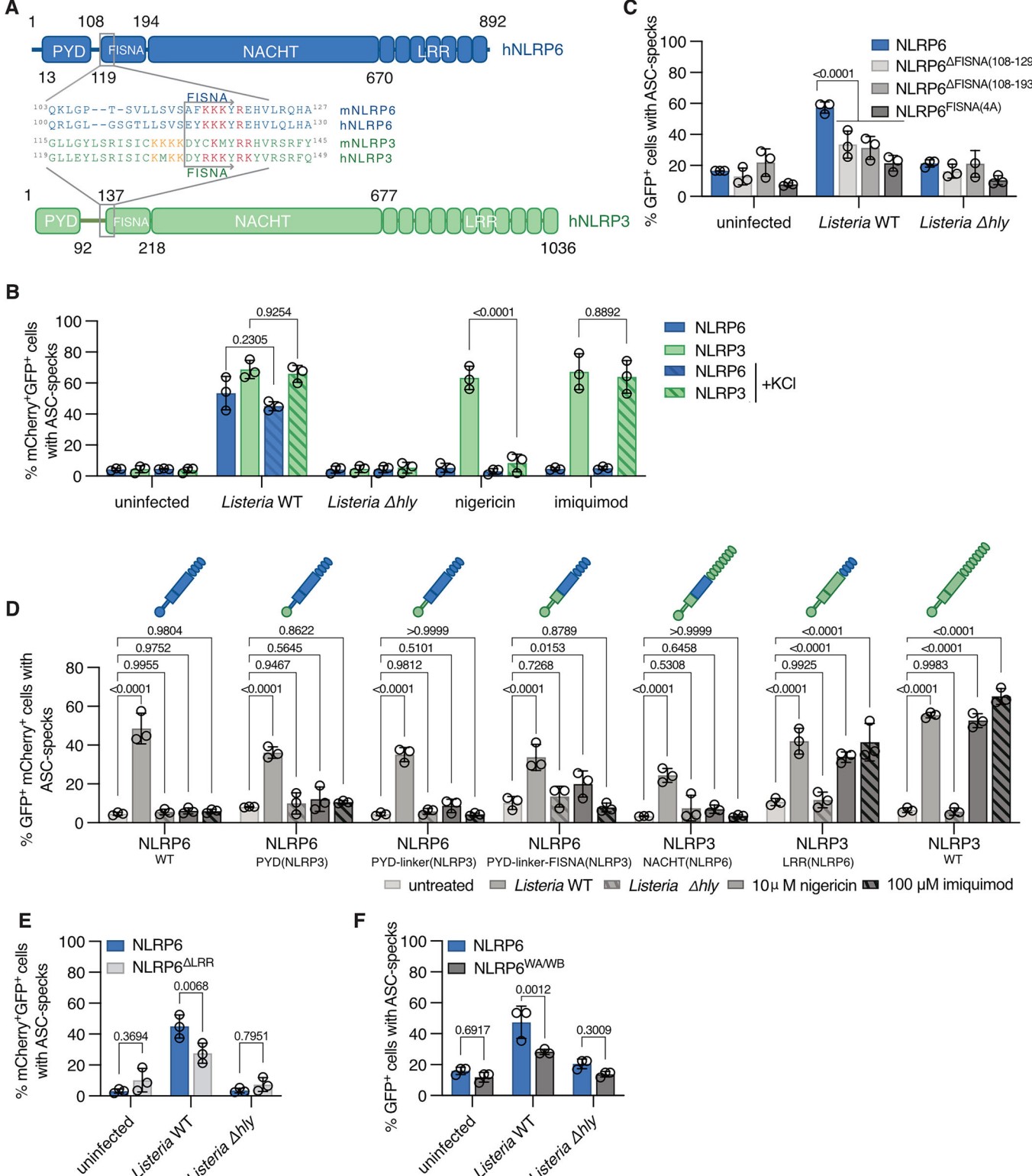

**Figure 3. Molecular characterization of the NLRP6 receptor.**

(A) Schematic representation of human NLRP6 and NLRP3 showing their domain structures and the respective polybasic regions within the FISNA of mouse and human NLRP3 and NLRP6 aligned. (B) Flow cytometry-based quantification of ASC speck formation in GFP-ASC[tg] HEK293T cells expressing hNLRP6-mCherry or hNLRP3-mCherry infected with WT or Δ*hly L. monocytogenes* EGD for 6 h, treated with 10 μM nigericin for 1 h or with 100 μM imiqumod for 6 h in the presence or absence of 60 mM extracellular potassium (KCl). (C) Flow cytometry-based quantification of ASC speck formation in GFP-ASC[tg] HEK293T cells, expressing the indicated NLRP6 proteins, infected with WT or Δ*hly L. monocytogenes* EGD for 6 h (D). Flow cytometry-based quantification of ASC speck formation in GFP-ASC[tg] HEK293T cells expressing either NLRP6-WT, NLRP3-WT, or the indicated chimeric NLRP6-3 proteins with mCherry tags, infected with WT or Δ*hly L. monocytogenes* EGD for 6 h, treated with nigericin for 1 h or with imiquimod for 6 h. (E) Flow cytometry-based quantification of ASC speck formation in GFP-ASC[tg] HEK293T cells, expressing NLRP6[ΔLRR]-mCherry, infected with WT or Δ*hly L. monocytogenes* EGD for 6 h. (F) Flow cytometry-based quantification of ASC speck formation in GFP-ASC[tg] HEK293T cells, expressing NLRP6[WA/WB], infected with WT or Δ*hly L. monocytogenes* EGD for 6 h. Graphs show mean ± SD from three pooled experiments. Statistics used: two-way ANOVA with Tukey's (B), Dunett's (C, D), or Šídák's (E, F) multiple comparisons test. Source data are available online for this figure.

## NLRP6 shares structural similarities and activators with NLRP3

To understand how NLRP6 detects *Listeria* infection, we next characterized its mode of activation and the domain(s) implicated in this process. Sequence alignment showed that NLRP6 is highly similar to NLRP3, comprising an N-terminal PYD domain, followed by a linker, a NACHT domain and C-terminal LRRs (Fig. 3A). NLRP3 has been proposed to detect alterations of cellular homeostasis, such as the disruption of the TGN or endosomal trafficking, but the exact nature of the signal remains unknown (Chen and Chen, 2018; Zhang et al, 2023). Interestingly, NLRP3 was also reported to detect infection by various pathogens, including *L. monocytogenes* (Kim et al, 2010; Mariathasan et al, 2006). Given these similarities, we tested whether NLRP3 and -6 share additional activators by infecting NLRP3- or NLRP6-expressing ASC-GFP[tg] HEK293T with wild-type or Δ*hly L. monocytogenes* or treating them with nigericin or imiquimod, two well-known NLRP3 activators. We tagged the NLRs with mCherry, which allowed us to gate on transfected cells displaying comparable low levels of protein before quantifying ASC speck formation (Fig. EV2A). Wild-type *L. monocytogenes* induced robust ASC speck formation in both NLRP3 and NLRP6-expressing cells, which was dependent on Listeriolysin O in both instances. By contrast, while nigericin and imiquimod efficiently triggered NLRP3 activation, they failed to activate NLRP6 (Fig. 3B), revealing that NLRP6 does not share all activators with NLRP3. Since potassium efflux was previously shown to be required for NLRP3 activation by nigericin and other triggers, but not for imiquimod, we tested its impact on NLRP6 activation (Gross et al, 2016; Munoz-Planillo et al, 2013). We found that the addition of extracellular potassium, to prevent potassium efflux, did not reduce *Listeria*-induced NLRP6 activation (Fig. 3B). Interestingly, extracellular potassium also did not block *Listeria*-induced NLRP3 activation, indicating a mode of activation distinct from nigericin but potentially similar to imiquimod (Fig. 3B).

A characteristic feature of NLRP3 is its FISNA domain, which is located between the linker and NACHT domain. Previous work has suggested that nigericin and imiquimod sensing maps to the linker-FISNA domains of NLRP3. Recent structural studies, however, showed that several disordered regions in the NLRP3 FISNA, such as loop1 and middle region/loop 3 (Fig. EV2B), fold during the transition from the inactive to the active receptor, thus allowing receptor oligomerization and correct positioning of the PYD (Tapia-Abellan et al, 2021; Xiao et al, 2023). Therefore, mutations within the FISNA domain could critically affect these conformational changes that are essential for receptor activation, rather than signal recognition/sensing. NLRP6 features a conserved polybasic motif preceding the NACHT in a domain that shows sequence similarity to the FISNA of NLRP3 and NLRP12, and which we thus propose to define as the NLRP6 FISNA (Fig. 3A) (Howe et al, 2016; Tapia-Abellan et al, 2021). Moreover, pairwise alignments of the predicted structure of the NLRP6-FISNA (Alphafold v2) with inactive and active NLRP3, show that it is proposed to adopt a similar structure with helix 1, loop1, loop2, and loop3, which likely would fold into helix 2 during receptor activation (Fig. EV2B). To test the role of the FISNA in NLRP6 activation, we expressed two deletion mutants (aa108-129 and aa108-193) in GFP-ASC[tg] HEK293T and found that both deletions rendered the receptor unable to induce ASC speck formation upon *Listeria* infection (Fig. 3C). The larger FISNA deletion (aa108-193) showed partially impaired autoactivation, potentially indicating a structural defect that reduces ASC recruitment (Fig. EV2C,D).

Mouse NLRP3 is activated by the recruitment to phosphatidylinositol-4-phosphate (PI4P)-enriched membranes via a positively charged motif (KKKK[130]) in the linker domain (Chen and Chen, 2018). Mutation of the corresponding motif (KMKK[134]) in human NLRP3 has been shown to have no impact on activation, potentially due to redundancies with a second polybasic motif located within the FISNA of human NLRP3 (RKKYRK[142]) (Tapia-Abellan et al, 2021). While NLRP6 does not feature the first polybasic motif, it features the second one in its FISNA (KKKYR[123]) (Fig. 3A). Mutating these four positively charged residues to alanine led to a significant decrease in NLRP6-dependent ASC-speck formation upon *Listeria* infection, even though the receptor was still able to induce ASC specks through autoactivation upon high levels of overexpression (Figs. 3C and EV2C,D). We wanted to understand whether the same positively charged residues in the NLRP3 FISNA domain were important for its activation. Similarly to NLRP6[FISNA(4A)], NLRP3[FISNA(5A)] was significantly impaired in recruiting ASC in response to *Listeria* infection, nigericin and imiquimod treatment, but this receptor also showed impaired autoactivation upon overexpression (Fig. EV2E–G).

In summary, these data show that even though NLRP3 and NLRP6 activation depend on conserved structural motifs, such as the FISNA domain and polybasic motif, the two inflammasome receptors recognize a distinct set of activators. Since the NLRP3 linker-FISNA is implicated in the spatial organization of NLRP3 in both the inactive cage oligomer and the inflammasome disc assembly, it is likely that the NLRP6 FISNA plays a similar role in the formation of active NLRP6 complexes (Andreeva et al, 2021;

Xiao et al, 2023). However, a function for the FISNA domain in signal sensing cannot be excluded from these findings.

## The NLRP6 NACHT plays an important role in receptor activation

To identify which domains within NLRP6 sense *Listeria* infection, we created a set of chimeras between NLRP6 and NLRP3, using corresponding domains in NLRP3 as place holders to generate functional receptors. We first tested if the chimeras were competent for ASC speck formation and found that all of them could induce between 40 and 50% spontaneous ASC speck formation when overexpressed in GFP-ASC$^{tg}$ HEK293T cells (Fig. EV2H). We next tested signal-specific activation upon *Listeria* infection in cells transfected with a lower amount of vector DNA, verifying equal expression of all chimeras by FACS and gating on a narrow NLR-mCherry expression level (Fig. EV2A,I). We found that exchanging the PYD, the PYD-linker or the PYD-linker-FISNA region of NLRP6 with the corresponding domains from NLRP3 had no significant impact on *Listeria*-induced ASC speck formation, nor did it confer a response to either of the NLRP3-specific activators nigericin or imiquimod. The NLRP6$^{PYD-linker-FISNA(NLRP3)}$ showed a limited response to nigericin but remained inert to imiquimod (Fig. 3D). Our data indicated that while the NLRP6 PYD-linker-FISNA region is important for ASC speck formation by mediating receptor oligomerization and the recruitment of ASC, it likely does not determine signal specificity. Therefore, *Listeria* recognition by NLRP6 must rely on either the NACHT or LRR domains, or both together. Consequently, our data also implies that NLRP3 signal specificity for nigericin and imiquimod is predominantly conferred by its NACHT-LRR region, and not the PYD-linker-FISNA region.

To determine whether the NACHT domain of NLRP6 was sufficient to confer NLRP6 signal specificity onto NLRP3, we replaced the NACHT of NLRP3 with the corresponding domain from NLRP6. The resulting NLRP3$^{NACHT(NLRP6)}$ chimera reacted to *Listeria* infection but was insensitive to nigericin and imiquimod treatment, thus mimicking the signal specificity profile of NLRP6 (Fig. 3D). This result demonstrated that the NLRP6 NACHT could confer NLRP6-like signal specificity on NLRP3, indicating that in both NLRP6 and NLRP3, specific signal detection requires their respective NACHT domains. Alternatively, *Listeria* sensing could lie within the PYD-linker-FISNA domains of both receptors and could therefore be interchanged non-specifically, while nigericin and imiquimod sensing by NLRP3 could be uncoupled and sensed by the NACHT.

To investigate if the NLRP6 LRRs also play a role in sensing of an activating signal, we generated the NLRP3$^{LRR(NLRP6)}$ receptor chimera. This protein showed ASC speck induction upon *Listeria* infection but also responded to nigericin and imiquimod treatment as observed with NLRP3 wild-type (Fig. 3D). Thus, NLRP6 LRRs can functionally substitute for NLRP3 LRRs, but do not confer NLRP6-like signal specificity. Since this indicated that the LRRs mainly serve a structural function that enhances receptor activation, we also deleted the LRRs in NLRP6. The resulting protein showed reduced responses to wild-type *Listeria* infection, yet responses were still higher than the control condition (*Δhly Listeria* infection) and the receptor was fully able to auto-activate (Figs. 3E and EV2J,K). Thus, we conclude that the NACHT domain is the cognate sensor domain of NLRP6 with an important contribution of the LRRs for full inflammasome activation.

Finally, a hallmark of NLR activation is ATP-dependent receptor oligomerization into wheel-shaped assemblies, requiring the Walker A and B motifs in the NACHT, which mediate ATP binding and hydrolysis. NLRs with mutations in either the Walker A or B motif are strongly reduced in their ability to respond to activators, but can oligomerize independently of ATP and recruit ASC upon overexpression (Brinkschulte et al, 2022; Coombs et al, 2024; Duncan et al, 2007; Sandall et al, 2020) (Fig. EV2L). Mutating the Walker A/B motifs in NLRP6 significantly reduced *Listeria*-induced ASC speck formation, indicating that ATP binding and likely the following hydrolysis is an essential step in NLRP6 activation upon signal-specific activation of the receptor (Figs. 3F and EV2M).

In conclusion, our data suggest that during *Listeria* infection, NLRP6 ligand sensing mainly maps to its NACHT domain and that ATP binding and hydrolysis are an important prerequisite for NLRP6 inflammasome formation. This implies that NLRP6 functions similarly to NLRC4 and NLRP3, where ATP-dependent receptor oligomerization into wheel-shaped inflammasome assemblies is a prerequisite for ASC speck formation (Brinkschulte et al, 2022; Duncan et al, 2007; Hu et al, 2013; Sandall et al, 2020).

## NLRP6 activation does not require bacterial PAMPs, but viable cytosolic bacteria

NLRP6 has been proposed to detect several chemically distinct ligands (LTA, RNA, or metabolites), all of which could theoretically be released by cytosolic *Listeria* (Hara et al, 2018; Levy et al, 2015; Shen et al, 2021). To define if one or several of these were detected by NLRP6, we individually evaluated their ability to activate NLRP6 in ASC-GFP$^{tg}$ HEK293T cells. Even though *Listeria* triggered NLRP6-dependent ASC speck formation, we found that transfection of the Gram-positive cell wall component LTA did not cause any ASC speck formation (Fig. 4A, 10 μg/300,000 cells), in contrast to published work in primary mouse macrophages (Hara et al, 2018). Double-stranded (ds) RNA has been shown to act as an NLRP6 ligand in vitro and to induce NLRP6 clustering and potentially inflammasome formation by liquid–liquid phase-separation (Shen et al, 2021; Wang et al, 2015). However, neither high nor low molecular weight poly(I:C) (a synthetic dsRNA analog) induced NLRP6-dependent ASC-speck formation upon transfection, even at high concentrations (Fig. 4A, 10 μg/300,000 cells). To verify that LTA and poly(I:C) were indeed delivered into cells, we transfected BODIPY-labeled LTA or fluorescein-labeled poly(I:C), respectively. Additionally, we transfected glycine-quenched BODIPY and fluorescein to control for nonspecific colocalization. While efficient delivery of LTA-BODIPY and poly(I:C)-fluorescein was detected, we found low to no colocalization with NLRP6, consistent with the inability of LTA or poly(I:C) to activate NLRP6 and induce ASC speck formation (Figs. 4B,C and EV3A,B). HEK293T cells might lack a co-factor needed for LTA- or dsRNA-induced NLRP6 activation, such as DHX15 that has been proposed to mediate NLRP6-induced IFN responses (Shen et al, 2021). Therefore, we tested LTA and poly(I:C) transfection in the HIEC-6 model, either inducing NLRP6 expression with doxycycline or not. LTA, as well as the transfection reagent Lipofectamine by itself, did not induce cell death as measured by LDH release, while poly(I:C) transfection did cause cell death, but independently of NLRP6 expression (Fig. 4D). In

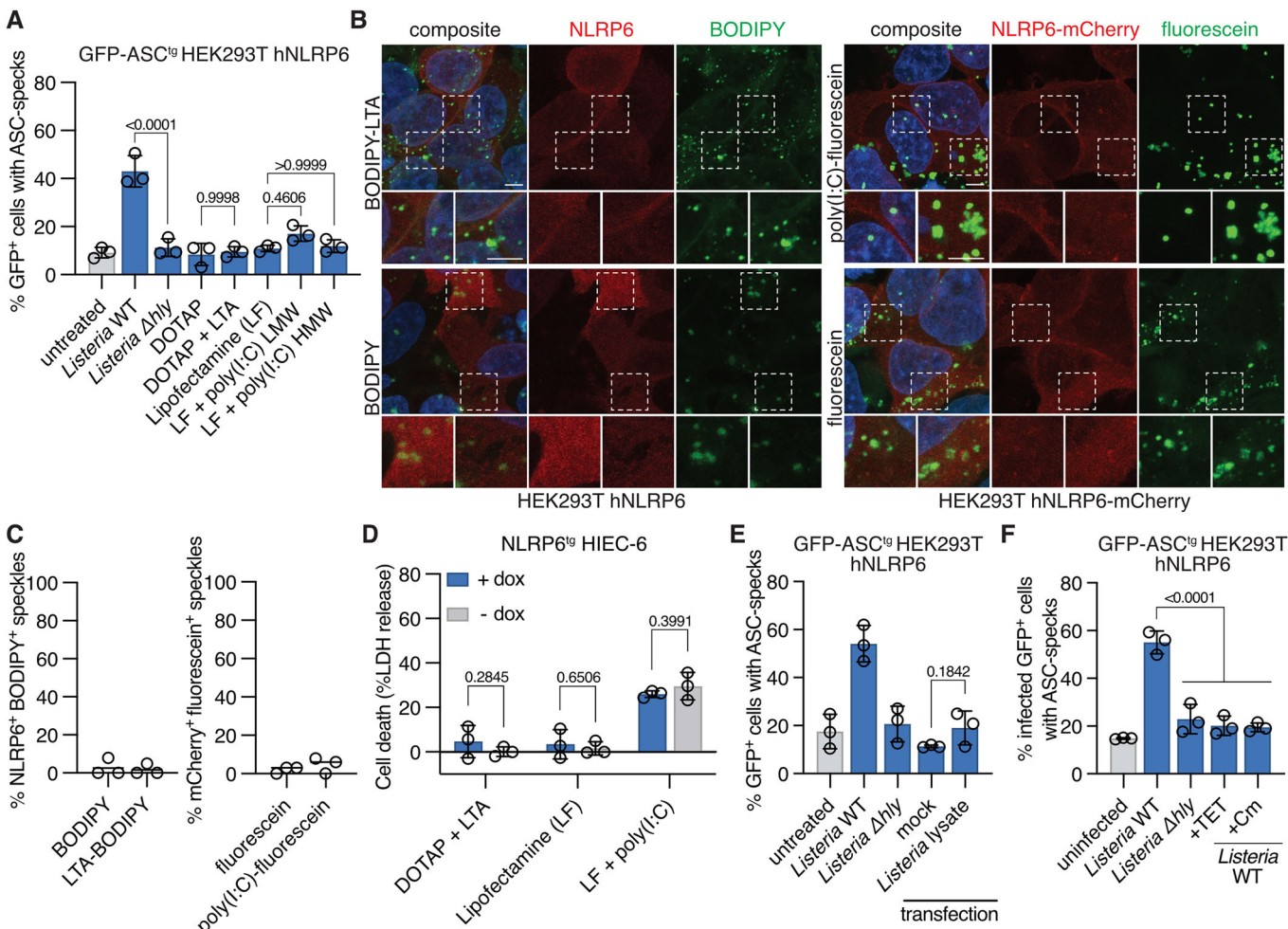

**Figure 4. NLRP6 activation does not require bacterial PAMPs, but viable cytosolic bacteria.**

(A) Flow cytometry-based quantification of ASC speck formation in hNLRP6-expressing GFP-ASC[tg] HEK293T cells either infected with WT or Δ*hly L. monocytogenes* EGD for 6 h or mock transfected with DOTAP or Lipofectamine (LF) only or transfected with LTA, poly(I:C) LMW or HMW for 24 h at 10 μg/300,000 cells. (B) Representative maximum projection confocal micrographs after transfection of hNLRP6 or hNLRP6-mCherry expressing HEK293T cells with LTA-BODIPY (7.5 μg/300,000), equivalent volume BODIPY only, poly(I:C)-fluorescein (250 ng/300,000 cells) or equivalent volume fluorescein only for 6 h. Dashed boxes indicate regions in insets. (C) Quantification of images represented in (B). For each condition, ten fields of view and at least 800 BODIPY or fluorescein speckles in NLRP6-positive cells were analyzed for NLRP6 colocalization per experiment. Three independent experiments were performed. (D) LDH release from NLRP6-WT[tg] HIEC-6 cells induced or not with 1 μg/ml doxycycline overnight and transfected with Lipofectamine (LF) only or transfected with LTA or poly(I:C) for 8 h. (E) Flow cytometry-based quantification of ASC speck formation in hNLRP6-expressing GFP-ASC[tg] HEK293T cells either infected with WT or Δ*hly L. monocytogenes* EGD, mock transfected or transfected with *L. monocytogenes* EGD lysates for 6 h. (F) Flow cytometry-based quantification of ASC speck formation in hNLRP6-expressing GFP-ASC[tg] HEK293T cells infected with WT or Δ*hly L. monocytogenes* for 6 h, in the presence of tetracycline (Tet) or chloramphenicol (Cm) added after 45 min of infection. Graphs show mean ± SD from three pooled experiments. (D) Shows mean ± SD from three pooled experiments with three technical replicates each. Each data point represents the mean of one experiment. Images are representative of three independent experiments. Scale bars represent 5 μm. Statistics used: one-way ANOVA with either Tukey's (A), Šídák's (E) or Dunnett's multiple comparisons test (F), two-way ANOVA with Tukey's multiple comparisons test (D). Source data are available online for this figure.

addition, we also tested if IFNγ priming confers responsiveness to LTA and poly(I:C) in NLRP6 expressing HIEC-6 cells but observed no NLRP6 activation in response to either PAMP under these conditions (Fig. EV3C). Finally, we also tested the bacterial metabolite Taurine, which was proposed to positively modulate NLRP6 in vivo (Elinav et al, 2011), but found that it did not induce cell death in NLRP6-expressing HIEC-6 cells (Fig. EV3D). We thus conclude that, in contrast to *Listeria* infection, none of LTA, poly(I:C) or Taurine are ligands of NLRP6 that can activate this inflammasome in our model systems.

Since we could not confirm NLRP6 activation by these published ligands, we investigated if other PAMPs that are known to be released by *Listeria* caused NLRP6 activation. Intracellular *Listeria* passively release the secondary messenger c-di-AMP, which is bound by STING to induce type I IFN (Woodward et al, 2010). We thus infected GFP-ASC[tg] HEK293T cells expressing NLRP6 with *Listeria* mutants releasing more (Δ*mdrMCAT*) or less (tetR::tn) c-di-AMP due to altered efflux pump activity but found similar levels of ASC-speck formation as with wild-type *Listeria* (Fig. EV3E). Moreover, cytosolic delivery of c-di-AMP by transfection did not induce ASC speck formation (Fig. EV3F).

Finally, we tested if a yet unknown *Listeria* ligand triggered NLRP6 activation by transfecting whole bacterial lysates, prepared by sonication, into NLRP6-expressing ASC-GFP^tg HEK293T cells. However, even the transfection of *Listeria* lysates, which contain a mixture of diverse PAMPs produced by *Listeria*, failed to induce ASC speck formation (Fig. 4E). The result suggests that NLRP6 does not recognize a *Listeria*-derived PAMP. However, at this point we are not able to exclude that NLRP6 responds to a very unstable PAMP, such as bacterial mRNA (Sander et al, 2011), that might get degraded when preparing the lysates. We next asked if the presence of live and replicating bacteria in the cytosol was required for NLRP6 activation. We thus treated *Listeria*-infected, NLRP6-expressing GFP-ASC^tg cells with the bacteriostatic antibiotics tetracycline and chloramphenicol 45 min after infection to allow primary escape from the invasion vacuole, but not continuation of the life cycle in the cytosol. To specifically assess ASC speck formation only in infected cells, we stained the cells using a *Listeria* antibody (Fig. EV3G). Antibiotic treatment both significantly reduced bacterial replication in the cytosol and NLRP6-dependent ASC speck formation in infected cells (Figs. 4F and EV3H). We thus concluded that primary invasion of the cell and escape into the cytoplasm was insufficient to activate NLRP6 to the full extent, and that live, replicating bacteria are needed for full inflammasome activation.

## NLRP6 recognizes bacterial cell-to-cell spread

Since our data indicated that live, replicating *Listeria* are required for NLRP6 activation, we next investigated if a disturbance of cellular homeostasis caused by *Listeria* infection drove NLRP6 activation. A hallmark of live cytosolic *Listeria* is the ActA-dependent polymerization of actin into characteristic actin tails that propel the bacteria forward and allow cell-to-cell spread (Travier and Lecuit, 2014). Since this causes a severe disturbance to the host cytoskeleton and endolysosomal system, we tested if cell-to-cell spread played a role in NLRP6 activation by infecting NLRP6-expressing GFP-ASC^tg cells with wild-type and ActA-deficient *Listeria*. Interestingly, we found that in comparison with cells infected with wild-type bacteria, cells infected with Δ*actA* bacteria formed significantly less ASC specks (Fig. 5A), even though Δ*actA*-infected cells contained on average higher numbers of intracellular bacteria (Fig. EV4A). Of note, we specifically analyzed the infected cell population for ASC speck formation, since the ActA-deficient mutant infects fewer cells, due to the loss of cell-to-cell spread (Fig. EV4B). Since this result indicated an important role for actin-based motility in activating NLRP6, we treated cells with the actin polymerization inhibitor Latrunculin A after invasion and found that in addition to blocking bacterial actin tail formation, it also reduced *Listeria*-induced ASC speck formation, thus phenocopying *Listeria* ActA-deficiency (Figs. 5A and EV4C). To test the importance of actin-based motility in a more physiological setting, we infected NLRP6-expressing HIEC-6 cells. ActA-deficient *Listeria* showed diminished inflammasome activation as evidenced by decreased GSDMD cleavage and cell death compared to cells infected with wild-type *Listeria*, confirming the results obtained in HEK293T cells (Fig. 5B,C and EV4D). In addition to *Listeria*, the cytosolic bacterium *S. flexneri* is well-known to use actin-based motility to spread from cell to cell (Choe and Welch, 2016). Consistently, we found that *Shigella* infection also caused NLRP6-

dependent cell death in infected HIEC-6 cells (Fig. 5D). However, the level of NLRP6-dependent death remained relatively low, likely because we observed much lower levels of infected cells and actin-positive *Shigella* than *Listeria* at similar timepoints (Fig. EV4E).

Finally, we tested if aberrant actin polymerization caused by the protein ActA itself triggered NLRP6 activation by ectopically expressing ActA in NLRP6-expressing GFP-ASC^tg HEK293T cells. Ectopically expressed ActA is reported to localize to mitochondria, where it nucleates actin (Pistor et al, 1994). However, while ActA was expressed and induced actin polymerization around mitochondria as expected, this did not induce NLRP6 activation (Figs. 5E and EV4F,G). Furthermore, we infected NLRP6-expressing GFP-ASC^tg HEK293T cells with vaccinia virus, which polymerizes host actin to promote spreading to new host cells (Choe and Welch, 2016). Vaccinia-infected cells did not show increased NLRP6 activation even though they were readily infected and featured actin-rich structures indicative of actin tails (Figs. 5F and EV4H,I). Importantly, while these virus-induced actin tails appear similar to bacteria-induced actin tails, the actual spreading of vaccinia between cells is different, as the enveloped viral particle resides extracellularly at the tip of the actin tail, as opposed to bacterial spreading, where the bacteria remain intracellular and end up in a double-membrane vacuole in the recipient cell (Welch and Way, 2013). Altogether, these results suggest that it is not actin polymerization itself, but rather the bacterial cell-to-cell spread, which is dependent on actin polymerization, that is the driver of NLRP6 activation.

In contrast to the primary infection, which involves the escape from a single-membrane endolysosome, actin-based invasion of neighboring cells (secondary infection) requires that the bacteria escape from a double-membrane vacuole formed by the plasma membranes of the cell of origin and the newly invaded cell. Since Listeriolysin O is insufficient to allow escape from such double-membrane vacuoles, *Listeria* uses its phospholipase PC-PLC (encoded by the *plcB* gene) in combination with Listeriolysin O to destroy these membranes and enter the cytosol (Fig. 5G). Consequently, Δ*plcB* bacteria are impaired in secondary vacuolar escape but not in primary invasion (Smith et al, 1995). To confirm that escape from double-membrane vacuoles during secondary infection was necessary for NLRP6 activation, we compared NLRP6-expressing GFP-ASC^tg cells infected with wild-type, Δ*hly*, Δ*actA* and Δ*plcB Listeria*, and found that Δ*plcB Listeria* induced reduced levels of ASC-speck formation in the infected cell population, mimicking the effects of ActA deficiency (Fig. 5H). Consistent with published results, we found that PC-PLC deficiency (Δ*plcB*) did not alter primary escape in the cells, as seen by the number of infected cells and actin-positive bacteria shortly after invasion. Instead, it resulted in higher levels of LAMP1-positive (membrane-bound) *Listeria* compared to the WT strain at later timepoints, due to the failure of Δ*plcB Listeria* to escape from double-membrane vacuoles during secondary infection (Fig. EV4J–L). These findings support the conclusion that escape from the double-membrane vacuole during secondary infection of cells is necessary for complete NLRP6 activation by *Listeria*. It is however important to note that ActA and PC-PLC deficiency did not reduce NLRP6 activation to the level elicited by the Listeriolysin O-deficient mutant, which indicates that while escape from the secondary vacuole is the main driver of NLRP6 activation, another event, for example, primary vacuolar escape, could still

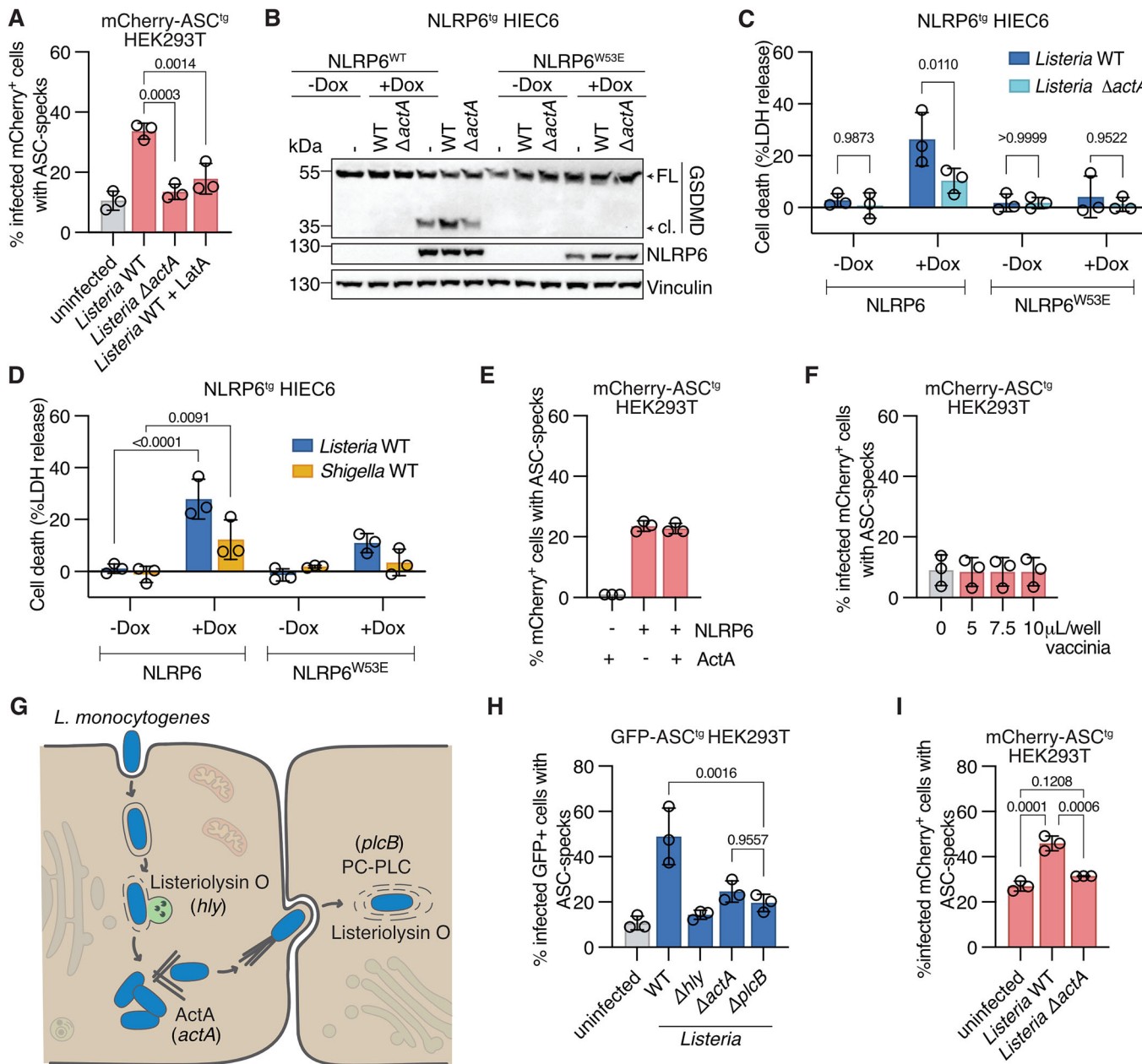

**Figure 5. NLRP6 recognizes bacterial cell-to-cell spread.**

(A) Flow cytometry-based quantification of ASC speck formation in hNLRP6-expressing mCherry-ASC[tg] HEK293T cells, infected with GFP-expressing WT or Δ*actA* L. *monocytogenes* EGD for 6 h, or infected with GFP-expressing WT *L. monocytogenes* EGD and treated with Latrunculin A (LatA) after 45 min of infection. (B, C) GSDMD processing and LDH release from NLRP6-WT[tg] or NLRP6-W53E[tg] HIEC-6 cells induced or not with 1 μg/ml doxycycline overnight and infected with WT or Δ*actA* L. *monocytogenes* EGD for 8 h. (D) LDH release from NLRP6-WT[tg] or NLRP6-W53E[tg] HIEC-6 cells induced or not with 1 μg/ml doxycycline overnight and infected with WT *L. monocytogenes* EGD or WT *Shigella flexneri* expressing Afal for 8 h. (E) Flow cytometry-based quantification of ASC speck formation in mCherry-ASC[tg] HEK293T cells expressing hNLRP6 or hNLRP6 and ActA. (F) Flow cytometry-based quantification of ASC speck formation in hNLRP6-expressing mCherry-ASC[tg] HEK293T cells, infected with the indicated volume of GFP-expressing vaccinia virus at 4.7e10 pfu/mL for 8 h. (G) Schematic drawing showing *L. monocytogenes* virulence factors needed for primary vacuolar escape (Listeriolysin O, encoded by *hly* gene), actin-based motility (ActA, encoded by *actA* gene), and secondary vacuolar escape (PC-PLC, encoded by *plcB* gene and Listeriolysin O). (H) Flow cytometry-based quantification of ASC speck formation in hNLRP6-expressing GFP-ASC[tg] HEK293T cells, infected with WT, Δ*hly*, Δ*actA*, and Δ*plcB* L. *monocytogenes* 10403S for 6 h and stained for intracellular bacteria by immunofluorescence. (I) Flow cytometry-based quantification of ASC speck formation in secondarily infected hNLRP6-expressing mCherry-ASC[tg] HEK293T cells that were incubated with infected HEK293T cells (WT or Δ*actA* L. *monocytogenes* EGD) for 14 h. (A, E, F, H) Show mean ± SD from three pooled experiments. (C, D) Show mean ± SD from three pooled experiments with three technical replicates each. Each data point represents the mean of one experiment. Blots are representative of three independent experiments. FL full length, cl. cleaved. Statistics used: one-way ANOVA with Dunnett's multiple comparisons test (A) or Šídák's multiple comparisons test (H), two-way ANOVA with Šídák's multiple comparisons test (C) or Dunnett's multiple comparisons test (D). Source data are available online for this figure.

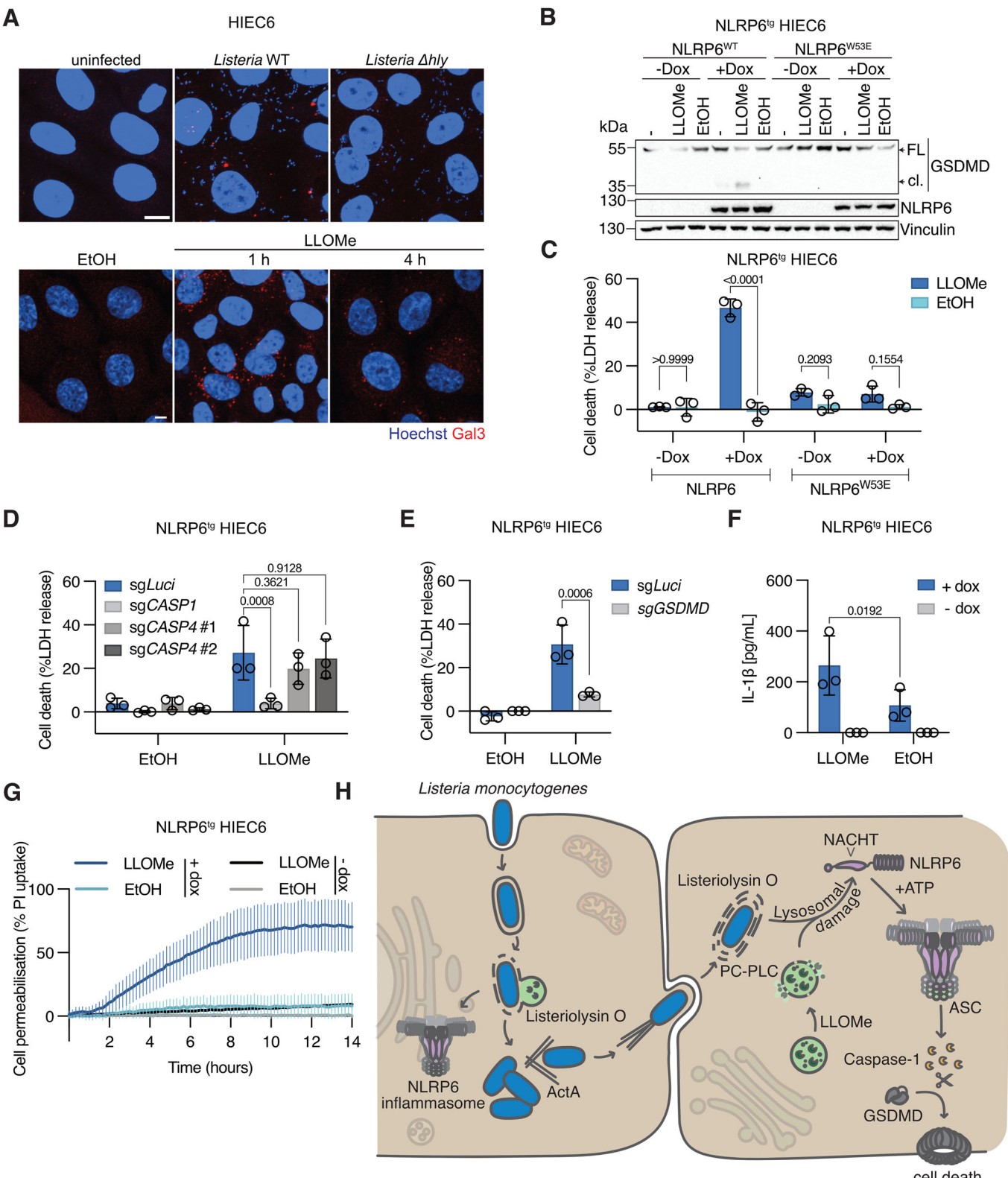

Figure 6.  Sterile endolysosomal damage is sufficient to activate NLRP6 in human intestinal epithelial cells.

(A) Confocal maximum projection micrographs of HIEC-6 cells either infected with WT or Δhly L. monocytogenes for 4 h or treated with vehicle control (ethanol, EtOH) for 4 h or 0.5 mM LLOMe for 1 h or 4 h and stained for Galectin-3. (B, C) GSDMD processing and LDH release from NLRP6-WT$^{tg}$ or NLRP6-W53E$^{tg}$ HIEC-6 cells induced or not with 1 μg/ml doxycycline overnight, left untreated or treated with 0.5 mM LLOMe or an equivalent volume of ethanol (EtOH) as vehicle control. (D, E) LDH release from NLRP6-WT$^{tg}$ HIEC-6 controls cells (sgLuci) or polyclonal populations lacking Caspase-1 (sgCASP1), Caspase-4 (sgCASP4 #1 and #2), or GSDMD (sgGSDMD) induced with 1 μg/ml doxycycline overnight and treated with 0.5 mM LLOMe or equivalent volume of ethanol (EtOH) as vehicle control. (F) IL-1β release from NLRP6-WT$^{tg}$ HIEC-6 cells induced or not with 1 μg/ml doxycycline overnight and treated with 0.5 mM LLOMe or an equivalent volume of ethanol (EtOH) as vehicle control for 4 h. (G) Time course of propidium iodide (PI) uptake of NLRP6-WT$^{tg}$ HIEC-6 cells induced or not with 1 μg/ml doxycycline overnight and treated with 0.5 mM LLOMe or equivalent volume of ethanol (EtOH). (H) Summary scheme of proposed NLRP6 inflammasome activation by endolysosomal damage caused by pathogenic bacteria or sterile triggers. (C–F) Show mean ± SD from three pooled experiments with three technical replicates each. Each data point represents the mean of one experiment. Images and blots are representative of three independent experiments. Scale bars represent 5 μm. Statistics used: two-way ANOVA with Šídák's (C, E, F) or Dunnett's (D) multiple comparisons. Source data are available online for this figure.

contribute to NLRP6 activation (Fig. 5H). To further substantiate that escape from secondary vacuoles drives NLRP6 activation, we infected HEK293T cells with Listeria WT or ΔactA for 1.5 h and then seeded the infected cells on top of mCherry-ASC$^{tg}$ HEK293T cells expressing NLRP6. This way, the only route of infection for NLRP6 and ASC-expressing cells is through cell-to-cell spread, thus allowing us to determine if secondary infections are sufficient to activate NLRP6. We found that after a reinfection period of 14 h, only WT bacteria that were able to spread from cell to cell activated NLRP6, while the immotile ΔactA mutant caused no NLRP6 activation (Fig. 5I), thus supporting our conclusion that secondary infection activates NLRP6.

## Sterile endolysosomal damage is sufficient to activate NLRP6 in human intestinal cells

Since our findings had shown that secondary infection and the escape from double-membrane vacuoles were crucial for NLRP6 activation, we wondered whether NLRP6 specifically recognizes the lysis of double-membrane vacuoles over the lysis of single-membrane vacuoles, or whether the sheer number of escaping bacteria during late timepoints triggers NLRP6 activation. In support of the latter hypothesis, NLRP6 activation was only detectable at 3-4 h post infection in both NLRP6$^{tg}$ HIEC-6 and NLRP6 expressing GFP-ASC$^{tg}$ HEK293T cells (Figs. 1B and 2E), at which time Listeria had already replicated around 10-fold (Fig. EV3H). We thus asked if other triggers of lysosomal damage could activate NLRP6 if it occurred frequently enough. L-leucyl-L-leucine methyl ester (LLOMe) is a lysosomotropic agent that polymerizes inside lysosomes to induce rapid rupture of endolysosomal compartments and is often used to study the effects of vacuolar rupture, membrane repair, and lysophagy (Eriksson et al, 2020; Radulovic et al, 2018; Thiele and Lipsky, 1990). We first tested if LLOMe induces endolysosomal permeabilization in HIEC-6 cells by staining for Galectin-3, which binds β-galactosides on the luminal side of endolysosomes and thus can serve as a marker for permeabilization (Fig. 6A) (Thurston et al, 2012). LLOMe caused endolysosomal permeabilization that was highest at 1 h post treatment and declined by 4 h, in line with published reports showing that damaged lysosomes are eventually either repaired or cleared (Fig. 6A) (Eriksson et al, 2020; Radulovic et al, 2018; Radulovic et al, 2022; Skowyra et al, 2018; Tan and Finkel, 2022). Additionally, we also looked at endolysosomal membrane permeabilization during Listeria infection and found that while the wild-type bacteria induced Galectin-3 positive endolysosomal damage,

the Listeriolysin O-deficient Δhly mutant did not (Fig. 6A). We next treated HIEC-6 cells expressing either wild-type NLRP6 or NLRP6$^{W53E}$ with LLOMe and monitored inflammasome activation at 4 h post treatment. LLOMe treatment caused robust GSDMD cleavage in cells expressing WT NLRP6, in contrast to controls not treated with doxycycline or expressing NLRP6$^{W53E}$ (Figs. 6B and EV5A). Furthermore, LLOMe-induced lytic cell death in HIEC-6 cells expressing NLRP6 but not in the respective control cells (Fig. 6C). As LLOMe-induced endolysosomal damage was previously shown to activate NLRP3 (Cullen et al, 2015; Hornung et al, 2008), we additionally treated the cells with the NLRP3 inhibitor MCC950, which did not alter cell death in the NLRP6$^{tg}$ HIEC-6, confirming once more that all cell death is due to NLRP6 activation and that NLRP6 is not inhibited by MCC950 (Fig. EV5B). Overall, this data indicates that while HIEC-6 cells are intrinsically resistant to LLOMe-induced lysosomal cell death, they could induce NLRP6-dependent pyroptosis in response to lysosomal disruption if they express NLRP6.

Next, we asked if NLRP6-dependent pyroptosis induced upon LLOMe treatment required inflammatory caspases. We thus treated NLRP6$^{tg}$ HIEC-6 expressing sgLuci, sgCASP1 or sgCASP4 with LLOMe, and found that NLRP6-dependent pyroptosis triggered by LLOMe was exclusively dependent on Caspase-1 (Fig. 6D). Using GSDMD knockout NLRP6$^{tg}$ HIEC-6, we confirmed that LLOMe-induced cell death was mediated by GSDMD-dependent pyroptosis (Fig. 6E). In addition, we found that LLOMe-treated, NLRP6-expressing HIEC-6 cells released significantly higher levels of IL-1β than cells treated with the vehicle control (Fig. 6F). Since LLOMe caused the highest levels of endolysosomal damage at 1 h (Fig. 6A), we wondered if NLRP6 activation could be detected at these early timepoints. We thus measured PI uptake over time and found that loss of plasma membrane integrity could indeed be detected as early as 1–2 h post LLOMe treatment, coinciding with high levels of Galectin-3 staining (Fig. 6G). In addition to LLOMe, we also tested other triggers of endolysosomal damage, such as the dipeptide glycyl-L-phenylalanine 2-naphthylamide (GPN), a synthetic Cathepsin C substrate that induces endolysosomal disruption upon cleavage (Berg et al, 1994), as well as monosodium urate crystals, which, once taken up into the cell, will pierce and rupture endolysosomes (Hoffstein and Weissmann, 1975). Treatment of HIEC-6 cells with GPN resulted in rapid NLRP6-dependent cell permeabilization and death (Fig. EV5E,F), while treatment with MSU crystals induced only moderate levels of plasma membrane permeabilization, likely due to the non-phagocytic nature of HIEC-6 cells. Nonetheless, this plasma membrane permeabilization was

also entirely dependent on NLRP6 (Fig. EV5G). Together, we could show that multiple triggers of sterile endolysosomal damage robustly and rapidly activate the NLRP6 inflammasome, thereby excluding a role for a bacterial PAMP in its activation.

Endolysosomal damage results in ion ($Ca^{2+}$) and proton fluxes, as well as the exposure of lipids and carbohydrates that can trigger the repair and regeneration of lysosomal membranes, or their removal by autophagy (Eriksson et al, 2020; Radulovic et al, 2018). Given the decline of Galectin-3 positive structures post LLOMe treatment, we speculated that autophagy might contribute to NLRP6 activation. We thus first assessed the formation of autophagosomes in HIEC-6 cells treated with LLOMe and found an increased size of LC3 puncta over time (Appendix Fig. S3A,B). We also found that the LC3 puncta area increased over time in Listeria-infected cells, consistent with reports that found an accumulation of pre-autophagosomal structures upon L. monocytogenes infection (Appendix Fig. S3C,D) (Tattoli et al, 2013). We next created NLRP6[tg] HIEC-6 cells lacking ATG7, the E1 enzyme that is required for LC3 lipidation, a critical step in autophagosome biogenesis, and tested the functional impact of loss of autophagy on NLRP6 activation. As expected, when treated with LLOMe or infected with Listeria, ATG7-deficient cells did not show an increase of LC3B lipidation (LC3B-II), indicating a complete inhibition of autophagy (Appendix Fig. S3E,F). However, infecting these cells with Listeria did not cause any difference in NLRP6-dependent LDH release compared to the control sgLuci cells, and NLRP6 activation by LLOMe treatment was only reduced in one of two guide RNA-expressing populations, despite both gRNAs showing knockout and blockade of LC3 lipidation (Appendix Fig. S3G,H). Altogether, these results indicate that autophagy does not play a role in NLRP6 activation.

In summary, we showed that NLRP6 does not exclusively detect cell-to-cell spread of pathogenic bacteria and the resulting vacuolar damage but can also detect general endolysosomal damage caused by sterile triggers and thus acts as a general sensor for endosomal integrity independent of bacterial PAMPs.

## Discussion

Our study demonstrates that active NLRP6 nucleates an inflammasome and induces pyroptosis in response to both sterile and pathogen-induced lysosomal damage, thus establishing NLRP6 as a sensor of HAMPs (homeostasis-altering processes) (Liston and Masters, 2017). Interestingly, we observe that during Listeria infection NLRP6 activation is rarely triggered in primary infected cells, as shown using mutant strains deficient for actin-based motility and the lysis of secondary vacuoles, but mainly activated in secondary infected cells once bacterial replication and the resulting endolysosomal damage have reached a threshold level. Such a threshold response could be necessary to avoid cell death caused by basal levels of endolysosomal damage, while maintaining the ability to respond to pathogens that replicate to high numbers within the epithelium. This might be especially important in barrier tissues like the intestinal epithelium, since excessive cell death responses could easily lead to a loss of barrier integrity. Consequently, the NLRP6 response appears to be specifically tailored towards pathogens that spread within the epithelium using actin-based motility as opposed to pathogens that replicate in an individually

infected cell, such as Salmonella. Among the pathogens that use actin-based motility for cell-to-cell spread are Listeria and Shigella, that infect the intestinal epithelium, spotted fever group (SFG) Rickettsia, that infect endothelial cells, and Burkholderia spp., which infect, among other cell types, the respiratory epithelium. The localized expression of NLRP6 in the intestinal epithelium thus correlates well with our observation that NLRP6 can detect both Listeria and Shigella in human IECs, and it will be interesting to investigate if other epithelia feature other immune sensors that are able to detect bacteria spreading between cells. Based on our data, we propose a model where the overwhelming spread of pathogens between cells, as well as pronounced lysosomal damage, generates a yet-to-be-defined signal that results in a NACHT-domain and ATP-binding dependent activation and oligomerization of NLRP6. Active NLRP6 then recruits ASC to activate Caspase-1, which cleaves GSDMD, thereby permeabilizing the plasma membrane and inducing pyroptotic cell death (Fig. 6H). Therefore, NLRP6 acts as a guard of the integrity of the endolysosomal system in IECs.

An important question that remains to be answered is what ligand or signal is sensed by NLRP6 during endolysosomal damage. The finding that sterile endolysosomal damage by LLOMe, GPN, and MSU is sufficient to activate NLRP6 in human IECs in the complete absence of bacteria and their associated PAMPs implies that NLRP6 is not activated by a specific bacterial effector, PAMP, or activity, but by endolysosomal damage itself. Damage to endolysosomes has been shown to trigger several responses that depend on the severity of the damage and range from the engagement of repair mechanisms and regenerative pathways to the removal of damaged endolysosomes by autophagy (Ferrari et al, 2024). It remains possible that one of these processes, such as the exposure of sphingomyelin, glycosylated lipids or proteins, the formation of stress granules, ubiquitination of lysosomal proteins, or the downstream recruitment of galectins or autophagy receptors, could serve as the activation signal for NLRP6. Alternatively, lysosomal enzymes such as proteases and lipases are released upon lysosomal damage, and these may generate an NLRP6 activating signal. Systematically examining the involvement of these factors or processes in follow-up studies will be important to identify the elusive NLRP6 signal. Interestingly, it is well-known that NLRP3 also reacts to endolysosomal damage, caused, for example, by LLOMe, pathogens, or the uptake of particulate matter like uric acid crystals or alum (Cullen et al, 2015; Hornung et al, 2008). However, it is likely that the endolysosomal damage-derived signal that is recognized by NLRP6 and NLRP3 is different. It has been shown that potassium efflux is a driver of NLRP3 activation following lysosomal damage in macrophages (Munoz-Planillo et al, 2013). In contrast to this, we found no activation of NLRP6 by the potassium ionophore nigericin, nor did extracellular potassium supplementation inhibit NLRP6 activation. However, this does not exclude that inhibition of retrograde trafficking or Golgi dispersion, which activate NLRP3 (Chen and Chen, 2018; Zhang et al, 2023), could serve as activators of NLRP6 as well.

The finding that NLRP6 functions as a sensor for disrupted homeostasis contradicts previous findings that suggest that NLRP6 detects specific PAMPs such as LTA, dsRNA or the bacterial metabolite taurine (Hara et al, 2018; Levy et al, 2015; Shen et al, 2021). Even though we could efficiently deliver dsRNA and LTA into cells, we were not able to detect any NLRP6 activation by these molecules, while Listeria triggered efficient NLRP6 activation in

both HEK293T and HIEC-6 cells. A possible explanation could be the lack of a co-factor that is needed for dsRNA or LTA recognition by NLRP6, in analogy to dsRNA-triggered NLRP6-induced IFN signaling that relies on the co-factor DHX-15 (Wang et al, 2015). On the other hand, it is also possible that the transfection methods previously used to deliver dsRNA or LTA into cells caused endolysosomal damage and thus resulted in NLRP6 activation. Thus, future studies on NLRP6 activators will need to consider the disruption of cellular homeostasis as a possible driver of NLRP6 activation. A role for dsRNA and LTA in driving *Listeria*-induced NLRP6 activation can further be excluded, as we have found that bacterial mutants that fail to spread between cells (e.g., Δ*actA* or Δ*plcB)* do not induce NLRP6 activation in infected cells, even though they are present at higher numbers within those cells than wild-type bacteria. This observation also excludes the possibility that NLRP6 is activated by another, potentially labile PAMP. Finally, the fact that we were able to activate NLRP6 in the absence of any bacterial PAMP by the induction of sterile endolysosomal damage shows that PAMPs are not essential for the activation of NLRP6.

Our work also allowed us to perform mechanistic studies on NLRP6, which had previously been impossible due to the lack of a simple cell-based system for NLRP6 activation. These findings show that activation of NLRP6 was crucially dependent on the ability of the receptor to bind ATP, thereby suggesting that NLRP6 assembles into ATP hydrolysis-dependent wheel-like structures like NLRP3 or NLRC4 upon activation (Brinkschulte et al, 2022; Duncan et al, 2007; Hu et al, 2013; Sandall et al, 2020). While this appears to contradict the model that NLRP6 undergoes liquid–liquid-phase separation to activate (Shen et al, 2021), we cannot exclude that LLPS-dependent activation of NLRP6 also involves the formation of wheel-shaped inflammasome complexes. Further support for the notion that NLRP6 forms complexes similarly to other NLRs comes from the fact that NLRP6 features a FISNA domain that displays sequence and structural similarity with the NLRP3 FISNA. In NLRP3, the FISNA domain was initially proposed to sense potassium efflux, and recently Xiao et al reported that the FISNA transitions from a partially disordered state to a folded state during receptor activation, which is crucially required for receptor oligomerization (Tapia-Abellan et al, 2021; Xiao et al, 2023). Our study shows that the FISNA domain is also important for NLRP6 activation, and most likely undergoes similar conformational changes during receptor activation. We were able to replace the entire PYD-linker-FISNA region of NLRP6 with the respective region of NLRP3 without changing ligand specificity. Moreover, replacing the NLRP3 NACHT with the NLRP6 NACHT conferred NLRP6-like signal specificity onto NLRP3, thus implying that for both NLRP6 and NLRP3, signal specificity lies within their respective NACHT domains. An alternative explanation for our findings would be that *Listeria* sensing lies within the PYD-linker-FISNA domains of both receptors and could therefore be interchanged non-specifically, while nigericin and imiquimod sensing could be uncoupled and sensed by the NACHT domain of NLRP3. Due to the lack of an NLRP6-only activator, we were unable to definitively determine whether sensing of imiquimod, nigericin, and *Listeria* are all mediated by the same domain.

Finally, the finding that NLRP6 is not just a sensor for pathogen infection, but a general sensor for perturbed homeostasis of the endolysosomal system could potentially link NLRP6 to a plethora of different pathological processes. Increased lysosomal membrane permeabilization has been implicated in different diseases such as lysosomal storage diseases, neurodegenerative diseases such as Parkinson's and Alzheimer's, as well as cancer (Iulianna et al, 2022; Kakuda et al, 2024; Lee et al, 2022; Micsenyi et al, 2013). The link of NLRP6 with both neuroinflammation and inflammatory bowel diseases such as Crohn's disease opens fascinating new research questions in the light of NLRP6 being a cellular homeostasis guard and its potential contribution to these malignancies (Angosto-Bazarra et al, 2022; Bracey et al, 2021).

A limitation of our study is the reliance on NLRP6 reconstitution systems, necessitated by the lack of cell-based models that endogenously express NLRP6. A plethora of different human and murine cell lines were tested for NLRP6 protein expression as part of our study, but none displayed detectable levels, even after priming with stimuli previously suggested to upregulate NLRP6 expression. Since the NLRP6 activation mechanism which we describe is distinct from previously published mechanisms, future validation in cells endogenously expressing NLRP6 and in a physiological setting, such as in organoids, primary epithelial cells or in vivo, will be important. Nevertheless, our characterization of the activation mechanism and the domains involved lays the groundwork for exploring NLRP6 function in these more complex endogenous systems, and to deciphering the contribution of NLRP6 to infection and autoinflammation.

# Methods

## Reagents and tools table

| Reagent/resource | Reference or source | Identifier or catalog number |
|---|---|---|
| **Experimental models** | | |
| *L. monocytogenes* EGD | Pascale Cossart (Institute Pasteur) | |
| *L. monocytogenes* Δ*hly* EGD | Pascale Cossart (Institute Pasteur) | |
| *L. monocytogenes* EGD GFP | Marc Lecuit (Institute Pasteur) | |
| *L. monocytogenes* Δ*actA* EGD GFP | Marc Lecuit (Institute Pasteur) | |
| *L. monocytogenes* 10403S | Daniel Portnoy (University of California) | |
| *L. monocytogenes* 10403S Δ*hly* | Daniel Portnoy (University of California) | |
| *L. monocytogenes* 10403S Δ*actA* | Daniel Portnoy (University of California) | |
| *L. monocytogenes* 10403S Δ*plcB* | Daniel Portnoy (University of California) | |
| *L. monocytogenes* 10403S Δ*mdrMCAT* | Daniel Portnoy (University of California) | |
| *L. monocytogenes* 10403S tetR::tn | Daniel Portnoy (University of California) | |
| *S. aureus* JE2 | Jan-Willem Veening | |
| *S. enterica* serovar Typhimurium | Santos et al, 2020 | |
| *S. flexneri* M90T *afal* | Jost Enninga (Institute Pasteur) | |

| Reagent/resource | Reference or source | Identifier or catalog number |
|---|---|---|
| HEK293T | ATCC | CRL-3216 |
| HIEC-6 | ATCC | CRL-3266 |
| U937 | ATCC | CRL-1593.2 |
| BSC-40 | ATCC | CRL-2761 |
| YAMC | Thomas Brunner (University of Konstanz) | PMID: 7678459 |
| mICcl2 | Thomas Brunner (University of Konstanz) | |
| HCT116 | ATCC | CCL-247 |
| HT-29 | Sigma-Aldrich | 91072201 |
| Caco-2 | ATCC | HTB-37 |
| Mino | Margot Thome (University of Lausanne) | |
| SW-1463 | Cytion | 300623 |
| Wild-type BMDM (*M. musculus*) | This study | |
| *Nlrp6-/-* BMDM (M. musculus) | Elinav et al, 2011 | |
| VACV Western Reserve EL EGFP | Florian Schmidt (University of Bonn) | |
| **Recombinant DNA** | | |
| psPAX2 | Addgene | 12260 |
| VSVG | Addgene | 12259 |
| ASC-GFP | Sino Biological | HG11175-ANG |
| pLJM1_EGFP | Addgene | 19319 |
| Human NLRP6 | Sino Biological | HG21799-UT |
| Murine NLRP6-flag | GenScript | Omu17754 |
| pINDUCER21 | Fabio Martinon (University of Lausanne) | |
| pEGFP-C2-NLRP3 | Addgene | 73955 |
| LentiCRISPRv2 | Sanjana et al, 2014 | |
| **Antibodies** | | |
| Anti-*Listeria* | Abcam | 35132 |
| Anti-NLRP6 (human) | AdipoGen | AG-20B-0046-C100 |
| Anti-NLRP6 (mouse) | AdipoGen | AG-25B-0045 |
| Anti-ASC | Santa Cruz Biotechnology | SC-22514-R |
| Anti-ASC | AdipoGen | AG-25B-0006 |
| Anti-TOMM20 | Abcam | AB565783 |
| Anti-ActA | Abnova | MAB8953 |
| Anti-LAMP1 | DSHB | H4A3-s |
| Anti-LC3 | MBL International | PM036 |
| Anti-Galectin-3 | Abcam | Ab76245 |
| Anti-Caspase-1 | AdipoGen | AG202-0948-C103 |
| Anti-Caspase-4 | Abcam | 22687 |
| Anti-GSDMD | Abcam | Ab210070 |

| Reagent/resource | Reference or source | Identifier or catalog number |
|---|---|---|
| Anti-GSDMD cleaved | Abcam | Ab215203 |
| Anti-FLAG-tag | Sigma-Aldrich | F3165 |
| Anti-IRGB10 | Jonathan Howard (Gulbenkian Institute) Steinfeldt et al, 2010 | |
| Anti-pro-IL1β | R&D systems | AF-401-NA |
| Anti-NLRP3 | AdipoGen | AG-20B-0014 |
| Anti-GBP2 | Proteintech | 11854-1-AP |
| Anti-Vinculin | Abcam | Ab91459 |
| Anti-Tubulin-HRP | Abcam | Ab40742 |
| Anti-Histone H3 | Cell Signaling Technology | 3638 |
| Anti-Actin | Cell Signaling Technology | 12262 |
| Anti-rabbit-HRP | SouthernBiotech | 4030-05 |
| Anti-mouse-HRP | SouthernBiotech | 1034-05 |
| Anti-rabbit AlexaFluor 568 | Invitrogen | A10042 |
| Anti-mouse AlexaFluor 568 | Invitrogen | A1103 |
| Anti-rabbit AlexaFluor 405 | Invitrogen | A31556 |
| Anti-rabbit AlexaFluor 488 | Invitrogen | A11008 |
| Anti-mouse AlexaFluor 488 | Invitrogen | A28175 |
| Anti-rabbit AlexaFluor 647 | Invitrogen | A32795 |
| Anti-mouse AlexaFluor 647 | Invitrogen | A21235 |
| **Oligonucleotides and other sequence-based reagents** | | |
| PCR Primers | This study | Table S1 |
| sgRNAs | This study | Table S2 |
| qPCR primers | This study | Table S3 |
| **Chemicals, enzymes, and other reagents** | | |
| BHI medium | BD | 237500 |
| TSB | BD | 211825 |
| Streptomycin | Sigma-Aldrich | S9137 |
| Ampicillin | Sigma-Aldrich | A9518 |
| Hygromycin | Invivogen | ANT-HG-1 |
| Blasticidin | Invivogen | ANT-BL-1 |
| Gentamicin | Gibco | 15710049 |
| Doxycycline | Sigma-Aldrich | D3447 |
| DMEM | Gibco | 31966021 |
| Opti-MEM | Gibco | 31985070 |
| RPMI 1640 | Gibco | 11875093 |
| EMEM | ATCC | 30-2003 |
| DMEM/F12 Ham | Gibco | 31331028 |
| McCoy's 5a | Gibco | 16600082 |
| PBS | Gibco | 14190-094 |
| HBSS | Gibco | 14025050 |
| L-Glutamine | Sigma-Aldrich | G7513 |

| Reagent/resource | Reference or source | Identifier or catalog number |
|---|---|---|
| hEGF | PeproTech | AF-100-15 |
| mEGF | PeproTech | 315-09-100UG |
| mIFNβ | BioLegend | 581302 |
| mIFNγ | PeproTech | 315-05-100UG |
| hIFNγ | PeproTech | 300-02-100UG |
| FCS | BioConcept | 2-01F30-I |
| HEPES | BioConcept | 5-31F00-H |
| NEAA | Gibco | 11140-035 |
| Pen/Strep | BioConcept | 4-01F00-H |
| ITS | Gibco | 41400045 |
| AgeI | NEB | R3552L |
| BstBI | NEB | R0519L |
| SnaBI | NEB | R0130S |
| NheI | NEB | R3131S |
| BamHI | NEB | R3136S |
| KpnI | NEB | R3142S |
| SpeI | NEB | R3133S |
| XbaI | NEB | R0145S |
| BsmBI | NEB | R0739L |
| PrimeSTAR MAX DNA Polymerase | Takara Bio | R045A |
| In-Fusion | Takara Bio | 6396650 |
| T4 Ligase | NEB | M0202L |
| XtremeGENE 9 | Roche | XTG9-RO |
| TransIT LT1 | Mirus Bio | MIR2300 |
| Polybrene | Sigma-Aldrich | TR-1003-G |
| DOTAP | Roche | 11202375001 |
| Lipofectamine 2000 | Thermos Fisher Scientific | 11668019 |
| BSA | Sigma-Aldrich | A9647 |
| Poly-Lysine | Sigma-Aldrich | P8920 |
| Collagen | Sigma-Aldrich | C4243 |
| Triton-X100 | AppliChem | A4975 |
| LLOMe | CHEM-IMPEX INT'L INC | 04578 |
| GPN | Abcam | Ab145914 |
| MSU | Invivogen | Tlrl-msu |
| LTA | Invivogen | Tlrl-pslta |
| Pam3CSK4 | Invivogen | Tlrl-pms |
| LPS | Invivogen | Tlrl-3pelps |
| Rosiglitazone | MedChemExpress | HY-17386 |
| Poly(I:C) LMW | Invivogen | Tlrl-picw |
| Poly(I:C) HMW | Invivogen | Tlrl-picf |
| fluoresceine | Sigma-Aldrich | 46950 |
| BDP FL NHS Eter | TCL | D5555 |

| Reagent/resource | Reference or source | Identifier or catalog number |
|---|---|---|
| Glycine | Sigma-Aldrich | G7126 |
| c-di-AMP | Invivogen | Tlrl-nacda |
| Lysozyme | Thermo Fisher Scientific | BP535-1 |
| EDTA | Invitrogen | 15575020 |
| Protease inhibitor | Roche | 11836170001 |
| Taurine | MedChemExpress | HY-B0351 |
| PFA | Electron Microscopy Sciences | 15710 |
| Saponin | Sigma-Aldrich | 47036 |
| LIVE/DEAD Fixable Violet dead cell stain kit 405 | Thermo Fisher Scientific | L34955 |
| Hoechst | Invitrogen | H3570 |
| CellMask deep red actin tracking stain | Invitrogen | A57245 |
| Vectashield | Adipogen | VC-H-1900-L010 |
| Trizol | Thermo Fisher Scientific | 15596026 |
| Glycogen | Thermo Fisher Scientific | AM9510 |
| DNase kit | Thermo Fisher Scientific | AM1907 |
| MMLV | Life Technologies | 28025013 |
| RnaseOut RNase Inhibitor | Invitrogen | 10777-019 |
| SyberGreen Master Mix | Roche | 04887352001 |
| Novex NuPAGE LDS Sample buffer | Invitrogen | NP0008 |
| SuperSignal West Pico PLUS chemiluminescent substrate | Thermos Fisher Scientific | 34580 |
| Propidium iodide | Life Technologies | P3566 |
| LDH release assay | Sigma-Aldrich | 11644793001 |
| Human IL-1b ELISA kit | Invitrogen | 88-7261-88 |
| T150 flasks | TPP | 90151 |
| 24-well plates | TPP | 92424 |
| 96-well plates | TPP | 92696 |
| 96-well MaxiSorp plates | Thermo Fisher Scientific | 442404 |
| Round glass coverslips 0.17 + /-0.005 mm | Biosystems Switzerland | 0117640 |
| 0.45 mm syringe filter | Sarstedt | 83.1826 |
| **Software** | | |
| Clustal Omega | www.ebi.ac.uk | |
| UniProt | www.uniprot.org | |
| FlowJo 10 | Flowjo.com | |
| GraphPad Prism v10 | www.graphpad.com | |
| Microsoft Excel | www.microsoft.com/de-ch/microsoft-365/p/excel/cfq7ttc0pbmf | |
| Fiji | Imagej.nih.gov/index/html | |
| **Other** | | |
| CytoFlex S | Beckman Coulter | |

| Reagent/resource | Reference or source | Identifier or catalog number |
|---|---|---|
| Leica DMI 6000b | Leica | |
| Zeiss LSM800 | Zeiss | |
| Transblot Turbo | BioRad | |
| iBright | Thermo Scientific | |
| Cytation 5 Imaging Reader | BioTek | |
| Epoch Plate Reader | BioTek | |
| Light Cycler 480 | Roche | |

## Bacterial strains and growth conditions

*Listeria monocytogenes* EGD wild-type and Δ*hly* were a kind gift from Dr. Pascale Cossart (Institute Pasteur, France) and were grown in BHI medium (BD #237500) at 37 °C (Santos et al, 2020). *L. monocytogenes* EGD ΔactA mutant and corresponding wild-type, expressing either GFP or not, were a kind gift from Dr. Marc Lecuit (Institute Pasteur, France) (Travier et al, 2013). *L. monocytogenes* 10403S and all its mutants were grown in BHI medium supplemented with streptomycin (50 µg/mL, Sigma-Aldrich #S9137) and were a kind gift from Dr. Daniel Portnoy (University of California, USA) (Smith et al, 1995; Woodward et al, 2010). *Staphylococcus aureus* JE2 was grown in TSB (BD #211825) at 37 °C and was a kind gift from Dr. Jan-Willem Veening (University of Lausanne, Switzerland). *Salmonella enterica* serovar Typhimurium strain SL1344 was grown in LB supplemented with 10 g/L NaCl and streptomycin (50 µg/mL) (Santos et al, 2020). *Shigella flexneri* M90T expressing the adhesin AfaI was a kind gift by Dr. Jost Enninga (Institute Pasteur, France) and was grown in TSB supplemented with ampicillin (50 µg/ml, Sigma-Aldrich #A9518) (Santos et al, 2020).

## Cell lines and culture conditions

HEK293T cells (ATCC CRL-3216) and BSC-40 (ATCC CRL-2761) were maintained in Dulbecco's modified Eagle medium (DMEM) with GlutaMAX and pyruvate (Gibco #31966021), supplemented with 10% fetal calf serum (FCS, BioConcept) at 37 °C and 5% $CO_2$. HIEC-6 cells (ATCC CRL-3266) were grown in Opti-MEM (Gibco # 31985070) supplemented with 4% FCS, 10 mM L-Glutamine (Sigma-Aldrich #G7513), and 10 ng/mL of epidermal growth factor (EGF, PeproTech #AF-100-15). U937 cells were maintained in Roswell Park Memorial Institute (RPMI) 1640 medium (Gibco # 11875093) supplemented with 10% FCS. WT and *Nlrp6*$^{-/-}$ mouse bone marrow-derived macrophages (BMDMs) from C57BL/6 male and female mice were harvested and differentiated in DMEM containing 20% L929 supernatant, as a source of macrophage colony-stimulating factor (M-CSF), 10% FCS, 10 mM HEPES (BioConcept #5-31F00-H), 1% penicillin/streptomycin (BioConcept #4-01F00-H), and nonessential amino acids (NEAA, Gibco #11140-035), and used for experiments on day 9 or 10 of differentiation. Caco2 cells (ATCC HTB-37) were maintained in Eagle's minimum essential medium supplemented with 20% FCS. HT-29 (Sigma-Aldrich #91072201) and HCT116 (ATCC CCL-247)

cells were maintained in McCoy's 5a medium supplemented with 10% FCS. YAMC and mlCcl2 cells were a gift from Thomas Brunner, University of Konstanz. YAMC were maintained in RPMI with 10% FCS, 2 mM L-glutamine, 10 µg/ml Insulin Transferrin Selenium (ITS) and 5 ng/ml IFNγ (PeproTech #315-05-100UG) at 33 °C with 5% $CO_2$. mlCcl2 were maintained in DMEM/F12 Ham (Gibco, # 11320033) with 2% FCS, 2 mM L-Glutamine, 10 µg/mL Insulin transferrin selenium (ITS, Gibco #41400045) and 10 ng/ml mEGF. Mino cells were a kind gift from Dr. Margot Thome (University of Lausanne, Switzerland) and maintained in RPMI 1640 medium supplemented with 15% FCS. SW-1463 cells (Cytion #300623) were maintained in DMEM/F12 Ham supplemented with 15 mM HEPES and 10% FCS. Unless indicated, all cell lines were grown at 37 °C with 5% $CO_2$ in humidified incubators.

## Vaccinia virus propagation

VACV Western Reserve EL EGFP was kindly provided by Dr. Florian Schmidt (University of Bonn, Germany). It was propagated on BSC-40 cells. Two 90% confluent T150 flasks (TPP #90151) were infected at MOI 0.01, and 2 days later scraped and subjected to two freeze–thaw cycles. Cellular debris was centrifuged out at $300 \times g$ for 10 min, and the supernatant was centrifuged at $15,000 \times g$ for 90 min. The viral pellet was resuspended in DMEM and titrated on BSC-40 cells, after which single-use aliquots of 4.7e10 pfu/mL were frozen at −80 °C for infection (Mercer and Helenius, 2008).

## Animals

All experiments implicating animals (mice C57CL/6, male and female 8–12 weeks) were performed under the guidelines and approval from the Swiss animal protection law (license VD3257, Service des Affaires Vétérinaires, Direction Générale de l'Agriculture, de la Viticulture et des Affaires Vétérinaires, état de Vaud). All mice were bred and housed in a specific-pathogen-free facility at 22 ± 1 C° room temperature, 55 ± 10% humidity, and a day/night cycle of 12 h/12 h at the University of Lausanne. *Nlrp6*-deficient mice have been described before (Elinav et al, 2011).

## Construction of plasmids and lentiviral transduction

All primers used in this study can be found in Appendix Table S1. ASC-mCherry was generated using an expression plasmid for ASC-GFP (Sino Biological #HG11175-ANG) as PCR template (PrimeSTAR MAX DNA Polymerase, Takara Bio #R045A) and was cloned into pLJM1_EGFP (addgene #19319) backbone, using AgeI-BstBI (New England Biolabs #R3552L and #R0519L) restriction digest and In-Fusion cloning (Takara Bio #6396650). In this pLJM1 backbone, both the CMV promoter was replaced by an EF1a promoter using SnaBI-NheI (New England Biolabs #R0130S and #R3131S) restriction digest and InFusion cloning, as well as the Puromycin resistance cassette was replaced by a Hygromycin cassette using BamHI-KpnI (New England Biolabs #R3136S and #R3142S) restriction digest and In-Fusion cloning. Human NLRP6 expression plasmid was obtained from Sino Biological (#HG21799-UT), murine NLRP6-FLAG expression plasmid was obtained from GenScript (#OMu17754). Human NLRP6 mutants for W53E, Walker A/B motives, FISNA deletion, and FISNA(4A) mutant were generated by site-directed mutagenesis PCR amplification using In-

Fusion cloning according to the manufacturer's instructions. The Walker A/B mutant was cloned in two consecutive steps, first mutating motif A and then motif B. pINDUCER21 was a kind gift from Dr. Fabio Martinon (University of Lausanne, Switzerland) (Spel et al, 2022). The gene for human NLRP6 WT or W53E was amplified by PCR from vectors described above and cloned into pINDUCER21 using the BstBI-SpeI (New England Biolabs #R3133S) restriction sites and In-Fusion cloning. Human NLRP3-mCherry constructs are derived from a pEGFP-C2-NLRP3 plasmid (addgene #73955), while human NLRP6-mCherry constructs were derived from the plasmid mentioned above. The receptors and their chimeras were cloned into pLJM1_EGFP backbones using AgeI-BstBI restriction digest and In-Fusion cloning. NLRP6-mCherry under a EF1a promoter was cloned by using AgeI-BstBI restriction digest and In-Fusion cloning in the aforementioned pLJM1_E-F1a_EGFP backbone with Puromycin resistance cassette. Successful cloning of all constructs was verified by Sanger sequencing.

To generate the stable ASC-GFP transgenic (tg) HEK293T (GFP-ASC$^{tg}$ HEK293T) cell line, human ASC-GFP expression plasmid obtained from Sino Biological was linearized by XbaI restriction digest (New England Biolabs #R0145S) and transfected into HEK293T cells using XtremeGENE 9 DNA transfection reagent (Roche #XTG9-RO). After 24 h, the cells were treated with 200 µg/mL Hygromycin (Invivogen #ANT-HG-1) to select for genome insertion. The cells were sorted using flow cytometry into a low-expressing population to avoid auto-aggregation.

For the generation of a stable mCherry-ASC$^{tg}$ HEK293T cell line, Lentiviral particles were produced from HEK293T cells. Briefly, $1 \times 10^6$ cells were transfected with 1.25 µg of plasmid, 1.25 µg psPax2 and 0.25 µg pVSV-G using TransIT LT1 transfection reagent (Mirus Bio #MIR2300). The following day media was exchanged for 4 mL DMEM + 10% FCS + 1% Bovine Serum Albumin (BSA, Sigma-Aldrich #A9647). Seventy-two hours post transfection, lentiviral particles were collected and filtered through a 0.45-µm syringe filter (Sarstedt #83.1826). In all, 1 mL Virus-containing supernatant was supplemented with 4 µg/ml Polybrene (Sigma-Aldrich #TR-1003-G) used to transduce HEK293T cells through spinfection at $500 \times g$ for 60 min. Twenty-four hours after transduction, cells were treated with 200 µg/mL Hygromycin to select for genome insertion. The cells were sorted using flow cytometry into a low-expressing population to avoid auto-aggregation.

To generate HIEC-6 stable cell lines, lentiviral particles were produced from HEK293T cells. Briefly, $3 \times 10^6$ cells were transfected with 1.9 µg of pINDUCER21, 1.9 µg psPax2 and 0.2 µg pVSV-G using TransIT LT1 transfection reagent. The following day, the media was exchanged for 4 mL DMEM + 10FBS + 1% BSA. Seventy-two hours post transfection, lentiviral particles were collected and filtered through a 0.45-µm syringe filter. Virus-containing supernatant was supplemented with 4 µg/mL Polybrene, and all 4 mL of supernatant were used to transduce HIEC-6 cells through spinfection at $500 \times g$ for 90 min. Cells were expanded, and transduced cells were sorted by flow cytometry for GFP-positive cells.

## Generation of CRISPR-Cas9 knock-out cell lines

All single-guide RNAs used in this study can be found in Appendix Table S2. CRISPR-Cas9 Knock-out cell populations were generated using the LentiCRISPRv2 plasmid (Sanjana et al, 2014) expressing Cas9 from Streptococcus pyogenes and the tracrRNA previously published (Hsu et al, 2013). The guide RNA sequences were designed using the CRISPick platform from the Broad Institute. Briefly, the vector was digested using BsmBI (New England Biolabs #R0739L) and ligated with the annealed oligonucleotides using T4 ligase system (New England Biolabs #M0202L). Plasmid sequence was verified using Sanger sequencing. Lentiviral particles were produced from HEK293T cells as described above. In all, 1 mL Virus-containing supernatant was supplemented with 4 µg/mL polybrene and used to transduce HIEC-6 cells through spinfection at $500 \times g$ for 90 min. 24 h after transduction, cells were treated with 7.5 µg/mL Blasticidin (Invivogen #ANT-BL-1) to select for genomic integration. Efficient knock-out was confirmed 7- and 10-days post transduction by immunoblot. All KO lines used in this manuscript are polyclonal populations and thus contain a mix of INDELs.

## Cell transfection

HEK293T cells were either plated onto 24-mm round glass coverslips (thickness 0.17 + /−0.005 mm, Biosystems Switzerland #0117640) coated first with poly-Lysine (0.01% poly-Lysine in water, Sigma-Aldrich #P8920), followed by collagen coating (5% collagen in PBS, Sigma-Aldrich #C4243), at a density of $7.5 \times 10^4$ cells/well for microscopy or onto collagen-coated 24-well plates (TPP #92424) at a density of $7.5 \times 10^4$ cells/well for flow cytometry or immunoblot samples 24 h before transfection. For reinfection experiments, HEK293T cells were seeded at a density of $5 \times 10^4$ cells/well in a collagen-coated 24-well plate. Cells were then transfected with one or two expression plasmids using XtremeGene 9 DNA transfection reagent for 24 h, according to the manufacturer's instructions. Human and murine NLRP6, as well as human NLRP3 containing expression plasmids and all chimeric constructs, were transfected at 80 ng/well in a 24-well plate, except for the NLRP3$^{NACHT(NLRP6)}$ and the NLRP3$^{LRR(NLRP6)}$, which were transfected at 200 and 150 ng/well, respectively.

## Cell infection and stimulation

*L. monocytogenes* grown overnight was diluted 1/15 in BHI and grown to mid-exponential phase (OD$_{600}$ = 0.6–0.8). *S. aureus* grown overnight was diluted to OD$_{600}$ = 0.4 in TSB and incubated for another hour to reach OD$_{600}$ = 1. *S.* Typhimurium grown overnight was diluted 1/50 in LB and grown to late exponential phase (OD$_{600}$ = 1.3–1.5). *S. flexneri* grown overnight was diluted 1/100 in TSB supplemented with ampicillin (50 µg/ml) and grown to mid-exponential phase (OD$_{600}$ = 0.4–0.6). Before infection, bacteria were washed and resuspended in either DMEM + 10% FCS or OptiMEM, depending on the experimental setup. *L. monocytogenes, S. aureus, S.* Typhimurium, and *S. flexneri* were added to confluent HEK293T cells at MOI 20. Plates were then centrifuged for 5 min at $300 \times g$ and incubated for 45 min (*L. monocytogenes, S. aureus, S.* Typhimurium, and *S. flexneri*) at 37 °C, 5% CO$_2$. Noninternalized bacteria were eliminated using 20 µg/mL Gentamicin (*L. monocytogenes*, Gibco #15710049), 100 µg/mL Gentamicin (*S.* Typhimurium and *S. flexneri*) or 2 µg/mL Lysostaphin (*S. aureus*, Sigma-Aldrich #L7386). For viral infection with vaccinia virus, cell medium was replaced for OptiMEM, and the desired volume of viral suspension was added as indicated (titer 4.7e10 pfu/mL). The cells were incubated for one hour at 37 °C, 5% CO$_2$, after which the

cell medium was replaced with DMEM + 10% FCS. At desired time points of infection, cells were processed for flow cytometry, Immunoblot, or microscopy. For CFU measurements, at desired time points, cells were washed with PBS, lysed with 0.2% Triton-X100 (AppliChem #A4975) in PBS (Gibco #14190-094), serially diluted, and plated in duplicates on BHI agar. The agar plates were incubated for 24 h at 37 °C, after which colonies were counted.

For infection of HIEC-6 cells, $4 \times 10^4$ cells/well were seeded in a 96-well plate (TPP #92696) and the following day, NLRP6 expression was induced overnight using 1 µg/mL doxycycline (Sigma-Aldrich #D3447). Bacteria were cultured as stated above. Before infection, bacteria were washed and resuspended in OptiMEM. *L. monocytogenes* and *S. flexneri* were added at MOI of 20. Plates were then centrifuged for 5 min at $300 \times g$ and incubated for one hour at 37 °C, 5% $CO_2$. Noninternalized bacteria were eliminated using 20 µg/mL Gentamicin (*L. monocytogenes*) or 100 µg/mL Gentamicin (*S. flexneri*). At desired time points of infection, cells and/or cell supernatants were processed for immunoblot or cell permeabilization/death analysis. To induce endolysosomal membrane damage, NLRP6-expressing cells were treated with 0.5 mM LLOMe (CHEM-IMPEX INT'L INC #04578), 200 µM GPN (Abcam #ab145914), 100 µM MSU (Invivogen #tlrl-msu) for the indicated time points, after which cells and/or cell supernatants were processed for immunoblot or cell permeabilization/death analysis. Where indicated, cells were treated with 10 µM MCC950 (Invivogen #inh-mcc) for 30 min prior to stimulation with *L. monocytogenes* or LLOMe in the presence of 10 µM MCC950.

For reinfection experiments, *Listeria* were cultured as stated above, washed, and resuspended in DMEM + 10% FCS and added to HEK293T cells at an MOI of 20. Plates were centrifuged for 5 min at $300 \times g$ and incubated for 45 min at 37 °C, 5% $CO_2$. Noninternalized bacteria were eliminated using 20 µg/mL gentamicin, and the cells were incubated for another 45 min. The infected cells were then washed with PBS + 20 µg/mL gentamicin and detached with Trypsin. After detachment, cells were centrifuged for 3 min at $500 \times g$ and resuspended in DMEM + 10% FCS + 20 µg/mL Gentamicin. The infected HEK293T cells were added to uninfected mCherry-ASC$^{tg}$ HEK293T cells (transfected 24 h prior with NLRP6) at a ratio of 1:1.5 (uninfected: infected) in DMEM + 10% FCS + 20 µg/mL Gentamicin, centrifuged for 5 min at $300 \times g$ and incubated for 14 h at 37 °C, 5% $CO_2$, after which they were processed for flow cytometry.

For ligand stimulation studies, unlabeled LTA (from *S. aureus*, Invivogen #tlrl-pslta) was transfected into cells at 10 µg/300,000 cells, while BODIPY labeled LTA (labeling protocol below) was transfected at a theoretical concentration of 7.5 µg/300,000 cells using DOTAP transfection reagent (Roche #11202375001). Glycine-quenched BODIPY was transfected in equal volume. Briefly, the required volume LTA, LTA-BODIPY or BODIPY, and 15µL DOTAP were separately diluted to 28 µL in HBSS, mixed, and incubated for five minutes, after which the LTA suspension was mixed with the diluted DOTAP, and the mixture was brought to 200 µL Hanks Buffered Salt Solution (HBSS, Gibco #14025050). Of this mixture, 66.7 µL/300,000 cells was added in each well on a 24-well or 96-well plate. For transfection with dsRNA, cells were transfected with 10 µg/300,000 cells unlabeled LMW (Invivogen #tlrl-picw) or HMW poly(I:C) (Invivogen #tlrl-pic) or 0.25 µg/300,000 cells for fluorescein-labeled HMW poly(I:C) (Invivogen #tlrl-picf) or glycine-quenched fluorescein (Sigma-Aldrich #46950) in equal volume using Lipofectamine 2000 (Thermo Fisher Scientific #11668019). Briefly, 1.5 µL Lipofectamine and the required volume poly(I:C) were separately diluted to 25 µL in OptiMEM and incubated for 5 min, after which the diluted poly(I:C) was added to the Lipofectamine suspension, mixed and incubated for 20 min. Of this mixture, 25 µL/300,000 cells was added for a 24- or 96-well plate. Transfection of c-di-AMP (Invivogen #tlrl-nacda) was performed at 10 µg/300,000 cells using Lipofectamine 2000 as described above. *L. monocytogenes* lysate was prepared as previously published (Hara et al, 2018). Briefly, $10^9$ bacteria grown overnight in BHI broth were pelleted, washed in 1 mL PBS, incubated with 100 µg/mL lysozyme (Thermo Fisher Scientific #BP535-1), 5 mM EDTA (Invitrogen #15575020), and protease inhibitor (Roche #11836170001) for 30 min at room temperature, after which the solution was sonicated and centrifuged for one minute at $21,000 \times g$. The supernatant was then used as bacterial lysate. Taurine (MedChemExpress #HY-B0351) was used at 70 mM in PBS and added to the cells.

For cell priming, if not indicated otherwise, 100 ng/ml mIFNβ (BioLegend #581302), 10 ng/ml IFNγ (murine: PeproTech 315.05-100UG, human: PeproTech #300-02-100UG), 300 ng/mL Pam3CSK4 (Invivogen #Tlrl-pms), 100 ng/mL LPS (Invivogen #Tlrl-3pelps), 100 ng/mL poly(I:C) (Invivogen #Tlrl-pcw), 100 µM rosiglitazone (MedChemExpress #HY-17386) was used overnight.

## Quantification of ASC-speck formation by flow cytometry

For flow cytometry analysis, supernatant and trypsinized cells were collected by centrifugation, washed in PBS, and fixed in 4% Paraformaldehyde (PFA, Electron Microscopy Sciences #15710) for 10 min. If indicated, cells were permeabilized using PBS + 0.1% BSA + 0.1% saponin (Sigma-Aldrich #47036) and intracellularly stained for *L. monocytogenes* (abcam #ab35132), followed by labeling of secondary antibody coupled to fluorophore (either Alexa fluor 568, Invitrogen #A10042 or Alexa fluor 405, Invitrogen #A31556). When indicated, live/dead staining was performed using the LIVE/DEAD™ Fixable Violet Dead Cell Stain Kit for 405 nm excitation (Thermo Fisher Scientific #L34955) according to the manufacturer's protocol.

For flow cytometry, a CytoFLEX machine from Beckman Coulter was used. Cells were determined by forward and side scatter profile, 15,000 events were gated based on singlets (SSC-H vs. SSC-A). Cells were then gated for ASC-GFP or -mCherry expressors, and if applicable for NLR-receptor expression based on the mCherry signal, live or dead cells based on the live/dead stain, or infected cells based on the intracellular staining (Alexa fluor 568 or 405) or GFP signal. ASC-speck formation was assessed in these gated cells by gating on GFP/mCherry-H vs. -A and selecting the shift towards higher signal according to previously published methods (Sester et al, 2015). At least 1000 cells per condition and experiment were analyzed for ASC-speck formation.

## Microscopy, time-lapse imaging, and image analysis

For fluorescent microscopy of fixed samples, cells were fixed with 4% PFA for ten minutes. Fixed cells were permeabilized using PBS + 0.1% BSA + 0.1% saponin. Cells were then incubated 1:100

in permeabilization buffer with either anti-*L. monocytogenes*, anti-NLRP6 (AdipoGen #AG-20B-0046-C100), anti-ASC (Santa Cruz Biotechnology #SC-22514-R), anti-TOMM20 (Abcam #AB56783), anti-ActA (Abnova #MAB8953), anti-LAMP1 (DSHB #H4A3-s), anti-LC3 (MBL International #PM036) or anti-Galectin-3 for one hour, washed three times with PBS and incubated with secondary antibody coupled to Alexa fluor-488, 568 or -647 at 2 µg/mL (Invitrogen, 488: #A11008 and #A28175, 568: Invitrogen #A10042 and #A1103, 647: Invitrogen #A32795 and #A21235). DNA was stained using Hoechst (1:5000, Invitrogen #H3570), while actin was stained using CellMask™ deep red actin tracking stain (Invitrogen #A57245), and samples coverslips were mounted in Vectashield mounting media (AdipoGen #VC-H-1900-L010). Samples were analyzed with a LEICA DMI 6000b wide field microscope using a 40×/1.25-0.75 NA (oil) objective or a Zeiss LSM800 confocal laser scanning microscope using a 40×/1.3 NA (oil) or 63×/1.4 NA (oil) objective, if required, acquiring Z-stacks with a 1-µm step size.

Images were analyzed, quantified, and processed using Fiji software. Below, procedure for quantification is described in more detail.

### Actin-positive bacteria

For Fig. 1F, the total number of bacteria was enumerated from maximum projections of the *Listeria* image using background subtraction with a rolling ball of 10. A manual threshold was determined to select bacteria; the same threshold was used for all images in one experiment. Next, the Watershed function was used to separate touching objects. The Analyze Particles function was used to quantify the number of total bacteria using a size constraint of 80-Infinity pixels and circularity = 0.30–1.00. Lastly, Actin-positive bacteria were enumerated manually.

For Fig. EV4L, actin-positive bacteria were quantified from a single middle plane of the Z-stack images. The total number of *Listeria* were enumerated with the same functions as described above. To identify actin-positive bacteria, the mean pixel intensity of the actin stain was determined in the area defined by the bacterial mask. The mean pixel intensity to be considered actin-positive was set using the highest values of the ActA-deficient *Listeria* at 6 hpi.

### ASC speck quantification

ASC specks were quantified from maximum projections of the Z-stack images. The total number of cells was enumerated by counting nuclei stained with Hoechst, using a Median Blur with a radius = 7. A manual threshold was determined to select nuclei; the same threshold was used for all images in one experiment. The Fill Holes and Watershed functions were used to process a uniform mask and separate touching objects. The Analyze Particles function was used to quantify the number of nuclei using a size constraint of 30-Infinity. ASC specks were quantified by using a Median Blur with a radius = 4 and setting a manual threshold; the same threshold was used for all images in one experiment. The Analyze Particles function was used to quantify the number of total ASC specks using a size constraint of 0.1-Infinity and circularity = 0.70-1.00. ASC-specks per cell were enumerated by manually correcting for nuclei associated with multiple ASC specks.

### LTA and poly(I:C) colocalization with NLRP6

For LTA-BODIPY, BODIPY, poly(I:C)-fluorescein, and fluorescein colocalization with NLRP6(-mCherry), NLRP6-positive cells were segmented using a Gaussian Blur with sigma = 7 and setting a

manual threshold. Next, the mask was homogenized using the Fill Holes function. Within each region of interest (ROI) defined by NLRP6 positivity, particles of LTA or poly(I:C) were identified using a Gaussian Blur with sigma = 2 and a manual threshold. To identify colocalization, the mean pixel intensity of the NLRP6 signal was determined in the area defined by the particle mask. The mean pixel intensity to be considered NLRP6 positive was set by manually checking the NLRP6 signal values for the particles. For each quantification, three independent experiments were used with each ten fields of view. At least 18 ROIs per condition and experiment were defined according to the manual NLRP6 positivity threshold, which could be comprised multiple cells due to touching of the cells. At least 800 BODIPY or fluorescein positive speckles were counted and analyzed per condition and experiment.

### LC3 puncta analysis

Maximum projections of LC3 image Z-stacks were processed using a background subtraction with a rolling ball of 50 and a Gaussian Blur with sigma = 1. Afterwards, a manual threshold was determined to select LC3 puncta; the same threshold was used for all images in one experiment. Next, the Analyze Particles function was used to calculate the area of each punctum. The mean area of the puncta in each field of view was obtained from the "Summary" of results.

### LAMP1-positive bacteria

LAMP1-positive bacteria were quantified from maximum projections of the Z-stack images. The total number of *Listeria* were enumerated with the same functions as described above. To identify LAMP1-positive bacteria, the mean pixel intensity of the LAMP1 staining was determined in the area defined by the bacterial mask. The mean pixel intensity to be considered LAMP1-positive was set using the highest values of the ActA-deficient *Listeria* at 6 hpi.

## Immunoblot analysis

Cells were lysed with 1× LDS NuPage sample buffer diluted in 66 mM Tris-HCl pH 7.4 with 2% SDS, supplemented with 5% β-mercaptoethanol or 10 mM DTT. For GSDMD cleavage immunoblots, proteins were precipitated from cell supernatants using chloroform and methanol. 1 volume of MeOH and 0.3 volumes of chloroform were added to the supernatant, vortexed, then centrifuged at $14{,}000 \times g$ at 4 °C for 10 min. The upper, aqueous phase was discarded and 1.3 volumes of MeOH added before being vortexed and centrifuged at $14{,}000 \times g$ at 4 °C for 10 min. Supernatant was aspirated, and protein pellets allowed to air-dry before being resuspended in the cell lysates from the corresponding wells. Samples were boiled at 95 °C for 7 min then separated on 12% Tris-Gly SDS-PAGE gels and transferred to nitrocellulose membranes. Membranes were blocked in 5% milk in TBS-T then incubated with primary antibodies for human NLRP6, murine NLRP6 (murine antibody purified from rabbit serum, inoculated with recombinant peptide, produced by YenZym. Now available from AdipoGen #AG-25B-0045), ASC (AdipoGen #AG-25B-0006), Caspase-1 (AdipoGen #AG202-0948-C103), Caspase-4 (Abcam #ab22687), GSDMD (Abcam full length: #ab210070, cleaved #ab215203), FLAG-tag (Sigma-Aldrich #F3165), IRGB10 (kind gift from Dr.

Jonathan Howard, Gulbenkian Institute of Science) (Steinfeldt et al, 2010), pro-IL1β (R&D Systems #AF-401-NA), NLRP3 (Adipogen #AG-20B-0014), GBP2 (Proteintech #11854-1-AP), Vinculin (Abcam #ab91459), Tubulin (HRP conjugated, Abcam #ab40742), Histone H3 (Cell Signaling Technology #3638), Actin (Cell Signaling Technology #12262) overnight at 4 °C (all antibodies at 1:1000, except murine NLRP6 at 1:2000 and Tubulin-HRP at 1:5000). Membranes were washed in TBS-T before incubation with HRP conjugated secondary antibodies (SouthernBiotech #4030-05 and #1034-05) and finally developed using chemiluminescence (SuperSignal West Pico PLUS Chemiluminescent Substrate, Thermo Fisher Scientific #34580) on an iBright digital imaging system.

For quantification of GSDMD cleavage in Western blots, images were exported as TIFFs and imported into ImageJ, and densitometry analysis was performed on bands corresponding to full-length and cleaved GSDMD as well as the loading control. The signal of the full-length and cleaved GSDMD bands was normalized to the signal of the respective loading control by division. The ratio of cleaved:full-length GSDMD was then calculated using these normalised values. This ratio was then normalised to that of the UT control.

## Cell permeabilization and death analysis

Cell membrane permeabilization was measured by PI uptake assay. After one hour of infection, cell medium was replaced by OptiMEM + 20 μg/mL Gentamicin + 12.5 μg/mL propidium iodide. The lysis control cells were lysed using 0.1% Triton-X100. 0.5 mM LLOMe, equivalent volume ethanol, 200 μM GPN or the equivalent volume DMSO or 100 mM MSU was added in OptiMEM + 12.5 μg/mL propidium iodide. Red fluorescence was measured using a BioTek Cytation 5 plate reader with Gen5 software at 37 °C, 5% $CO_2$, reading every ten minutes. The percentage of PI uptake was calculated using the formula:

$$\frac{Fluorescence\ sample - Fluorescence\ untreated\ control}{Fluorescence\ 100\%\ lysis\ control - Fluorescence\ untreated\ control}.$$

Lytic cell death was measured by LDH release assay (Sigma-Aldrich #11644793001). In total, 25 μL of cell culture supernatant was incubated with 25 μL of LDH assay reagent for 10–15 min before absorbance was measured at 490 nm on a BioTek Epoch plate reader. 100% lysis control was prepared by adding a final concentration of 0.1% Triton-X to the corresponding wells. All conditions were performed in technical triplicate and the percentage of LDH released calculated using the formula:

$$\frac{Absorbance\ sample - Absorbance\ untreated\ control}{Absorbance\ 100\%\ lysis\ control - Absorbance\ untreated\ control}.$$

## IL-1β ELISA

The levels of IL-1*β* were measured by ELISA (Invitrogen # 88-7261-88) according to the manufacturer's protocol. Briefly, MicroWell MaxiSorp flat-bottom plates (Thermo Fisher Scientific #442404) were coated with capture antibody and incubated overnight at 4 °C. The wells were washed three times with PBS + 0.05% Tween20 and blocked with ELISA/ELISPOT diluent for one hour at room temperature. After washing, twofold serial dilutions of the standard were performed for eight points, and samples were diluted 1:10.

The plates were incubated at room temperature for two hours, after which the plates were washed and incubated with detection antibody for one hour. After washing, Avidin-HRP was added to the wells, incubated for 30 min at room temperature, and washed again. TMB solution was added, and the reaction was stopped with stop solution. The plate was read at 450 nm, and background at 570 nm was subtracted from the values.

## qPCR

RNA was extracted from tissue samples and cell pellets using TRIzol (Thermo Fisher Scientific #15596026) following the manufacturer's protocol. In all, 5 μL glycogen (Thermo Fisher Scientific #AM9510) was added per sample during RNA precipitation and two ethanol washes were performed. RNA was then DNase treated with TURBO DNA-free kit (Thermo Fisher Scientific #AM1907) and equal amounts (2 μg) were reverse transcribed with MMLV (Life Technologies #28025013) following the manufacturer's protocol using RnaseOut RNase Inhibitor (Invitrogen #10777-019). qPCR for each biological replicate was performed in technical triplicate on a LightCycler LC480 II (Roche) with gene-specific primers using SyberGreen Master Mix (Roche #04887352001. All results are normalized to 18S levels, and the NLRP6 expressing control condition is set to 1.qPCR primers used for this study can be found in Appendix Table S3.

## Protein sequence alignments

Protein sequences for NLRP6 and NLRP3 from different species were obtained from UniProt (UniProtKB accession numbers Q8R4B8, Q96P20, P59044 and Q91WS2) and a multiple protein sequence alignment was generated using Clustal Omega (www.ebi.ac.uk). Domains were identified and delineated based on previous publications (Shen et al, 2021; Shen et al, 2019; Tapia-Abellan et al, 2021) or homology to NLRP3.

## LTA labeling with BODIPY

Labeling protocol for LTA was adapted from a previous publication (Hara et al, 2018). LTA was prepared as 5 mg/mL stock in endotoxin free water. In all, 500 μL of this stock was diluted to 1.25 mL and sonicated for ten minutes at 70% power on ice. After, 1.25 mL 200 mM Sodium Bicarbonate was added to a final LTA concentration of 1 mg/mL (125uM). BDP FL NHS Ester (TCI, D5555) was added gradually to reach molar ratio of LTA to dye of 1:4. The solution was allowed to react for 1.5 h at room temperature in the dark, after which the mixture was centrifuged at 15,000 rpm for 10 min to remove precipitants. The staining reaction was quenched with 2.5 mM Glycine (Sigma-Aldrich #G7126) at neutral pH for 30 min. Gel filtration of the quenched solution was performed on a Superdex 200 column and collected in the void fractions. From the final volume collected and pooled after gel filtration, concentration of labeled LTA was calculated to be 100 μM (0.8 mg/mL). The fractions containing the quenched BODIPY was used as a negative control for the transfection of cells with fluorophore only.

### Experimental study design

There was no randomization for these experiments. This study is not a randomized control trial, and randomization is not conventionally

used in in cellulo studies such as this one. All groups of experiments were performed using the same protocols and experimental conditions. Assays used objective quantification methods that are not susceptible to bias, so samples were not blinded.

### Quantification and statistical analysis

Data were verified for normality by graphical methods, and analysis was performed using FlowJo 10, GraphPad Prism v10, Microsoft Excel, and Fiji software. Statistical significance is given in numerical $P$ values. For comparison of two individual groups, a two-tailed, unpaired $t$ test was used, while for comparison of three or more groups, one-way ANOVA or two-way ANOVA was used with Dunnett's/Tukey's/Šídák's multiple comparisons test as indicated. The number of independent biological replicates as well as the number of events quantified for microscopy analysis is indicated in the figure legend.

## Data availability

Our study includes no data deposited in public repositories. Data and resources used in this study are available from the lead contact (petr.broz@unil.ch).

The source data of this paper are collected in the following database record: biostudies:S-SCDT-10_1038-S44318-025-00637-4.

## Peer review information

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

## Acknowledgements

We would like to thank Dr. Pascale Cossart, Dr. Marc Lecuit, Dr. Daniel Portnoy, Dr. Jan-Willem Veening, Dr. Jost Enninga, Dr. Florian Schmidt, and Dr. Fabio Martinon for providing bacterial or viral strains, as well as constructs. We would also like to thank Dr. Jakub Began for his help with LTA BODIPY labeling and Lukas Bissegger for support with cloning and cell culture. We would also like to acknowledge the support from the UNIL imaging (CIF), FACS, and animal core facilities. Further thanks go to the rest of the current and former members of the Broz laboratory for constructive discussions and support. This work was supported by Swiss National Science Foundation Project funding (310030B_219286, 310030B_192523) to PB.

## Author contributions

**Alexandra Boegli**: Conceptualization; Formal analysis; Supervision; Investigation; Visualization; Methodology; Writing—original draft. **Elliott M Bernard**: Conceptualization; Formal analysis; Investigation; Visualization; Methodology; Writing—review and editing. **Louise Lacante**: Formal analysis; Investigation; Visualization; Methodology; Writing—review and editing. **Gaël Majeux**: Formal analysis; Investigation; Methodology. **Ella Hartenian**: Resources. **Vanessa Mack**: Resources. **Petr Broz**: Conceptualization; Supervision; Funding acquisition; Writing—original draft; Project administration.

Source data underlying figure panels in this paper may have individual authorship assigned. Where available, figure panel/source data authorship is listed in the following database record: biostudies:S-SCDT-10_1038-S44318-025-00637-4.

## Disclosure and competing interests statement

The authors declare no competing interests.

# Expanded View Figures

**Figure EV1. NLRP6 detects the entry of *L. monocytogenes* into the host cell cytosol.**

(**A**) Flow cytometry gating strategy used to quantify ASC speck formation. (**B**) ASC speck formation quantified by flow cytometry and corresponding NLRP6 expression of GFP-ASC[tg] HEK293T cells transfected with hNLRP6 DNA at indicated concentrations for 24 h. (**C**) ASC speck formation in hNLRP6-expressing GFP-ASC[tg] HEK293T cells infected with *L. monocytogenes* strain EGD or 10403S, *S. aureus* or *S.* Typhimurium for 6 h. (**D**) ASC speck formation and corresponding NLRP6 expression of GFP-ASC[tg] HEK293T cells transfected with mNLRP6 DNA at indicated concentrations for 24 h (**E**). Representative images of actin positive, intracellular *Listeria* after 1 h of infection in HEK293T cells left untreated or treated with bafilomycin A1 (bafA) for 2 h and infected with WT or Δ*hly L. monocytogenes* EGD for 6 h. *Listeria* were stained by immunofluorescence and Actin was stained with CellMask Actin Stain. Actin colocalization is indicated by arrowheads (**F**). Propidium iodide (PI) uptake of HEK293T cells after treatment with purified Listeriolysin O at indicated concentrations over time (**G**). Percentage of cells with permeabilized membranes among cells containing an ASC speck, hNLRP6-expressing GFP-ASC[tg] HEK293T cells infected with *L. monocytogenes* EGD for 4, 6 or 8 h as measured by flow cytometry LIVE/DEAD staining. Graphs show means ± SD from at three pooled independent experiments, except panel F, which is representative of two independent experiments. Images are representative of three independent experiments, immunoblots are representative of two independent experiments. Scale bar represents 1 μm. Statistics used: one-way ANOVA with Dunnett's multiple comparisons.

▶

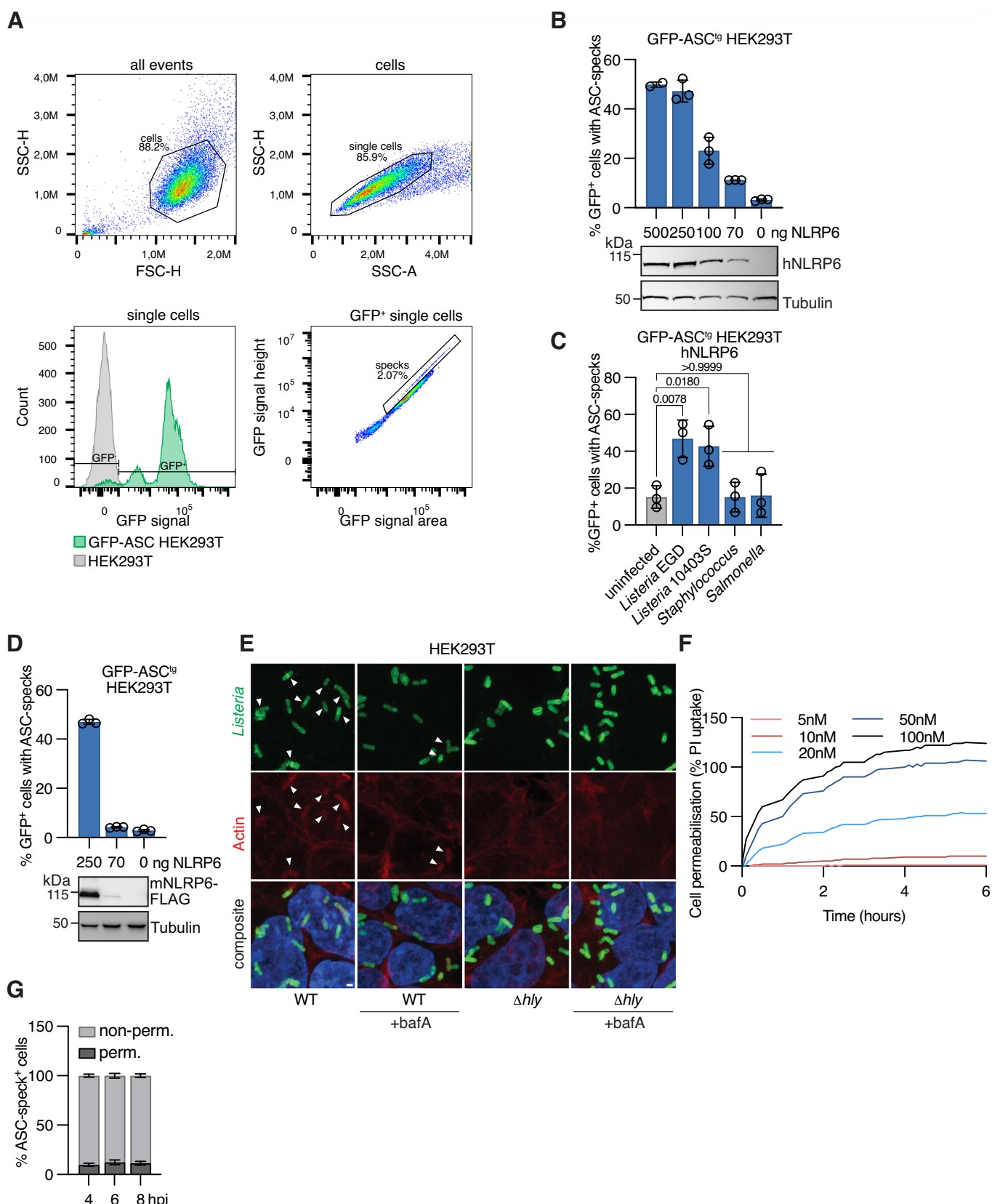

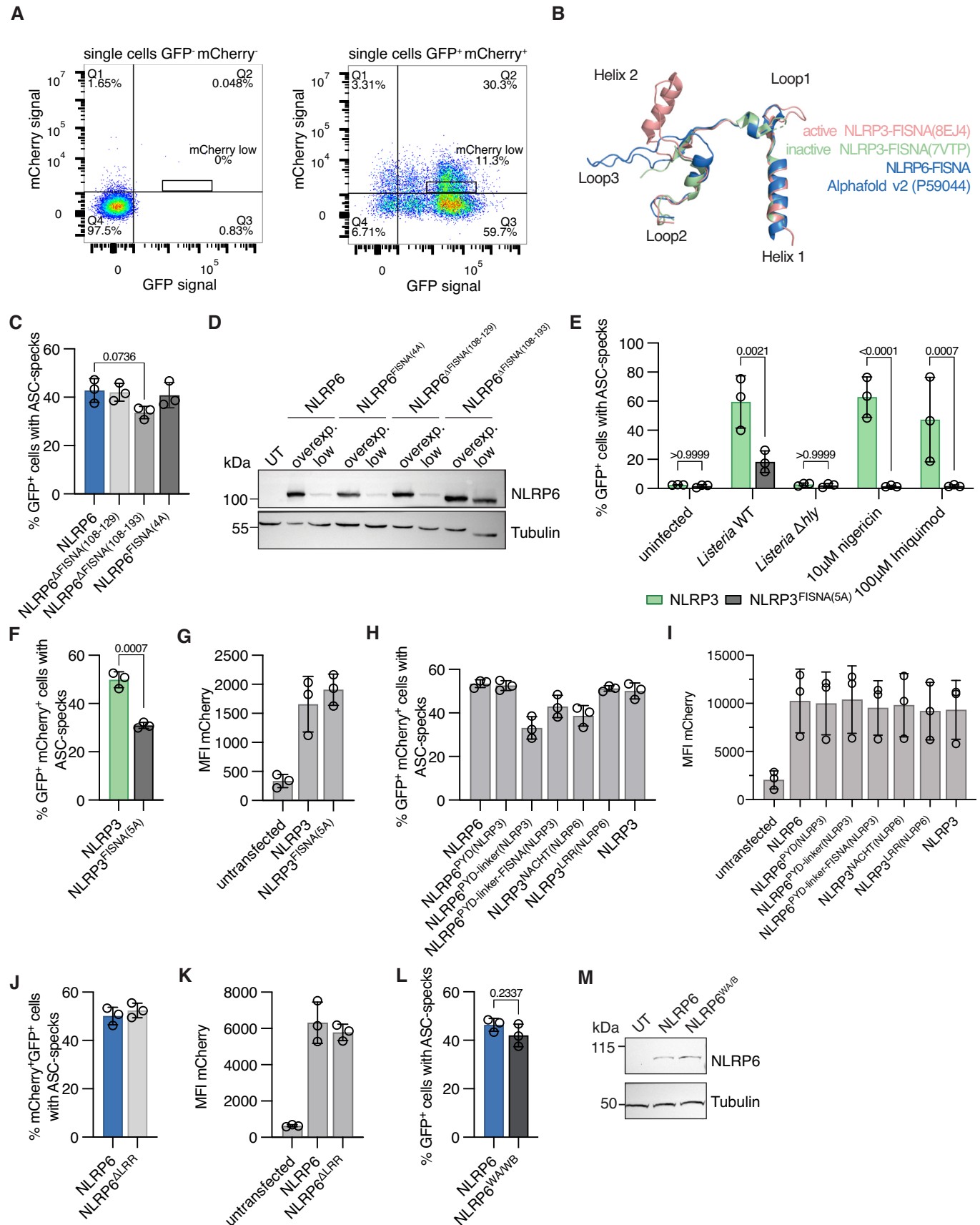

◀ **Figure EV2.  Molecular characterization of the NLRP6 receptor.**

(A) Flow cytometry gating strategy used for mCherry-tagged NLRs. (B) Structural alignment of the FISNA domain of active and inactive NLRP3 (8EJ4 and 7VTP) with the NLRP6-FISNA predicted by Alphafold v2 (P59044). (C) Flow cytometry-based quantification of ASC speck formation in GFP-ASC[tg] HEK293T cells overexpressing the indicated NLRP6 proteins at 250 ng DNA/well (D). Immunoblot showing expression levels of constructs depicted in C expressed at either 250 ng/well (overexp.) or 80 ng/well (low). Tubulin serves as loading control. (E) Flow cytometry-based quantification of ASC speck formation in GFP-ASC[tg] HEK293T cells, expressing the indicated NLRP3-mCherry proteins, infected with WT or Δ*hly L. monocytogenes* for 6 h or treated with 10 μM nigericin or 100 μM imiquimod for 1 or 6 h, respectively (F). Flow cytometry-based quantification of ASC speck formation in GFP-ASC[tg] HEK293T cells overexpressing the indicated NLRP3-mCherry proteins at 250 ng DNA/well (G). Mean fluorescence intensity (MFI) of NLRP3-mCherry in GFP-ASC[tg] HEK293T cells expressing indicated NLRP3-mCherry proteins in E. For untransfected cells, this is the MFI of all single cells. For cells expressing mCherry-NLRP3 this is the MFI of cells in the mCherry low gate indicated in (D). (H) Flow cytometry-based quantification of ASC speck formation in GFP-ASC[tg] HEK293T cells, overexpressing the indicated NLRP3-6-mCherry chimeric proteins at 250 ng DNA/well. (I) Mean fluorescence intensity (MFI) of NLRP3-6-mCherry chimeric proteins in GFP-ASC[tg] HEK293T cells transfected at 80 ng DNA/well, except for chimeras NLRP3[NACHT(NLRP6)] and NLRP3[LRR(NLRP6)] which were transfected at 200 and 150 ng/well, respectively. For untransfected cells, this is the MFI of all single cells. For cells expressing mCherry-NLR chimeras this is the MFI of cells in the mCherry low gate indicated in (D). (J) Flow cytometry-based quantification of ASC speck formation in GFP-ASC[tg] HEK293T cells, overexpressing the indicated NLRP6 proteins at 250 ng DNA/well. (K) Mean fluorescence intensity (MFI) of indicated NLRP6-mCherry proteins in GFP-ASC[tg] HEK293T cells transfected at 80 ng DNA/well. For untransfected cells, this is the MFI of all single cells. For cells expressing mCherry-NLRP6 this is the MFI of cells in the mCherry low gate indicated in (D). (L) Flow cytometry-based quantification of ASC speck formation in GFP-ASC[tg] HEK293T cells overexpressing the indicated NLRP6 proteins at 250 ng DNA/well (M). Immunoblot showing expression levels of indicated NLRP6 proteins in GFP-ASC[tg] HEK293T transfected at 80 ng/well. Tubulin serves as loading control. Graphs show mean ± SD from three independent experiments. Immunoblots are representative of at least two independent experiments. Statistics used: one-way ANOVA with Dunett's multiple comparisons test (B), two-way ANOVA with Šídák's multiple comparisons test (E), unpaired *t* test (F, L).

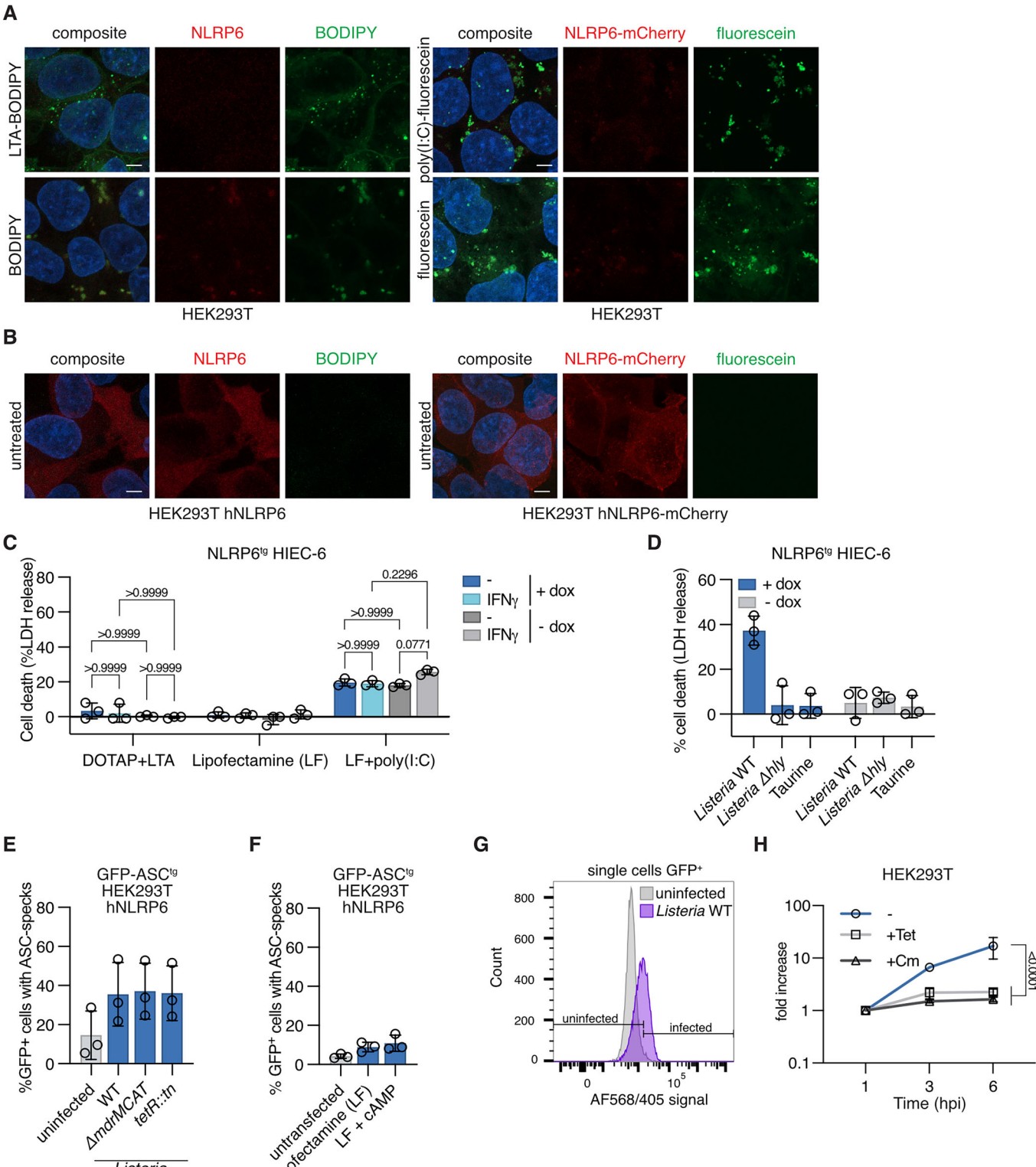

◀ **Figure EV3. NLRP6 activation does not require bacterial PAMPs, but viable cytosolic bacteria.**

(**A**) Representative maximum projection confocal micrographs after transfection of HEK293T cells with LTA-BODIPY (7.5 µg/300,000), equivalent volume BODIPY only, poly(I:C)-fluorescein (250 ng/300,000 cells) or equivalent volume fluorescein only for 6 h. (**B**) Representative maximum projection confocal micrographs of hNLRP6- or hNLRP6-mCherry-expressing HEK293T cells left untreated. (**C**) LDH release from NLRP6-WT[tg] HIEC-6 cells induced or not with 1 µg/ml doxycycline and primed or not with 10 ng/mL IFNγ overnight and transfected with Lipofectamine (LF) only or transfected with LTA or poly(I:C) for 8 h. (**D**) LDH release from NLRP6-WT[tg] HIEC-6 cells induced or not with 1 µg/ml doxycycline overnight and infected with WT or Δhly *L. monocytogenes* EGD for 8 h or treated with 70 mM Taurine. (**E**) Flow cytometry-based quantification of ASC speck formation in hNLRP6-expressing GFP-ASC[tg] HEK293T cells infected with the indicated strains of *L. monocytogenes* 10403S for 8 h. (**F**) ASC speck formation in hNLRP6-expressing GFP-ASC[tg] HEK293T cells mock treated or transfected with c-di-AMP for 24 h. (**G**) Flow cytometry gating strategy for antibody-stained *Listeria*-infected cells. (**H**) Replication of WT *L. monocytogenes* EGD over time in HEK293T cells in the presence of tetracycline (Tet) or chloramphenicol (Cm) added after 45 min of infection. Graphs show means ± SD from three independent experiments. Graphs (**C**, **D**) show mean ± SD from three pooled experiments with three technical replicates each. Each data point represents the mean of one experiment. Images are representative of three independent experiments. Scale bars represent 5 µm. Statistics used: two-way ANOVA with Šídák's multiple comparisons test.

 

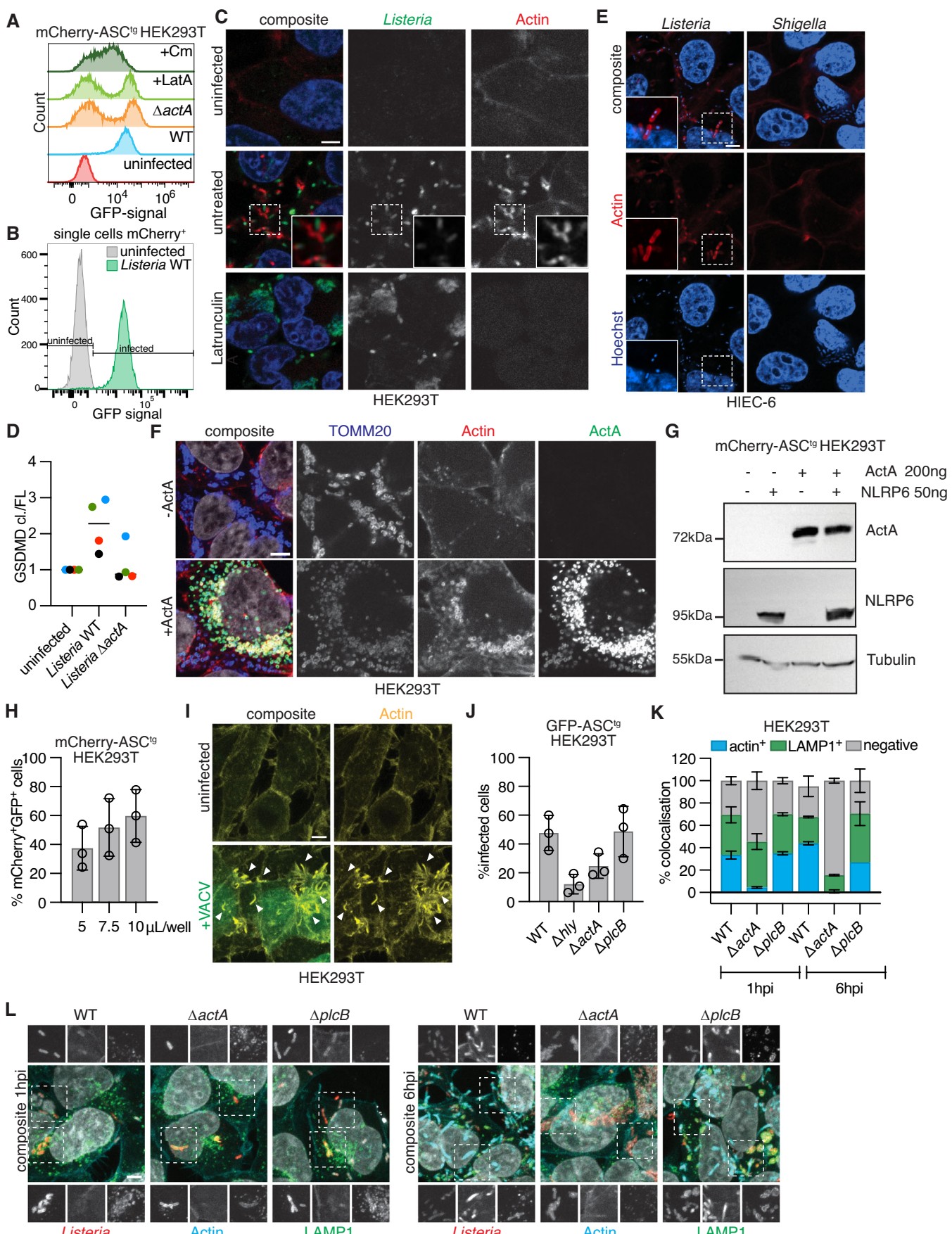

◀  **Figure EV4.  NLRP6 recognizes bacterial cell-to-cell spread.**

(**A**) Distribution of GFP signal as analyzed by FACS in hNLRP6-expressing mCherry-ASC[tg] HEK293T cells, infected with GFP-expressing WT or Δ*actA L. monocytogenes* EGD for 6 h, or infected with GFP-expressing WT *L. monocytogenes* EGD and treated with Latrunculin A (LatA) or chloramphenicol (Cm) after 45 min of infection. (**B**) Flow cytometry gating strategy for GFP-tagged *Listeria*-infected cells. (**C**) Maximum projection confocal micrographs of actin tail formation by GFP-expressing WT *L. monocytogenes* EGD, treated with Latrunculin A (LatA) and imaged at 6hpi. Actin was stained with CellMask Actin Stain and regions of insets are marked by dashed boxes. (**D**) Quantification of GSDMD cleavage from western blots of NLRP6-WT[tg] HIEC-6 induced with 1 μg/mL doxycycline overnight and then infected, or not, with *Listeria* for 8 h. Each color represents one biological replicate. Line represents the median. (**E**) Maximum projection micrographs of actin tail formation by *L. monocytogenes* EGD or *S. flexneri* expressing AfaI after 6 h of infection in HIEC-6, stained for Actin with CellMask Actin Stain. Arrowheads indicate actin-tails. (**F**) Maximum projection confocal micrograph of HEK293T cells exogenously expressing ActA or not, stained for ActA and TOMM20 by immunofluorescence and Actin was stained with CellMask Actin Stain. (**G**) Immunoblot for expression of hNLRP6 and ActA in mCherry-ASC[tg] HEK293T cells expressing hNLRP6 or hNLRP6 and ActA. (**H**) Percentage of vaccinia-infected (GFP-positive) hNLRP6-expressing mCherry-ASC[tg] HEK293T cells, infected with indicated volumes of vaccinia virus at 4.7e10 pfu/mL for 8 h. (**I**) Actin tails formed in vaccinia virus infected (GFP-positive) HEK293T cells. Actin was stained with CellMask Actin Stain and actin tails indicated by arrowheads. (**J**) Percentage of infected cells in hNLRP6-expressing mCherry-ASC[tg] HEK293T cells, infected with WT, Δ*hly*, Δ*actA* and Δ*plcB L. monocytogenes* 10403S for 6 h. Intracellular *Listeria* were stained by immunofluorescence and analyzed by flow cytometry. (**K, L**) Quantification and representative maximum projection confocal micrographs of intracellular *Listeria* colocalizing with Actin or the lysosomal marker LAMP1 in HEK293T cells infected with WT, Δ*act* and Δ*plcB L. monocytogenes* 10403S after 1 and 6hpi. Inserts are marked by dotted squares. *Listeria* and LAMP1 were stained by immunofluorescence and for Actin CellMask Actin Stain was used. (**H, J**) Show mean ± SD from three independent experiments. Graph (**K**) shows means ± SD from two independent experiments. Images and blots are representative of at least two independent experiments. Scale bars represent 5 μm.

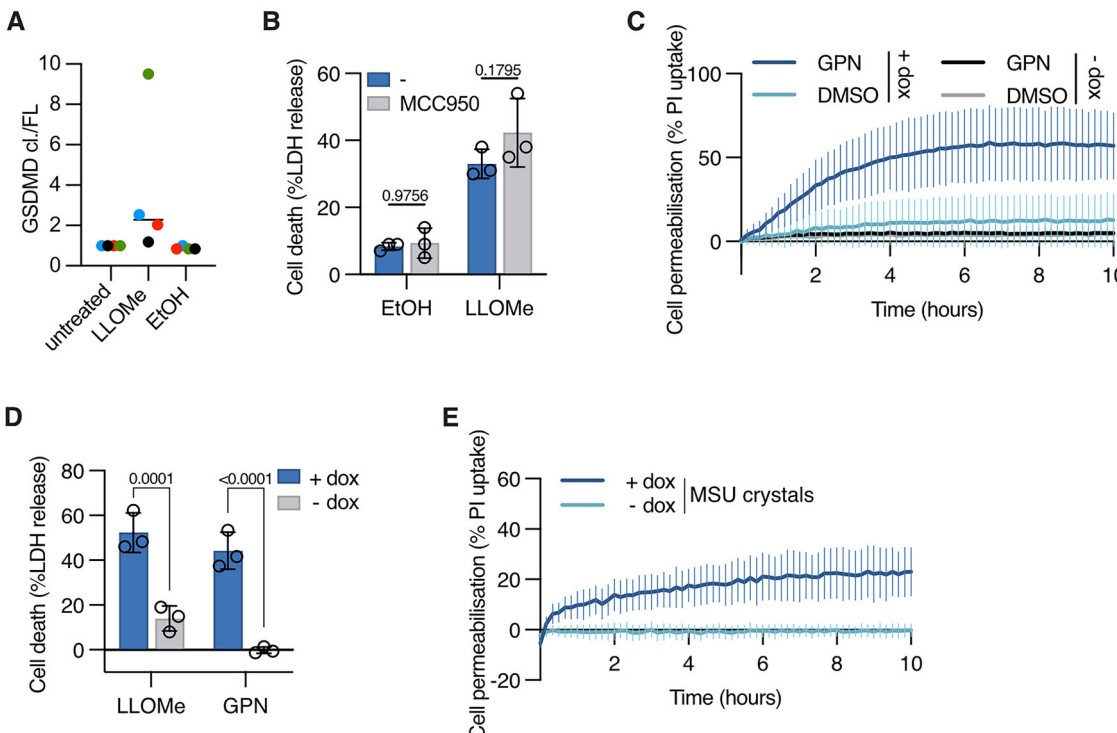

**Figure EV5.   Sterile endolysosomal damage is sufficient to activate NLRP6 in human intestinal epithelial cells.**

(**A**) Quantification of GSDMD cleavage from Western blots of NLRP6-WT[tg] HIEC-6 induced 1 μg/mL doxycycline overnight and then treated with LLOMe or EtOH for 4 h. Each color represents one biological replicate. Line represents the median. (**B**) LDH release from NLRP6-WT[tg] HIEC-6 cells induced or not with 1 μg/ml doxycycline overnight, subsequently treated or not with 10 μM MCC950 30 min before and during treatment with LLOMe for 4 h. (**C**) Time course of propidium iodide (PI) uptake of NLRP6-WT[tg] HIEC-6 cells induced or not with 1 μg/ml doxycycline overnight and treated with 200 μM GPN or equivalent volume of DMSO (**D**). Cell death measured by LDH release from NLRP6[tg] HIEC-6 cells induced or not with 1 μg/mL doxycycline overnight and treated with 0.5 mM LLOMe or 200 μM GPN for 8 h. (**E**) Propidium iodide (PI) uptake of NLRP6-WT[tg] HIEC-6 cells induced or not with 1 μg/ml doxycycline overnight and treated with 100 μM MSU crystals for 10 h. Graphs show the mean ± SD of three independent experiments with three technical replicates each, respectively (**B–E**). Each data point shows the mean of one experiment. Statistics used in graphs (**B, D**): two-way ANOVA with Šídák's multiple comparisons.

