## [Peer Review File · The EMBO Journal]

The NLRP6 inflammasome is activated by sterile or pathogen-induced endolysosomal damage

Alexandra Boegli, Elliott Bernard, Louise Lacante, Gaël Majeux, Ella Hartenian, Vanessa Mack, and Petr Broz

Corresponding author(s): Petr Broz (petr.broz@unil.ch)

Review Timeline:

Submission Date:	24th Mar 25
Editorial Decision:	2nd May 25
Revision Received:	28th Jul 25
Editorial Decision:	6th Sep 25
Revision Received:	17th Sep 25
Accepted:	28th Sep 25

Editor: Ioannis Papaioannou

Transaction Report:

Dear Petr,

Thank you again for submitting your manuscript EMBOJ-2025-120875 for consideration by The EMBO Journal, and for your patience during peer review. It has now been seen by three experts in the field, and we have received the full set of their detailed, constructive, and well-informed reports, which you can find below.

I am pleased to say that, as you will see, the referees all indicate interest in the study and the findings, recognize the novelty and significance of the work, and conclude that it will make a significant contribution to the field. However, they also identify a number of limitations and list major and minor concerns that should be addressed. There is consensus among the referees that the identify of the cells that were used in the study limits to some extent the overall impact of the work, and this is a relevant point that we would like to discuss. The referees acknowledge the objective limitations when it comes to cell lines endogenously expressing NLRP6, but they all point out this limitation and provide specific suggestions for validating main findings of the study in physiologically more relevant systems (e.g. naturally NLRP6-expressing cell lines, primary cells, mouse intestinal epithelial cells). In line with the referees' advice, we recognize and agree that this is a more molecularly oriented study, and therefore we will not request full in vivo confirmation, but we think that validation of the results in more relevant cells according to the reviewers' suggestion is necessary for increasing the impact of the work on the field. Furthermore, the referees identify several other limitations including technical issues (regarding quantification, controls, and statistics among others) that must be addressed in a revised version of the manuscript.

Given the referees' supportive comments and positive recommendations, I would like to invite you to submit a revised version of your manuscript taking the referees' suggestions on board, along with a detailed point-by-point response addressing all referees' comments. I should add that it is The EMBO Journal policy to allow only a single round of major revision, and acceptance of your manuscript will therefore depend on the completeness of your responses in this revised version. Please let me know if you have any questions or comments that you would like to discuss with me. If there are any major points you do not agree with or cannot address during your revision, I would encourage you to share them with me as early as possible to discuss how to proceed further in the most efficient way.

We generally allow three months as standard revision time (August 1, 2025). As a matter of policy, competing manuscripts published during this period will not negatively impact our assessment of the conceptual advance presented by your study. However, we request that you contact us as soon as possible upon publication of any related work, to discuss how to proceed. Should you foresee a problem in meeting this three-month deadline, please let us know in advance and we will be able to grant an extension.

Thank you for the opportunity to consider your work for publication in The EMBO Journal. I look forward to your revision.

Best wishes,

Ioannis

Instructions for preparing your revised manuscript

1. When you are ready to submit the revision, please upload:

- A Word file of the manuscript text (including legends of main Figures, EV Figures and Tables). Please make sure that changes are highlighted (or "tracked") to be clearly visible.

- Individual production-quality figure files (one file per figure). When assembling your figures, please refer to our figure preparation guidelines in order to ensure proper formatting and readability in print as well as on screen:

If the data shown in a figure are obtained from n {less than or equal to} 2, please use scatter plots showing the individual data points.

- i. the name of the statistical test used to generate error bars and P values
- ii. the number (n) of independent experiments (please specify technical or biological replicates) underlying each data point (discussion of statistical methodology can be reported in the Materials and Methods section, but figure legends should contain a basic description of n, P, and the test applied)
- iii. the nature of the bars and error bars (s.d., s.e.m.).

- A point-by-point response to the referees' comments, with a detailed description of the changes made (as a word file). All referees' concerns must be fully addressed and their suggestions taken on board. When preparing your letter of response to the referees' comments, please bear in mind that this will form part of the Review Process File and will therefore be available online to the community. Please note that you have the possibility to opt out of the transparent process at any stage prior to publication by letting the editorial office know (contact@embojournal.org); if you do opt out, the Review Process File link will point to the following statement: "No Review Process File is available with this article, as the authors have chosen not to make the review process public in this case.". For more details on our Transparent Editorial Process, please visit our website: <https://www.embopress.org/page/journal/14602075/authorguide#transparentprocess>

- Expanded View (EV) files (replacing Supplementary Information) that are collapsible/expandable online. A maximum of 5 EV Figures can be typeset. EV Figures should be cited as "Figure EV1, Figure EV2" etc. in the text, and their respective legends should be included in the manuscript file after the legends of regular figures. See detailed instructions regarding Expanded View files here:

- For the figures that you do NOT wish to display as Expanded View figures, they should be bundled together with their legends in a single PDF file called "Appendix", which should start with a short Table of Contents (including page numbers). Appendix figures should be referred to in the main text as: "Appendix Figure S1, Appendix Figure S2" etc. Please see detailed instructions here: <https://www.embopress.org/page/journal/14602075/authorguide#expandedview>

- A complete author checklist, which you can download from our author guidelines (<https://www.embopress.org/page/journal/14602075/authorguide>). Please note that the checklist will also be part of the Review Process File.

2. Please note that no statistics should be calculated and shown in Figures if n=2. Please also note that each p value should be reported as an exact value.

3. Before submitting your revision, primary datasets (and computer code, where appropriate) produced in this study need to be deposited in appropriate public databases (see <https://www.embopress.org/page/journal/14602075/authorguide#dataavailability>). The accession numbers, database, and the specific URLs (links) should be listed in a formal "Data availability" section (placed after Methods), following the example below:

"The RNA-seq datasets produced in this study are available in the following database:
Gene Expression Omnibus GSE46843 (<https://www.ncbi.nlm.nih.gov/geo/query/acc.cgi?acc=GSE46843>)"

*** All links should resolve to a page where the data can be accessed. ***

*** Please remember to provide in the Data availability section of your revised manuscript reviewer passwords if the datasets are not yet public. ***

*** The Data Availability Section is restricted to new primary data that are part of this study. In case you have no data that require deposition in a public database, please state so instead of referring to the database: "Our study includes no data deposited in public repositories." under the heading "Data availability". ***

4. The materials and methods need to be described in the manuscript using our structured methods format, which is now required for all research articles. According to this format, the Methods section includes a single "Reagents and Tools Table" - listing key reagents, experimental models, software and relevant equipment including their sources and relevant identifiers- followed by a "Methods and Protocols" section describing the methods. Please download and fill our Reagents and Tools Table template (.docx), which you can find in our author guide:

<https://www.embopress.org/page/journal/14602075/authorguide#structuredmethods>. When submitting your revised manuscript, please do not include the Reagents and Tools Table in the Methods section of the manuscript but instead upload it as a separate file choosing the file type "Reagent Table".

5. Please check that the title and the abstract of the manuscript are brief, yet explicit, even to non-specialists. The length of the title should not exceed 100 characters, and the abstract should be a single paragraph not exceeding 175 words.

6. Please also note our reference format: <https://www.embopress.org/page/journal/14602075/authorguide#referencesformat>.

8. Please remember: digital image enhancement is acceptable practice, as long as it accurately represents the original data and conforms to community standards. If a figure has been subjected to significant electronic manipulation, this must be noted in the figure legend or in the "Materials and Methods" section. The editors reserve the right to request original versions of figures and the original images that were used to assemble the figure.

9. Our journal encourages inclusion of data citations in the reference list to directly cite datasets that were obtained from public databases. Data citations in the article text are distinct from normal bibliographical citations and should directly link to the database records from which the data can be accessed. In the main text, data citations are formatted as follows: "Data ref: Smith et al, 2001" or "Data ref: NCBI Sequence Read Archive PRJNA342805, 2017". In the Reference list, data citations must be labeled with "[DATASET]". A data reference must provide the database name, accession number/identifiers, and a resolvable link to the landing page from which the data can be accessed at the end of the reference. Further instructions are available at: <https://www.embopress.org/page/journal/14602075/authorguide#referencesformat>.

10. We request authors to consider both actual and perceived competing interests. Please review our policy (<https://www.embopress.org/page/journal/14602075/authorguide#conflictsofinterest>) and update your competing interests statement if necessary. Please name this section 'Disclosure and competing interests statement' and place it after the Acknowledgements section.

11. Please note that all corresponding authors are required to provide an ORCID ID upon submission of a revised manuscript (<https://orcid.org/>). Please find instructions on how to link your ORCID ID to your account in our manuscript tracking system in our Author guidelines (<https://www.embopress.org/page/journal/14602075/authorguide#authorshipguidelines>).

12. We use CRediT to specify the contributions of each author in the journal submission system. CRediT replaces the author contribution section, which should be removed from the manuscript. Please use the free text box to provide more detailed descriptions. See also guide to authors: <https://www.embopress.org/page/journal/14602075/authorguide#authorshipguidelines>.

14. We would also welcome the submission of cover suggestions or motifs to be used by our Graphics Illustrator in designing a cover.

15. Please use the link below to submit your revision:
<https://emboj.msubmit.net/cgi-bin/main.plex>

Referee #1:

General summary and opinion about the principle significance of the study, its questions and findings

+ In their present work, Boegli et al investigate the pattern or process driving the activation of the human danger sensor, NLRP6, one of a group of cytosolic so-called pattern recognition sensors which is able to form a so-called inflammasome, a molecular machinery driving inflammation and pyroptotic cell death. Whilst metabolites, dsRNA and lipoteichoic acid have been proposed to act as direct ligands in immune cells, there has been some uncertainty about this and the role of NLRP6 in the gut, where so far its function has been studied (albeit non-mechanistically, i.e. in terms of activating mechanism) most and where its expression is prominent. For example, NLRP6 has been ascribed important roles in gut homeostasis, host-microbe interactions and colorectal cancer but the underlying mechanistic framework has been missing. Boegli et al provide important new insights in that they propose a mechanism of lysosomal damage as a trigger for NLRP6 in the context of infections with the gut-pathogenic bacteria, *Listeria monocytogenes*. In a largely well controlled and meticulous manner they study NLRP6 function in genetically modified intestinal epithelial cell line (HIEC-6) using purified known triggers (which, on their own, do not trigger NLRP6 activity) and also genetically modified bacteria. By this strategy, they can identify specific domains in NLRP6 (NACHT, LRR) to be critical for mediate *Listeria*-mediated activation. Moreover, they find that bacterial escape from secondary vacuoles is essential for NLRP6 activity. Interestingly, NLRP6 was also activated by lysosome-destabilizing sterile treatments, identifying endolysosomal damage as an NLRP6 activating process in this cellular system. One major caveat is that the cellular systems used by the

authors all naturally lack NLRP6 and only become responsive after introduction of NLRP6. Their work suggests that NLRP6 is a sensor of a specific, infection-relevant process of homeostasis perturbation, rather than (or at least in addition) to the specific MAMPs previously suggested, when NLRP6 is introduced. Whilst it remains limited to cell lines, the paper proposes a novel and potentially very relevant new mechanism of sensing infection but also sterile perturbations with relevance to gut homeostasis. + The work is comprehensive and well designed and was already received with great enthusiasm when presented at conferences and, since appearing on Biorxiv in January 2025, has already been downloaded in PDF format ~400 times. As NLRP6 is a critical PRR whose mechanism of activation is far less understood than other PRRs, this suggesting an expected high impact in the field.

Specific major concerns essential to be addressed to support the conclusions

A) Physiological relevance of the systems:

- 1) Whilst the switch from HEK293T cells to HIEC-6 is plausible and increases relevance, HIEC-6 are still an immortalized cell line and only becomes responsive upon NLRP6 expression. Why this cell line was chosen is not entirely clear. Ideally the authors would provide data in a naturally NLRP6-expressing cell system. According to human protein atlas (https://www.proteinatlas.org/ENSG00000174885-NLRP6/cell+line#colorectal_cancer) several CRC lines are NLRP6 expressing and a comparison of 2 (e.g. low vs high) expressors with non-induced HIEC-6 would be critical. The authors should also then try to find a way to demonstrate NLRP6 dependence, e.g. by CRISPR KO of NLRP6 or siRNA.
- 2) Additionally the authors would find some way to move to some sort of primary cell system. Since they have Nlrp6 KO mice at hand, would explants from WT and Nlrp6 KO be possible? We have done cultures of intestinal pieces in well plates for several hours.
- b) This also begs the question of whether the system they propose operates in mouse, which would be very useful for the work to trigger in vivo work by others - an in vivo conformation I would consider outside the scope of this molecularly oriented study. But at least some work in a mouse intestinal epithelial cell line for some of the key experiments would be very important to show.

B) Demonstration of bacterial spread to non-infected cells

The hypothesis is very intriguing and the data basis good, but I find the schematic of Fig. 6G showing that NLRP6 gets activated in the neighbouring, not initially infected cell, not entirely validated by the presented experiments as only bulk infected cultures are used. I would recommend that cells without the NLRP6 inducible system and lacking ASC are infected with fluorescent bacteria or their mutants and then sorted and re-seeded together with uninfected cells with the inducible system to check for activation. Alternatively, the infection could be carried out as suggested in a more sparsely seeded non-inducible culture, extracellular bacteria killed by adding antibiotics immediately after internalization (1 h?), washed and then inducible-cells are seeded on top. LLOMe and escape-inefficient mutants of *Listeria* would be good controls. Maybe the authors could think of other/more practical ways of showing what I am getting at.

C) Important technical issues

-
- 1) ideally blots should be quantified and the different replicates be plotted, especially for GSDMD cleavage as this is a vital readout for this important molecular cell death factor
 - 2) Immunoblots for the NLRP6/NLRP3 chimeras should also be shown in supplement as the mention of equal expression by FACS is not documented in the manuscript; Moreover, for the critical ones responsiveness to LLOMe should be tested for Fig. 3D which also lacks statistics.
 - 3) Have the authors considered that dsRNA, LTA (and/or Taurin) alone are insufficient and Actin destabilization is also required? And have they considered IFN priming may also be needed in the HIEC-6? I would recommend and would be interested to see at least as a reviewer figure, a co-treatment of these proposed agonists and actin destabilized, with/without IFN pre-treatment like in some of the other agonist papers. Same for a co-treatment of dsRNA or LTA with heat-killed lysate;
 - 4) In Figure 4 the showing of actin tails requires larger magnifications.
 - 5) At least a representative FACS result and a precise description of the gating strategy used for the ASC aggregation assay should be shown in Fig. S1 to understand the most critical assay used throughout better. For fig. 5, the FACS plots and gating also should be shown to illustrate for the disparate infection rates mentioned (but not specified).
 - 6) Statistics: ANOVA assumes a normal distribution of data but how normality was tested for is not described in methods. This should be amended and the choice of test modified if necessary.

D) Minor concerns that should be addressed

-
- + The expression of pattern of NLRP6 in line 48 should be more finely delineated between mouse vs human.
 - + Bafilomycin A1 can also interfere with DNA and RNA sensing, albeit in the endosomes. Can the authors be sure that it does not affect sensing of bacterial RNA, which the authors concede to be a labile and potentially hard to track trigger?
 - + That HIEC-6 are a non-immortalized cell line should be specified in line 118/119 as the reader currently might wrongly assume these are either primary human intestinal endothelial cells or a colorectal cancer cell line
 - + The blotting for NLRP3 in HIEC-6 in combination with THP-1 might mask low NLRP3 expression in these cells next to an expectedly very high NLRP3 signal from THP-1. An additional blot of HIEC-6 next to HEK293T cells (certainly negative) would

be more informative and useful to rule out basal expression of NLRP3 in their system. mRNA quantification for NLRP6 and NLRP3 would also be useful.

- + The targeting strategy of KO generation for Caspase-1 and -4 should be described. What resulted in the KO, e.g. a frameshift leading to a truncated protein, should be mentioned.
- + For completeness sake the authors might want to test the effect of Taurin on NLRP6 in their system as Taurin is mentioned but not assessed.
- + Controls that the NLRP6 the NLRP6-non-activators LTA and dsRNA are intact and generally active (e.g. for other pathways) are missing, e.g. for dsRNA to show IFN induction from the same cells.
- + The authors need to mention in line 344 that LLoMe is also a well-established NLRP3 agonist. An inclusion of LLoMe in the chimera analysis would hence have been helpful too. MCC950 might be a good control in some of the experiments of Fig. 5/6 to exclude NLRP3 completely.
- + In Fig. 4B white boxes with small digits 1 or 2, rather than arrowheads (which might be mistaken to point to "specks" of some sort) might be more useful to show the 2 inserts?
- + Why in Fig. 3F was an uncorrected Fisher's LSD used?
- + The discussion between line 414 and 434 (approx) is quite dense and could maybe be simplified.

Any additional non-essential suggestions for improving the study (which will be at the author's/editor's discretion)

- + The microscopy analysis in fig. 2A shows that ASC specks seem to occur in direct proximity to the nucleus. What is the significance?

Referee #2:

While several previous publications have claimed to report mechanisms activating the NLRP6 inflammasome, these have been controversial in the field. Here, the Broz lab provide compelling evidence in support of a model in which NLRP6 is a sensor for sterile endolysosomal damage, or bacterial-induced membrane damage. In an elegant series of cell infection assays (*Listeria*, *Shigella*) of NLRP6-expressing epithelia, the authors untangle functions for NLRP6 in sensing membrane damage during infection and with lysomotropic agents. In general, the study is compelling and worthy of publication in *EMBO J*, and I congratulate the authors on the study. There are however, several aspects that require addressing to meet the required level of scientific rigour for publication.

Major issues:

1. Throughout the figures (Figs 1-5, S1-6), there is a perplexing lack of uninfected control samples shown. Because ectopic NLRP6 expression itself triggers inflammasome signalling, it is often difficult to determine whether it is infection or NLRP6 that is driving the response. This makes it very difficult to agree with some of the author's statements (e.g. it is often stated that infection triggers an NLRP6-dependent response, but without an infected control, a reader can only conclude it is an NLRP6-dependent response). Upon close inspection of the methods, it seems the uninfected control was indeed performed and used to somehow "normalise" or background subtract the data. In this Reviewer's opinion, the data should not be transformed in this way, and the graphs should show the uninfected controls, so that the reader properly assess the full dataset.
2. In Figure 3 and S3, flow cytometry is performed to analyse Cherry-tagged NLRP3 versus NLRP6 function in ASC-GFP cells. The authors show Cherry expression is somewhat similar between constructs. Many studies, including this study, show the strong impact of NLR expression level on inflammasome signalling outputs, so small differences in expression can generate spurious results. The advantage of having cherry-tagged constructs is that you can precisely control for cherry expression by gating on cherry when analysing the cytometry data. It is preferable to analyse these data using a narrow cherry gate (rather than all cherry+ cells), as this will allow you to compare ASC specks in cell populations that express equivalent levels of NLR-cherry proteins.
3. Some graphs and statistics appear to represent pseudo-replicated data. In several figures (Figs:2D, 2G, 4D, 5C-D, 6C-F ; Supp 6A, 6F, 6I, 6J) technical replicates are shown alongside biological replicates, and statistics appear to have utilised both sets of data. Only biological replicates should be shown in graphs, and analysed by stats.
4. The authors propose (e.g. Fig. 6G, discussion) that NLRP6 is activated in epithelial cells to assemble an inflammasome that cleaves GSDMD and thereby induces pyroptotic death. This is a reasonable presumption, but the GSDMD dependency of cell death was not formally tested. Suggest showing this with GSDMD knockout HIEC.
5. A major substrate of inflammasomes are immature cytokines, such as pro-IL-1, pro-IL-18 and pro-IL-37. Do NLRP6-activated HIEC cleave and release these cytokines, or is cell death the only outcome of NLRP6 inflammasome signalling?
6. Is the elusive signal detected by NLRP6 actually galectin redistribution upon membrane damage?

Study limitations:

1. While unavoidable until cell lines endogenously expressing NLRP6 are identified, it is a weakness of the study that NLRP6 was ectopically, rather than endogenous expressed throughout the study.
2. Related to the above point, it is a pity that ectopic NLRP6 expression in HIEC was strong enough to bypass NLRP6 autoinhibition in the absence of a stimulus (i.e. NLRP6 expression without stimulus triggered some cells to signal via this inflammasome). Because of this, it is important to show the untransformed data throughout the manuscript (see major point 1).

In this Reviewer's opinion, these limitations should not preclude the publication of this interesting work, as so little is known about NLRP6 functions and the study represents a major step forward.

Minor suggestions for improving clarity:

1. Line 122: suggest mentioning in the text that NLRP6 is expressed as an NLRP6-IRES-GFP cassette, so that it is not a surprise for the reader to see GFP detected on western.
2. Figure 1B should be analysed statistically
3. Fig 1E lacks a bafA-untreated uninfected control
4. Line 150: "the results confirm that LLO-dependent escape of *Listeria* from the endosomal compartment is an essential prerequisite for NLRP6 activation in human epithelial cells". While this is a reasonable interpretation for LLO action, the authors have not directly shown that endosomal escape is necessary. Please modify to ensure it is clear this is speculation. Also true of "cytosolic entry" in line 163.
5. A large portion (~25%) of the discussion is dedicated to discussing possible involvement of autophagy, while the data did not support such an involvement. This makes the discussion feel unfocused. Suggest shortening this section substantially.

Referee #3:

Boegli et al. present compelling evidence that NLRP6 is activated through endolysosomal damage during *Listeria monocytogenes* infection in human intestinal epithelial cells (IECs). Their findings challenge earlier reports suggesting that NLRP6 directly senses bacterial pathogen-associated molecular patterns (PAMPs), such as lipoteichoic acid and dsRNA, via its leucine-rich repeat (LRR) domain to initiate inflammasome activation. Instead, the study demonstrates that NLRP6 activation is dependent on *Listeria*-induced endolysosomal disruption, specifically requiring Listeriolysin O-mediated cytosolic invasion and escape from secondary vacuoles. This mechanism highlights NLRP6's role as a sensor of cellular homeostasis rather than direct pathogen recognition, which contrasts with previous studies reporting PAMP-mediated activation. The study's robust datasets are well-controlled, and the authors carefully contextualize their findings in light of prior research. The manuscript is clear and requires minimal editing.

However, it should be noted that all experiments were conducted in cell lines reconstituting NLRP6 expression. Since NLRP6 is endogenously expressed primarily in intestinal epithelial cells, further validation using NLRP6 knockout (KO) mice and cultured primary intestinal cells would strengthen the conclusions. For example:

- Testing in primary cells: Isolating intestinal epithelial cells from KO and wild-type mice could help assess responses to identified stimuli like LLOMe and *Listeria* strains compared to lipoteichoic acid and dsRNA.
- Alternative approaches, if mice are not available: Using viral delivery of shRNA targeting NLRP6 in primary intestinal epithelial cells could provide insights into inflammasome activation mechanisms in a more physiologically relevant setting.

It would also be interesting to investigate whether LLOMe-induced NLRP6 activation also depends on its NACHT domain.

Overall, this study significantly advances our understanding of NLRP6 inflammasome biology and challenges earlier paradigms of direct PAMP sensing. While experiments in primary cells would be ideal, the data presented are sufficiently compelling to warrant publication without these additional validations.

Referee #1:

General summary and opinion about the principle significance of the study, its questions and findings

+ In their present work, Boegli et al investigate the pattern or process driving the activation of the human danger sensor, NLRP6, one of a group of cytosolic so-called pattern recognition sensors which is able to form a so-called inflammasome, a molecular machinery driving inflammation and pyroptotic cell death. Whilst metabolites, dsRNA and lipoteichoic acid have been proposed to act as direct ligands in immune cells, there has been some uncertainty about this and the role of NLRP6 in the gut, where so far its function has been studied (albeit non-mechanistically, i.e. in terms of activating mechanism) most and where its expression is prominent. For example, NLRP6 has been ascribed important roles in gut homeostasis, host-microbe interactions and colorectal cancer but the underlying mechanistic framework has been missing. Boegli et al provide important new insights in that they propose a mechanism of lysosomal damage as a trigger for NLRP6 in the context of infections with the gut-pathogenic bacteria, *Listeria monocytogenes*. In a largely well controlled and meticulous manner they study NLRP6 function in genetically modified intestinal epithelial cell line (HIEC-6) using purified known triggers (which, on their own, do not trigger NLRP6 activity) and also genetically modified bacteria. By this strategy, they can identify specific domains in NLRP6 (NACHT, LRR) to be critical for mediate *Listeria*-mediated activation. Moreover, they find that bacterial escape from secondary vacuoles is essential for NLRP6 activity. Interestingly, NLRP6 was also activated by lysosome-destabilizing sterile treatments, identifying endolysosomal damage as an NLRP6 activating process in this cellular system. One major caveat is that the cellular systems used by the authors all naturally lack NLRP6 and only become responsive after introduction of NLRP6. Their work suggests that NLRP6 is a sensor of a specific, infection-relevant process of homeostasis perturbation, rather than (or at least in addition) to the specific MAMPs previously suggested, when NLRP6 is introduced. Whilst it remains limited to cell lines, the paper proposes a novel and potentially very relevant new mechanism of sensing infection but also sterile perturbations with relevance to gut homeostasis.

+ The work is comprehensive and well designed and was already received with great enthusiasm when presented at conferences and, since appearing on Biorxiv in January 2025, has already been downloaded in PDF format ~400 times. As NLRP6 is a critical PRR whose mechanism of activation is far less understood than other PRRs, this suggesting an expected high impact in the field.

We thank the referee for the very positive feedback on our study. Below, we address the points raised, incorporating additional experiments where possible.

Specific major concerns essential to be addressed to support the conclusions.

A) Physiological relevance of the systems:

1) Whilst the switch from HEK293T cells to HIEC-6 is plausible and increases relevance, HIEC-6 are still an immortalized cell line and only becomes responsive upon NLRP6 expression. Why this cell line was chosen is not entirely clear.

We agree that, using an intestinal cell line with endogenous NLRP6 expression would have been ideal. As outlined below, we evaluated several commonly used cell lines, including Caco-2, HT-29, and HCT116, and found that none express NLRP6, with many also lacking other key components of the inflammasome pathway.

We specifically selected HIEC-6 cells because they are non-cancerous intestinal epithelial cells that endogenously express core inflammasome components such as ASC, caspase-1, and GSDMD, but lack inflammasome-forming receptors. This made them a suitable model in which exogenous NLRP6 expression was sufficient to reconstitute the pathway.

Ideally the authors would provide data in a naturally NLRP6-expressing cell system.

According to human protein atlas (https://www.proteinatlas.org/ENSG00000174885-NLRP6/cell+line#colorectal_cancer) several CRC lines are NLRP6 expressing and a comparison of 2 (e.g. low vs high) expressors with non-induced HIEC-6 would be critical. The authors should also then try to find a way to demonstrate NLRP6 dependence, e.g. by CRISPR KO of NLRP6 or siRNA.

As suggested, we tested SW1463 cells, which had the highest TPM on the Human Protein Atlas, as well as Mino cells that have the third highest levels of NLRP6 mRNA. We also ordered the second highest expressors, DV-90, but the only supplier of these cells did not manage to send the cells before the re-submission deadline of EMBO J.

We were unable to detect NLRP6 mRNA by qPCR in the SW1463 although some mRNA was detected in the Mino cells. Unfortunately, we were unable to detect any NLRP6 protein in either cell line by immunoblotting for NLRP6, despite testing various priming conditions that, according to the literature, should induce NLRP6 expression.

These findings indicate that neither cell line expresses NLRP6 at detectable levels, thereby precluding their use as models for studying endogenous NLRP6 function. Consequently, we were unable to generate NLRP6 knockout or knockdown lines to assess NLRP6 dependency. A summary of these results is provided below.

qPCR analysis of NLRP6 mRNA levels in HIEC-6, NLRP6^{tg} HIEC-6 (NLRP6 expression induced with 1 ug/mL doxycycline overnight) Mino and SW1463 cells, normalized to 18S transcripts.

Immunoblot analysis of NLRP6 protein levels in HIEC-6, NLRP6^{tg} HIEC-6 (NLRP6 expression induced with 1 μg/mL doxycycline overnight) and primed Mino cells with 300ng/mL Pam3CSK, 100ng/mL LPS, 10ng/mL IFN γ , 100 μM Rosiglitazone or 1 μg/mL poly(I:C). The immunoblot shows two independent biological replicates, using short and long exposure for the NLRP6 signal. Tubulin was used as a loading control.

Immunoblot analysis of NLRP6 protein levels in SW1463, HIEC-6, HIEC-6 NLRP6^{tg} (NLRP6 expression induced with 1μg/mL doxycycline o/n) and primed Mino cells with 300ng/mL Pam3CSK, 100ng/mL LPS, 10ng/mL IFN γ , 100μM Rosiglitazone or 1μg/mL poly(I:C) (o/n). The immunoblot shows a third independent biological replicate for the Mino cells and short and long exposure for the NLRP6 signal. Tubulin was used as a loading control.

Immunoblot analysis of NLRP6 protein levels in SW1463 and HIEC-6 NLRP6^{tg} (NLRP6 expression induced with 1μg/mL doxycycline o/n), including an independent experiment of primed SW1463 cells with 300ng/mL Pam3CSK, 100ng/mL LPS, 10ng/mL IFN γ , 100μM Rosiglitazone or 1μg/mL poly(I:C) (o/n). The immunoblot shows two independent biological replicates for the SW1463 cells and short and long exposure for the NLRP6 signal. Tubulin was used as a loading control.

2) Additionally the authors would find some way to move to some sort of primary cell system. Since they have Nlrp6 KO mice at hand, would explants from WT and Nlrp6 KO be possible? We have done cultures of intestinal pieces in well plates for several hours.

We appreciate the suggestion of using intestinal explants to test our hypothesis in a primary cell system. We have attempted to use such intestinal explants, but we found these experiments technically challenging, and the results inconsistent, with highly variable IL-18 levels being produced independent of *Listeria* infection or treatment. We are therefore unable to pursue this further in the limited timeframe for submission of a revised manuscript.

b) This also begs the question of whether the system they propose operates in mouse, which would be very useful for the work to trigger in vivo work by others - an in vivo confirmation I would consider outside the scope of this molecularly oriented study. But at least some work in a mouse intestinal epithelial cell line for some of the key experiments would be very important to show.

We tested two different mouse intestinal epithelial cell lines (YAMC, mCcl2) and were unable to detect endogenous NLRP6 expression. Western blotting for NLRP6 showed a band at approximately 100 kDa in both cell lines, however this was smaller than NLRP6 from HEK cells transfected to express the protein. qPCR analysis of the YAMC line revealed no detectable mRNA expression, we thus concluded that this band is a result of cross-reaction of the antibody with another protein. Moreover, the mCcl2 did not express GSDMD and the YAMC only expressed a truncated form of ASC that is unable to form an inflammasome (PMID: 20492797) and thus neither were suitable for reconstitution studies as we performed in the HIEC-6 model. We have thus been unable to test whether this mechanism of NLRP6 activation is conserved between mouse and human systems apart from Fig. 1D, where we test mNLRP6 in the HEK293T cell reconstitution assay.

A) Immunoblot analysis of NLRP6 and GSDMD expression in two murine colonic and intestinal epithelial cell lines (mCcl2 and YAMC, PMIDs: 19109407 & 7678459). The cells were primed overnight with 100 ng/ml IFN-β, 10 ng/ml IFN-γ, 300 nM Pam3Csk4, 100 ng/ml LPS, 100 ng/ml poly(I:C) or 100 μM Rosiglitazone or left untreated (UT) as indicated. HEK cells transfected to express mNLRP6 were used as a positive control. **B)** Immunoblot analysis of YAMC cells for the indicated inflammasome components following the indicated priming treatments overnight. **C)** qPCR analysis of *Nlrp6* from primary murine intestine samples of WT and *Nlrp6*^{-/-} animals as positive and negative controls respectively, compared with RNA extracted from unprimed YAMC cells.

B) Demonstration of bacterial spread to non-infected cells

The hypothesis is very intriguing and the data basis good, but I find the schematic of Fig. 6G showing that NLRP6 gets activated in the neighbouring, not initially infected cell, not entirely validated by the presented experiments as only bulk infected cultures are used. I would recommend that cells without the NLRP6 inducible system and lacking ASC are infected with fluorescent bacteria or their mutants and then sorted and re-seeded together with uninfected cells with the inducible system to check for activation. Alternatively, the infection could be carried out as suggested in a more sparsely seeded non-inducible culture, extracellular bacteria killed by adding antibiotics immediately after internalization (1 h?), washed and then inducible-cells are seeded on top. LLOMe and escape-inefficient mutants of *Listeria* would be good controls. Maybe the authors could think of other/more practical ways of showing what I am getting at.

The co-culture experiment of infected cells with uninfected NLRP6-competent cells is an interesting suggestion and we tried to perform it in both the HIEC-6 and HEK293T systems that we use to study NLRP6 inflammasome signaling.

Unfortunately, seeding infected HIEC-6 cells on top of uninfected, induced NLRP6^{tg} HIEC-6 cells did not work. The infected cells did not reattach and died upon re-seeding. We tried 8h and overnight reinfections after 1.5h of pre-infection. We did not see any spread of GFP bacteria from the uninduced to the induced cells, indicating that detaching and re-seeding infected HIEC-6 cells is challenging, and thus we did not further pursue this experiment in HIEC-6 cells.

We thus used the same approach in the ASC/NLRP6-expressing HEK cell model, which has the advantage that it allows us to look at single cells by FACS, discern between infected and uninfected, as well as ASC positive and negative cells. After trying different infection times and ratios, we settled on a ratio of 1:1.5 between *Listeria*-infected HEK293T cells and uninfected mCherry-ASC/NLRP6-expressing HEK293T cells co-cultured overnight to allow the bacteria to infect the NLRP6-competent cells.

The experiment showed the desired outcome, where secondary infections of NLRP6 expressing cells lead to ASC speck formation, and the bacteria deficient in actin-based motility could not induce NLRP6 activation. NLRP6 activation was not as high as previously observed for WT infected cells, however, due to differences in the experimental setup (transfection efficiency, infection time, multiplicity of infection) it is challenging to compare this quantification with that from other assays. Due to the limitations of the HEK293T reconstitution system, we were not able to use LLOMe as a control (reasoning see below). The data is presented in Fig. 5I.

While this shows the importance of secondary infections for NLRP6 activation, we also agree with the reviewer that the primary infection can also result in some level of NLRP6 activation, which we discuss in our manuscript. The reasoning is that $\Delta actA$ and $\Delta plcB$ mutants, which do not cause secondary vacuolar disruption, cause more NLRP6 activation than the Δhly mutant, that causes no vacuolar damage in general. Therefore, we adapted the schematic in Fig. 6H to include inflammasome activation in the primary infected cells with a smaller complex indicating a minor contribution as deduced from the data displayed in Fig. 5H.

C) Important technical issues

1) ideally blots should be quantified and the different replicates be plotted, especially for GSDMD cleavage as this is a vital readout for this important molecular cell death factor

We have quantified all GSDMD cleavage blots. The results can be found in Figures S1, EV4 and EV5, and show elevated GSDMD cleavage in cells infected with WT *L. monocytogenes* or treated with LLOMe compared to controls. In addition, we have generated GSDMD-deficient HIEC-6 cells and shown the dependency on GSDMD for pyroptosis (Figure 2H-I and 6E).

2) Immunoblots for the NLRP6/NLRP3 chimeras should also be shown in supplement as the mention of equal expression by FACS is not documented in the manuscript.

Expression is documented in the form of MFI (mean fluorescence intensity) in Figure EV2I (former Fig. S3H). In the revised manuscript (Figure EV2I) we report the MFI of the cells analyzed for ASC specks in the narrow mCherry gate requested by Reviewer #2, while in former Fig. S3H, MFI was reported for the entire mCherry+ population. Moreover, the implementation of this narrow mCherry+ gate as suggested by Reviewer #2 means that we have only analyzed cells with similar levels of receptor expression, as can be seen in the MFIs presented in figure EV2I.

Moreover, for the critical ones responsiveness to LLOMe should be tested for Fig. 3D which also lacks statistics.

Unfortunately, we cannot explore LLOMe-induced NLRP6 activation in ASC^{sg} HEK293T cells, for two reasons. First, HEK293T are poorly responsive to LLOMe due to the low expression of Cathepsin C (PMID: 32319717), and second, we have observed that LLOMe-induced stress causes differential expression from CMV or EF1a promoters, which we use for the expression of our constructs in HEK293T cells. We have now added statistical analysis to 3D.

To investigate whether NLRP6 activation depends on the NACHT domain, we planned to express NLRP6, NLRP3, and NLRP6/NLRP3 chimeras in HIEC-6 cells, using extracellular potassium supplementation to differentiate between NLRP6- and NLRP3-like responses. The reasoning behind this experiment was that in macrophages, NLRP3 activation by endolysosomal damage depends on cathepsin release and potassium efflux (PMID: 18604214, PMID: PMC3730833), and therefore extracellular potassium should block NLRP3-like activation. In contrast, as demonstrated in our manuscript, NLRP6 activation is not blocked by extracellular potassium.

Unexpectedly, we observed that while nigericin-induced LDH release in NLRP3-expressing HIEC-6 cells was potassium-dependent, LLOMe-induced NLRP3 activation was potassium-independent (if anything, it was enhanced by potassium supplementation, see below). Consequently, it was not possible to distinguish a NLRP6-type response from a NLRP3-type response to LLOMe based on extracellular potassium supplementation. For this reason, we did not pursue further analysis of the NLRP6/NLRP3 chimeras in HIEC-6 cells. The underlying reason why LLOMe activates NLRP3 independently of potassium efflux in HIEC-6 cells remains unclear and could be linked to intrinsic differences between macrophages and epithelial cells or the source of lysosomal damage (particulate matter vs. LLOMe).

LDH release from NLRP3-WT^{tg} HIEC-6 cells induced with 1 μg/ml doxycycline overnight, left untreated or treated with 0.5mM LLOMe (4h), 10μM nigericin (1h) or an equivalent volume of ethanol (EtOH, 4h) as vehicle control. Potassium supplemented medium was added 1h before treatment and was maintained thereafter.

3) Have the authors considered that dsRNA, LTA (and/or Taurin) alone are insufficient and Actin destabilization is also required? And have they considered IFN priming may also be needed in the HIEC-6? I would recommend and would be interested to see at least as a reviewer figure, a co-treatment of these proposed agonists and actin destabilized, with/without IFN pre-treatment like in some of the other agonist papers. Same for a co-treatment of dsRNA or LTA with heat-killed lysate.

We excluded the involvement of actin destabilization/polymerization in NLRP6 activation for several reasons. First, *Listeria ΔplcB* still causes actin destabilization in the form of actin tails and presumably presents all the required PAMPs to the cells similarly to the wild-type bacteria. Nevertheless, these bacteria did not activate NLRP6 fully in infected cells. Additionally, LLOMe mediated activation of NLRP6 is independent of actin-destabilization as well.

Nevertheless, we addressed this question by expressing NLRP6 and ActA protein in mCherry-ASC^{sg} HEK293T cells, causing actin destabilization, and subsequently infecting them with either WT or Δ hly *L. monocytogenes*, or transfecting LTA or poly(I:C) into these cells. We observed that infection with *Listeria* caused NLRP6 activation, albeit to a lesser degree as usual, which might be due to impairment of actin tail formation caused by the exogenous expression of ActA. Transfection of LTA or poly(I:C) did not cause any NLRP6 activation, despite the presence of ActA, thus demonstrating that NLRP6 cannot be activated by combining actin destabilization and PAMPs.

As the results were negative for LTA and poly(I:C), the two best described ligands, we did not further test Taurine or heat-killed lysates.

Flow cytometry-based quantification of ASC speck formation in mCherry-ASC^{tg} HEK293T cells expressing hNLRP6 and ActA, infected with WT or $\Delta actA$ *L. monocytogenes* EGD for 6h or transfected with DOTAP or Lipofectamine (LF) only or transfected with LTA or poly(I:C) 6h at 10 μ g/300,000 cells.

As for the requirement of IFN γ priming: We transfected NLRP6 expressing HIEC-6 cells either primed with IFN γ overnight or left unprimed with LTA and poly(I:C) and found that IFN γ priming did not lead to a NLRP6 dependent response to LTA or poly(I:C) transfection. These findings are now displayed in Fig. EV3C.

4) In Figure 4 the showing of actin tails requires larger magnifications.

We presume the reviewer was referring to the former Figure S5, now presented as Figure EV4, which is the only figure showing small actin tails associated with bacteria (Figure 4 did not show actin tails). In response, we have provided magnified insets highlighting the bacterial actin tails in Figures EV4C and E.

5) At least a representative FACS result and a precise description of the gating strategy used for the ASC aggregation assay should be shown in Fig. S1 to understand the most critical assay used throughout better. For fig. 5, the FACS plots and gating also should be shown to illustrate for the disparate infection rates mentioned (but not specified).

We now show the general gating strategy for the ASC speck formation assay in Figure EV1B (uninfected cells). We also added the gating strategy for the mCherry tagged NLRs in EV2A and the strategy for infected cells in EV3G and EV4B. As for the disparate infection rates, we showed in former Fig. S5 the FACS plots displaying how the $\Delta actA$ mutant infects fewer cells, but to a higher degree (now EV4A).

6) Statistics: ANOVA assumes a normal distribution of data but how normality was tested for is not described in methods. This should be amended and the choice of test modified if necessary.

We have amended the Methods section accordingly. Since inflammasome assays, such as LDH release, are typically performed with a limited number of technical replicates, normality can only be verified by graphical means.

D) Minor concerns that should be addressed

+ The expression of pattern of NLRP6 in line 48 should be more finely delineated between mouse vs human.

We added a specification that NLRP6 protein in humans is expressed in the small intestine, while in mouse it is expressed in the small and large intestine (PMID: 33376129).

+ Bafilomycin A1 can also interfere with DNA and RNA sensing, albeit in the endosomes. Can the authors be sure that it does not affect sensing of bacterial RNA, which the authors concede to be a labile and potentially hard to track trigger?

While we cannot completely rule out the possibility that bafilomycin affects cytosolic RNA sensing, previous studies suggest its primary mechanism involves altering endosomal pH. For instance, bafilomycin has been shown to disrupt RNA sensing by raising the pH of endosomes. In the case of TLR3, which senses RNA within endosomes, blocking endosomal acidification impairs RNA binding, an interaction that depends on both RNA length and pH (PMID: 18172197). Additionally, bafilomycin treatment has been reported to reduce TLR3 transcript and protein levels, likely due to disrupted endosomal maturation (PMID: 38906273). Since NLRP6 functions in the cytosol at neutral pH, it is unlikely to be directly affected by bafilomycin.

Furthermore, our data showing that the ActA- and PC-PLC mutants show reduced NLRP6 activation in infected cells, even though they have the same PAMP profile as the WT bacteria, as well as the data showing that endolysosomal damage without any bacterial PAMP present activates NLRP6, both suggest that a labile PAMP is not the activator of NLRP6.

+ That HIEC-6 are a non-immortalized cell line should be specified in line 118/119 as the reader currently might wrongly assume these are either primary human intestinal endothelial cells or a colorectal cancer cell line

We added the specification that HIEC-6 are a non-immortalized intestinal epithelial cell line.

+ The blotting for NLRP3 in HIEC-6 in combination with THP-1 might mask low NLRP3 expression in these cells next to an expectedly very high NLRP3 signal from THP-1. An additional blot of HIEC-6 next to HEK293T cells (certainly negative) would be more informative and useful to rule out basal expression of NLRP3 in their system. mRNA quantification for NLRP6 and NLRP3 would also be useful.

We have performed further analysis in the HIEC6 cells treated with different priming signals and run Western blots with HEK293T and U937 in parallel and have confirmed that there is no NLRP3 expression in the HIEC-6 cells. This is also in line with a previous report that found no expression of NLRP3 in HIEC-6 (PMID 37558421). This data is in supplementary figure 1B.

+ The targeting strategy of KO generation for Caspase-1 and -4 should be described. What resulted in the KO, e.g. a frameshift leading to a truncated protein, should be mentioned.

For all KO lines generated in our study, we worked with a polyclonal population following antibiotic selection for transgene integration. Therefore, it is likely that a range of INDELS are present. We have clarified this in the materials and methods section, as well as added a column to the table of sgRNAs showing which exon was targeted. In most cases, it is an early exon where an INDEL is likely to lead to truncated, non-functional protein or nonsense mediated decay of the mRNA. Western blot analysis of all polyclonal KO populations is shown throughout the manuscript, showing successful reduction of protein levels in the cells (Figure 2F and H for Caspase-1 and GSDMD, Appendix S2E-F for ATG7)

+ For completeness sake the authors might want to test the effect of Taurin on NLRP6 in their system as Taurin is mentioned but not assessed.

We have tested Taurine in the HIEC-6 cell model and found that it did not induce NLRP6-dependent cell death (EV3D).

+ Controls that the NLRP6 the NLRP6-non-activators LTA and dsRNA are intact and generally active (e.g. for other pathways) are missing, e.g. for dsRNA to show IFN induction from the same cells.

To address this, we tested whether LTA and poly(I:C) could activate signaling via TLR2 and TLR3, respectively, by assessing NF- κ B activation through phosphorylation of p65. As expected, both ligands successfully induced TLR-mediated signaling. Furthermore, transfection of poly(I:C) into HIEC-6 cells led to the expression of GBP1 (an interferon-stimulated gene) after 8 hours, indicating interferon production, likely as a result of MDA5 activation. Thus, we can conclude that the ligands used were intact and able to stimulate pattern recognition receptors.

Representative Immunoblots of HIEC-6 cells, either treated with 1ug/mL poly(I:C) or 10ug/mL Pam3CSK4 for 1h or 100ug/mL LTA for 1 or 2h and analyzed for phosphorylation of p65 (left). On the right, HIEC-6 cells were transfected for 8 hours with Lipofectamine (LF) only, or with Lipofectamine and poly(I:C) using 10ug/300,000 cells and analyzed for expression of GBP1. Tubulin was used as a loading control.

Quantification of Immunoblots of HIEC-6 cells, either treated with 1ug/mL poly(I:C) or 10ug/mL Pam3CSK4 for 1h or 100ug/mL LTA for 1 or 2h and analyzed for phosphorylation of p65 (left, n=3). On the right, HIEC-6 cells were transfected for 8 hours with Lipofectamine (LF) only, or with Lipofectamine and poly(I:C) using 10ug/300,000 cells and analyzed for expression of GBP1 (n=2). Tubulin was used as a loading control.

+ The authors need to mention in line 344 that LLoMe is also a well-established NLRP3 agonist. An inclusion of LLOMe in the chimera analysis would hence have been helpful too. MCC950 might be a good control in some of the experiments of Fig. 5/6 to exclude NLRP3 completely.

We have updated the text to clarify this. We have performed the LLOMe stimulations in the presence of MCC950 in HIEC-6 cells and confirmed that LDH release is unaffected, further ruling out any involvement of NLRP3 in the induction of cell death. This data can be found in figure EV 5B. We also performed *Listeria* infections in the presence of MCC950 and confirmed this had no effect on LDH release (Figure S2C).

Unfortunately, for the reasons stated above, we were not able to explore LLOMe-induced NLRP6 activation in ASC^{tg} HEK293T or HIEC-6 cells.

+ In Fig. 4B white boxes with small digits 1 or 2, rather than arrowheads (which might be mistaken to point to "specks" of some sort) might be more useful to show the 2 inserts?

We clarified the data using white boxes with dashed lines to conform to the rest of the manuscript.

+ Why in Fig. 3F was an uncorrected Fisher's LSD used?

We corrected the analysis and used a two-way ANOVA with Šídák's multiple comparisons for both Fig. 3E and F

+ The discussion between line 414 and 434 (approx) is quite dense and could maybe be simplified.

We have simplified the respective part of the discussion.

Any additional non-essential suggestions for improving the study (which will be at the author's/editor's discretion)

+ The microscopy analysis in fig. 2A shows that ASC specks seem to occur in direct proximity to the nucleus. What is the significance?

Previous studies have reported that ASC specks are perinuclear structures, particularly following NLRP3 inflammasome activation, leading to speculations that speck formation occurs at the microtubule-organizing center (MTOC). While we have not investigated whether NLRP6-induced ASC speck formation also involves the MTOC, it would be interesting to explore whether NLRP6-ASC specks co-localize with specific subcellular structures or organelles, as this could provide valuable insights into the mechanism of NLRP6 activation.

Referee #2:

While several previous publications have claimed to report mechanisms activating the NLRP6 inflammasome, these have been controversial in the field. Here, the Broz lab provide compelling evidence in support of a model in which NLRP6 is a sensor for sterile endolysosomal damage, or bacterial-induced membrane damage. In an elegant series of cell infection assays (*Listeria*, *Shigella*) of NLRP6-expressing epithelia, the authors untangle functions for NLRP6 in sensing membrane damage during infection and with lysotrophic agents. In general, the study is compelling and worthy of publication in EMBO J, and I congratulate the authors on the study. There are however, several aspects that require addressing to meet the required level of scientific rigour for publication.

We would like to thank the referee for the supportive feedback and the suggestions for improvement of our study. Below we have summarized how we addressed the points that were raised.

Major issues:

1. Throughout the figures (Figs 1-5, S1-6), there is a perplexing lack of uninfected control samples shown. Because ectopic NLRP6 expression itself triggers inflammasome signalling, it is often difficult to determine whether it is infection or NLRP6 that is driving the response. This makes it very difficult to agree with some of the author's statements (e.g. it is often stated that infection triggers an NLRP6-dependent response, but without an infected control, a reader can only conclude it is an NLRP6-dependent response). Upon close inspection of the methods, it seems the uninfected control was indeed performed and used to somehow "normalise" or background subtract the data. In this Reviewer's opinion, the data should not be transformed in this way, and the graphs should show the uninfected controls, so that the reader properly assess the full dataset.

We would like to clarify that all our assays included uninfected or untreated control samples, which is particularly important given that NLRP6 is prone to autoactivation. We tried to indicate that results were normalized to these background controls, and we apologize if this was not communicated clearly enough.

To address the referee's criticism, we now show non-normalized data, such as for imaging analysis and flow cytometry analysis, and include the untreated/uninfected control samples. As can be appreciated in all figures, this does not affect our conclusions. For LDH assays, we continue to present data normalized to the uninfected or untreated control, this is a well-accepted way to analyze such assays and is in line with the recommended procedure from the manufacturer.

2. In Figure 3 and S3, flow cytometry is performed to analyse Cherry-tagged NLRP3 versus NLRP6 function in ASC-GFP cells. The authors show Cherry expression is somewhat similar between constructs. Many studies, including this study, show the strong impact of NLR expression level on inflammasome signalling outputs, so small differences in expression can generate spurious results. The advantage of having cherry-tagged constructs is that you can precisely control for cherry expression by gating on cherry when analysing the cytometry data. It is preferable to analyse these data using a narrow cherry gate (rather than all cherry+ cells), as this will allow you to compare ASC specks in cell populations that express equivalent levels of NLR-cherry proteins.

This is an excellent suggestion, and in response we transformed all the data using mCherry tagged NLR constructs to only analyze the cells that fall within a narrowly defined mCherry gate. The gating strategy is displayed in Fig. EV2A. As can be appreciated in the figures, this gating strategy did not alter the quality or validity of our data and conclusions. It did however increase the overall ASC speck formation in activated conditions, such as in *L. monocytogenes* infected cells.

3. Some graphs and statistics appear to represent pseudo-replicated data. In several figures (Figs:2D, 2G, 4D, 5C-D, 6C-F; Supp 6A, 6F, 6I, 6J) technical replicates are shown alongside biological replicates, and statistics appear to have utilised both sets of data. Only biological replicates should be shown in graphs, and analysed by stats.

We have altered all relevant graphs, each data point now represents the mean of the 3 technical replicates in each biological replicate and statistical analysis was performed using the 3 resulting values that are shown on the graph.

4. The authors propose (e.g. Fig. 6G, discussion) that NLRP6 is activated in epithelial cells to assemble an inflammasome that cleaves GSDMD and thereby induces pyroptotic death. This is a reasonable presumption, but the GSDMD dependency of cell death was not formally tested. Suggest showing this with GSDMD knockout HIEC.

We generated GSDMD knockout HIEC-6 (Fig. 2H) and confirmed that LDH release was reduced during both *Listeria* infection (Fig. 2I) and LLOMe treatment (Fig. 6E).

5. A major substrate of inflammasomes are immature cytokines, such as pro-IL-1 β , pro-IL-18 and pro-IL-37. Do NLRP6-activated HIEC cleave and release these cytokines, or is cell death the only outcome of NLRP6 inflammasome signalling?

We analyzed IL-1 β release from NLRP6-transgenic HIEC-6 cells, in which NLRP6 expression was either induced with doxycycline or left uninduced, followed by treatment with either *L. monocytogenes* or LLOMe. Consistent with the GSDMD cleavage pattern, we observed a baseline level of cytokine release upon doxycycline-induced NLRP6 expression, which was further enhanced following infection with wild-type *Listeria* (Fig. 2J) or treatment with LLOMe (Fig. 6F).

6. Is the elusive signal detected by NLRP6 actually galectin redistribution upon membrane damage?

This is an intriguing hypothesis and one we are interested in pursuing in future studies. Galectin redistribution is however just one of many cellular events triggered by endolysosomal damage, and thus a comprehensive, systematic analysis of all these events will be needed to identify the elusive activating signal of NLRP6.

Study limitations:

1. While unavoidable until cell lines endogenously expressing NLRP6 are identified, it is a weakness of the study that NLRP6 was ectopically, rather than endogenous expressed throughout the study.

As all reviewers made this point, we conducted a further search for cell lines endogenously expressing NLRP6 but were unsuccessful in identifying such a cell line so far (see response to Reviewer 1 for details).

2. Related to the above point, it is a pity that ectopic NLRP6 expression in HIEC was strong enough to bypass NLRP6 autoinhibition in the absence of a stimulus (i.e. NLRP6 expression without stimulus triggered some cells to signal via this inflammasome). Because of this, it is important to show the untransformed data throughout the manuscript (see major point 1).

We have made attempts to further sort our NLRP6 expressing HIEC-6 cells for GFP-low populations (GFP is expressed via IRES from the NLRP6 construct the HIEC-6 were transduced with) in the hope this would represent a population with lower NLRP6 expression. Unfortunately, the difference in protein levels was minimal and there was no difference in signal-independent activation of NLRP6 upon induction of expression.

As described above, we now present non-normalized data throughout the study. However, for the LDH assays, we prefer to display the data normalized to untreated and 100% lysis controls, consistent with conventions in the field. A consequence of this normalization is that cell death due to NLRP6 autoactivation is not visible in the LDH data. Nevertheless, this effect is clearly reflected in the IL-1 β levels and GSDMD cleavage patterns. We have also explicitly stated this in the text and therefore we believe that the potential impact of NLRP6 autoactivation is transparently disclosed to the reader.

In this Reviewer's opinion, these limitations should not preclude the publication of this interesting work, as so little is known about NLRP6 functions and the study represents a major step forward.

We sincerely appreciate the referee's support for publishing the study despite its limitations.

Minor suggestions for improving clarity:

1. Line 122: suggest mentioning in the text that NLRP6 is expressed as an NLRP6-IRES-GFP cassette, so that it is not a surprise for the reader to see GFP detected on western.

We have clarified this in the text.

2. Figure 1B should be analysed statistically

We performed a two-way ANOVA with Šídák's multiple comparisons on the data and display the found significance in Fig. 1B in relation to the uninfected control.

3. Fig 1E lacks a bafA-untreated uninfected control

We transformed this data to display the non-normalized values, which then allows us to display the untreated, uninfected control.

4. Line 150: "the results confirm that LLO-dependent escape of Listeria from the endosomal compartment is an essential prerequisite for NLRP6 activation in human epithelial cells". While this is a reasonable interpretation for LLO action, the authors have not directly shown that endosomal escape is necessary. Please modify to ensure it is clear this is speculation. Also true of "cytosolic entry" in line 163.

We have made this alteration.

5. A large portion (~25%) of the discussion is dedicated to discussing possible involvement of autophagy, while the data did not support such an involvement. This makes the discussion feel unfocussed. Suggest shortening this section substantially.

We have shortened this section.

Referee #3:

Boegli et al. present compelling evidence that NLRP6 is activated through endolysosomal damage during *Listeria monocytogenes* infection in human intestinal epithelial cells (IECs). Their findings challenge earlier reports suggesting that NLRP6 directly senses bacterial pathogen-associated molecular patterns (PAMPs), such as lipoteichoic acid and dsRNA, via its leucine-rich repeat (LRR) domain to initiate inflammasome activation. Instead, the study demonstrates that NLRP6 activation is dependent on *Listeria*-induced endolysosomal disruption, specifically requiring Listeriolysin O-mediated cytosolic invasion and escape from secondary vacuoles. This mechanism highlights NLRP6's role as a sensor of cellular homeostasis rather than direct pathogen recognition, which contrasts with previous studies reporting PAMP-mediated activation.

The study's robust datasets are well-controlled, and the authors carefully contextualize their findings in light of prior research. The manuscript is clear and requires minimal editing.

We thank the referee for the positive assessment our study.

However, it should be noted that all experiments were conducted in cell lines reconstituting NLRP6 expression. Since NLRP6 is endogenously expressed primarily in intestinal epithelial cells, further validation using NLRP6 knockout (KO) mice and cultured primary intestinal cells would strengthen the conclusions. For example:

- Testing in primary cells: Isolating intestinal epithelial cells from KO and wild-type mice could help assess responses to identified stimuli like LLOMe and *Listeria* strains compared to lipoteichoic acid and dsRNA.

We tested intestinal explants to test our hypothesis in a primary cell system as also suggested by Reviewer #1, but we found these experiments technically challenging, and the results were inconsistent. Additionally, we would like to generate organoids and use primary epithelial enterocytes to further validate our data. However, due to the limited time for submission of a revised manuscript and the technical difficulties that come with establishing such models in our lab (we have no previous experience with these model systems) we were so far not able to systematically test primary cell models and consider these outside of the scope of this study.

- Alternative approaches, if mice are not available: Using viral delivery of shRNA targeting NLRP6 in primary intestinal epithelial cells could provide insights into inflammasome activation mechanisms in a more physiologically relevant setting.

We have recently generated a new NLRP6-KO mouse line on the C57BL/6 background, but the limitations mentioned above did not allow us to explore primary systems to the required degree.

It would also be interesting to investigate whether LLOMe-induced NLRP6 activation also depends on its NACHT domain.

Unfortunately, we cannot explore LLOMe-induced NLRP6 activation in ASC^{tg} HEK293T cells, for two reasons. First, HEK293T are poorly responsive to LLOMe due to the low expression of Cathepsin C (PMID: 32319717), and second we have observed that LLOMe-induced stress causes differential expression from CMV or EF1a promoters, which we use for the expression of our constructs in HEK293T cells.

To investigate whether NLRP6 activation depends on the NACHT domain, we planned to express NLRP6, NLRP3, and NLRP6/NLRP3 chimeras in HIEC-6 cells, using extracellular potassium supplementation to differentiate between NLRP6-like and NLRP3-like responses. The reasoning behind this experiment was that in macrophages, NLRP3 activation by endolysosomal damage depends on cathepsin release and potassium efflux (PMID: 18604214, PMID: PMC3730833), and thus therefore extracellular potassium can block NLRP3 activation. In contrast, as demonstrated in our manuscript, NLRP6 activation is not blocked by extracellular potassium.

Unexpectedly, we observed that while nigericin-induced LDH release in NLRP3-expressing HIEC-6 cells was potassium-dependent, as expected, LLOMe-induced NLRP3 activation was potassium-independent (if anything, it was enhanced by potassium supplementation, see below). Consequently, it was not possible to distinguish a NLRP6-type response from a NLRP3-type response to LLOMe based on extracellular potassium supplementation. For this reason, we did not pursue further analysis of the NLRP6/NLRP3 chimeras in HIEC-6 cells. The underlying reason why LLOMe activates NLRP3 independently of potassium efflux in HIEC-6 cells remains unclear and could be linked to intrinsic differences between macrophages and epithelial cells or the source of lysosomal damage (particulate matter vs. LLOMe).

LDH release from NLRP3-WT⁹ HIEC-6 cells induced with 1 μg/ml doxycycline overnight, left untreated or treated with 0.5mM LLOMe (4h), 10μM nigericin (1h) or an equivalent volume of ethanol (EtOH, 4h) as vehicle control. Potassium supplemented medium was added 1h before treatment and was maintained thereafter.

Overall, this study significantly advances our understanding of NLRP6 inflammasome biology and challenges earlier paradigms of direct PAMP sensing. While experiments in primary cells would be ideal, the data presented are sufficiently compelling to warrant publication without these additional validations.

We sincerely appreciate the referee's support for publishing the study despite its limitations.

Dear Petr,

Thank you again for the submission of your revised manuscript (EMBOJ-2025-120875R) to The EMBO Journal for our consideration, and for your patience during peer review. As I have already informed you, your manuscript has been sent back to the three original referees who had previously assessed the first version of the work, and we have now received the complete set of their comments, which are included below.

I am very pleased to say that all three referees are very supportive of the revised manuscript, mentioning that their initially raised criticisms and concerns have been adequately and sufficiently addressed in a strengthened and compelling manuscript that now meets the standards of The EMBO Journal. In light of this input, I am glad to inform you that your manuscript has been accepted in principle for publication in our journal - congratulations on an excellent work!

There are only a few minor remaining suggestions for improvement from referees #1 and #3 that are very reasonable and should be easy to address in a final version of the manuscript. When you are ready, please submit this final version along with a detailed point-by-point response detailing any changes to the manuscript. Please let me know if you face any difficulties in addressing any of these minor points.

There are also a few changes from the editorial side that we kindly request you to address in this final version of your manuscript, before we can move forward with its formal acceptance and publication:

- Please include the funding information in the "Acknowledgements" section of the revised manuscript.
- The maximum number of keywords that can be listed after the Abstract is 5 (7 keywords are currently listed).
- Please rename heading "Resource availability" to "Data availability".
- Please rename heading "Declaration of interests" to "Disclosure and competing interests statement".
- Please rename heading "Methods and Protocols" to "Methods".
- The author contributions statement should be removed from the manuscript file. Instead, we use CRediT to specify the contributions of each author in the journal submission system. Please feel free to use the free text box to provide more detailed descriptions during submission. See also our guide to authors for more information: <https://www.embopress.org/page/journal/14602075/authorguide#authorshipguidelines>.
- The nomenclature of the Expanded View Figures, throughout the manuscript and in the file names, should be "Figure EV1-EV5" instead of "Extended View Figure 1-5".
- The Appendix file needs to be uploaded in PDF format; the heading on its first (title) page should be "Appendix for:", followed by the manuscript's title and a brief Table of Contents including page numbers for all listed items; the nomenclature of the Appendix Figures must be "Appendix Figure S1-S2" instead of "Appendix Supplemental Figure 1-2" (throughout the Appendix and in all callouts in the main manuscript file).
- Please make sure to prepare and upload the final Appendix PDF file at a higher resolution, as the Appendix Figures are currently rather pixelated.
- Thank you for providing the Source Data for your study; please also fill in and upload a Source Data checklist according to the instructions you have previously received from our team; please contact contact@embojournal.org if you have any questions or need assistance.
- Please note that EMBO press papers are accompanied online by:
 - A) a short (2 sentences) summary of the findings and their significance,
 - B) 2-5 short bullet points highlighting the key results, and
 - C) a synopsis image in .jpg or .png format that is exactly 550 pixels wide and 300-600 pixels high (the height is variable). Please note that all text needs to be legible at the final size.Please upload this information along with your revised manuscript (the text for A and B should be provided in a separate Word file).
- Please provide the exact p-values in the legends of Figures 1B, 2D, 3B, C, D; 3A, F; 4F, 5D, 6C, EV2 E, EV3 H, EV5 D.
- Please consider including the contents of Tables S1 (primers) and S2 (single guide RNAs) in the main Reagents and Tools Table. Alternatively, you could keep the tables if you prefer, but in that case they should be named "Table 1" and "Table 2" and

placed between the main the EV Figure legends; or they could be renamed to "Appendix Table S1" and "Appendix Table S2" and moved to the Appendix file.

- The order of manuscript sections must be corrected as follows: Title page - Abstract and Keywords - Introduction - Results - Discussion - Methods - Data Availability - Acknowledgements - Disclosure and Competing Interests Statement - References - Figure Legends - main Tables (Table 1 and Table 2 in this case, if you prefer to keep them as "main" Tables) - Expanded View Figure Legends.

Please also note that as part of the EMBO publications' Transparent Editorial Process, The EMBO Journal publishes online a Peer Review File along with each accepted manuscript. This File will be published in conjunction with your paper and will include the referee reports, your point-by-point response and all pertinent correspondence relating to the manuscript. You can opt out of this by letting the editorial office know (contact@embojournal.org). If you do opt out, the Peer Review File link will point to the following statement: "No Peer Review File is available with this article, as the authors have chosen not to make the review process public in this case."

We look forward to seeing a final version of your manuscript as soon as possible. Please let us know if you have any questions and use this link to submit your revision: <https://emboj.msubmit.net/cgi-bin/main.plex>.

Best regards,

Ioannis

Referee #1:

In their revised work, Boegli et al made substantial efforts to address the comments of all reviewers and provide a comprehensive and satisfactory discussion of most of the issues flagged up. For the most critical point - exogenous systems/overexpression - their efforts unfortunately did not lead to the discovery of an endogenously NLRP6 competent system. I have a some remaining minor comments on how to deal with this finding, but the infection-re-infection experiment in Fig. 5I is very convincing and nicely supports the proposed mechanism. Collectively I consider the efforts adequate and that the remaining weakness should not preclude publication, as the authors are facing difficulties that seem inherent in the field, because the work nevertheless provides a substantial step forward in NLRP6 biology and is generally a meticulously performed, comprehensive and conclusive piece of work that authors can be congratulated upon and that reaches the high standard of EMBO Journal.

May remaining minor concerns which should be easily addressable in a minor revision are as follows:

- 1) To save readers from re-checking the same cell lines all PCR and IB data in the reviewer letter should be integrated into the manuscript as supplemental information. So data for Caco-2, HT-29, HCT116, SW1463, YAMC and mcCcl2 should be shown as this is vital for the NLRP6 field when interpreting other studies that make claims about NLRP6 in such cellular systems. Moreover, it just does not make sense that others waste and effort on such an exploration of what has already been done by the authors. Inclusion of the data would serve as a valuable resource of information to the community. Others might find it helpful that their negative data on NLRP6 expression have been confirmed. As the authors already mention themselves (and I agree), this fact it does not compromise or drastically alter their work - so inclusion of the data is no concern. As the number of repeats was not described in the reviewer figure, I cannot judge how many repeats of these data were run. Ideally, the authors would present at least 2 repeats of these negative data, but also single tests, labelled clearly as "preliminary data" would be acceptable as comprehensive repeats are unlikely to provide different results (provided the below controls are considered). Inclusion of this negative data would also make the apparent lack of data from such an endogenous system more understandable to the reader and hence increase the credibility of the work.
- 2) I would request 2 more Immunoblots. a) the inclusion of explant extracts in the IB of murine cell lines. It is ok that the authors do not include the data from infections attempted in this explant system but at least an additional blot of running WT and NLRP6 KO explant lysates side by side to their mouse cell line lysates would show their antibody is working as these samples should be positive (they confirmed mRNA by qPCR). The mNLRP6 HEK lysate is not a good control as many antibodies that work well on

highly overexpressed proteins fail for endogenous ones. Thus their negative result requires these explant control lysates - as they have the mice this appears straightforward. b) I also ask the same for a blot of all human cell line lysates with a human colon total protein lysate available commercially e.g. HT-311 from Zyagen (there is an EU distributor and there may also be other vendors) to test the human antibody too. Collectively, this would reassure both reviewers and readers that their negative data are not due to poor reagents - of which unfortunately many circulate in the NLR field. Moreover, since their study strongly and convincingly challenges previous work on RNA and LTA as NLRP6 ligands it should excel in rigor also on this point. And as the lack of an endogenous system has been a major criticism of all three reviewers this single experimental request hopefully appears reasonable and the data should be added to the supplement/appendix in combination with the full reviewer data. 3) I would welcome a slight expansion of their "limitations" statement in line 564-564 about the extrapolation of their work into a physiological context, potentially also referring to the abovementioned supplemental data. This would help especially newcomers to the field to directly appreciate the limitations all NLRP6 researchers are faced with.

Referee #2:

The authors have substantially improved the manuscript during its revision, and have comprehensively addressed my former concerns. I recommend manuscript publication without further revision, and congratulate the authors on this excellent and compelling study.

Referee #3:

Summary:

In my initial review, I raised concerns regarding the exclusive use of engineered systems for studying NLRP6 activation, and the resulting uncertainty around the physiological relevance of the findings. The revised manuscript addresses this issue in a transparent and thoughtful way, but I believe a small clarification in the discussion is still warranted.

Comment

The authors convincingly demonstrate that NLRP6 can sense endolysosomal damage and mediate inflammasome activation when reconstituted in human epithelial cells. However, all findings rely on inducible or overexpression systems in cell lines that do not endogenously express NLRP6. While the authors acknowledge this limitation and note the challenges of studying NLRP6 in vitro, the manuscript would benefit from a more explicit statement that these results have yet to be validated in physiologically relevant models such as organoids, primary epithelial cells, or in vivo systems.

Even a short note outlining this gap as a direction for future work would strengthen the framing of the conclusions.

Verdict: Minor Revision

The revised manuscript addresses my concern with appropriate transparency. I support publication after a brief clarification in the discussion.

Referee #1:

In their revised work, Boegli et al made substantial efforts to address the comments of all reviewers and provide a comprehensive and satisfactory discussion of most of the issues flagged up. For the most critical point - exogenous systems/overexpression - their efforts unfortunately did not lead to the discovery of an endogenously NLRP6 competent system. I have a some remaining minor comments on how to deal with this finding, but the infection-re-infection experiment in Fig. 5I is very convincing and nicely supports the proposed mechanism. Collectively I consider the efforts adequate and that the remaining weakness should not preclude publication, as the authors are facing difficulties that seem inherent in the field, because the work nevertheless provides a substantial step forward in NLRP6 biology and is generally a meticulously performed, comprehensive and conclusive piece of work that authors can be congratulated upon and that reaches the high standard of EMBO Journal.

We thank the reviewer for their positive feedback and for supporting the publication of our work despite the remaining limitations.

May remaining minor concerns which should be easily addressable in a minor revision are as follows:

1) To save readers from re-checking the same cell lines all PCR and IB data in the reviewer letter should be integrated into the manuscript as supplemental information. So data for Caco-2, HT-29, HCT116, SW1463, YAMC and mCcl2 should be shown as this is vital for the NLRP6 field when interpreting other studies that make claims about NLRP6 in such cellular systems. Moreover, it just does not make sense that others waste and effort on such an exploration of what has already been done by the authors. Inclusion of the data would serve as a valuable resource of information to the community. Others might find it helpful that their negative data on NLRP6 expression have been confirmed. As the authors already mention themselves (and I agree), this fact it does not compromise or drastically alter their work - so inclusion of the data is no concern. As the number of repeats was not described in the reviewer figure, I cannot judge how many repeats of these data were run. Ideally, the authors would present at least 2 repeats of these negative data, but also single tests, labelled clearly as "preliminary data" would be acceptable as comprehensive repeats are unlikely to provide different results (provided the below controls are considered). Inclusion of this negative data would also make the apparent lack of data from such an endogenous system more understandable to the reader and hence increase the credibility of the work.

We agree that sharing the data with the field will be helpful when interpreting studies making claims about NLRP6 in such system. We have thus added the cell line data into the appendix figure S1.

2) I would request 2 more Immunoblots. a) the inclusion of explant extracts in the IB of murine cell lines. It is ok that the authors do not include the data from infections attempted in this explant system but at least an additional blot of running WT and NLRP6 KO explant lysates side by side to their mouse cell line lysates would show their antibody is working as these samples should be positive (they confirmed mRNA by qPCR). The mNLRP6 HEK lysate is not a good control as many antibodies that work well on highly overexpressed proteins fail for endogenous ones. Thus their negative result requires these explant control lysates - as they have the mice this appears straightforward.

The polyclonal anti-mouse NLRP6 antibody generated as part of this study has been validated using organ lysates from WT and *Nlrp6*-deficient mice and made available to the community through Adipogen (anti-NLRP6 (mouse), pAb (IN121))
<https://adipogen.com/ag-25b-0045-anti-nlrp6-mouse-pab-in121.html>

During the validation of our antibody, we tested organ lysates and confirmed its specificity for murine NLRP6 using knockout (KO) mice. Western blotting for both organ lysates and cell lines was performed under identical conditions, including the same developer and exposure times. However, no detectable signal was observed in the cell lines.

For the reviewer's convenience we added the validation blot with lysates below.

b) I also ask the same for a blot of all human cell line lysates with a human colon total protein lysate available commercially e.g. HT-311 from Zyagen (there is an EU distributor and there may also be other vendors) to test the human antibody too. Collectively, this would reassure both reviewers and readers that their negative data are not due to poor reagents - of which unfortunately many circulate in the NLR field. Moreover, since their study strongly and convincingly challenges previous work on RNA and LTA as NLRP6 ligands it should excel in rigor also on this point. And as the lack of an endogenous system has been a major criticism of all three reviewers this single experimental request hopefully appears reasonable and the data should be added to the supplement/appendix in combination with the full reviewer data.

We agree that it is extremely important to use an properly validated antibody, considering the number of poor antibodies on the market.

This is why we chose to use 'anti-NLRP6/NALP6 (human), mAb (Clint-1)' from Adipogen. This antibody was extensively validated in a study from the Muruve lab PMID: 33376129. The authors of the study not only defined the epitope recognized, but also validated it with different human organs and tissue. They even indicated that freezing the samples would lead to a loss of signal (something we considered in our experiments).

Considering the difficult access to the indicated human samples in Switzerland (which is not part of the EU), and considering that the antibody has already been validated, we believe that further validation would not yield additional insights.

3) I would welcome a slight expansion of their "limitations" statement in line 564-564 about the extrapolation of their work into a physiological context, potentially also referring to the abovementioned supplemental data. This would help especially newcomers to the field to directly appreciate the limitations all NLRP6 researchers are faced with.

We appreciate this comment and expanded the limitations statement.

Ref. 2:

“The authors have substantially improved the manuscript during its revision, and have comprehensively addressed my former concerns. I recommend manuscript publication without further revision, and congratulate the authors on this excellent and compelling study.”

We thank the reviewer for their positive feedback and recommendation for publication.

Referee #3:
Summary:

In my initial review, I raised concerns regarding the exclusive use of engineered systems for studying NLRP6 activation, and the resulting uncertainty around the physiological relevance of the findings. The revised manuscript addresses this issue in a transparent and thoughtful way, but I believe a small clarification in the discussion is still warranted.

Comment

The authors convincingly demonstrate that NLRP6 can sense endolysosomal damage and mediate inflammasome activation when reconstituted in human epithelial cells. However, all findings rely on inducible or overexpression systems in cell lines that do not endogenously express NLRP6. While the authors acknowledge this limitation and note the challenges of studying NLRP6 in vitro, the manuscript would benefit from a more explicit statement that these results have yet to be validated in physiologically relevant models such as organoids, primary epithelial cells, or in vivo systems.

Even a short note outlining this gap as a direction for future work would strengthen the framing of the conclusions.

Verdict: Minor Revision

The revised manuscript addresses my concern with appropriate transparency. I support publication after a brief clarification in the discussion.

We thank the reviewer for the positive feedback and recommendation to publish the work despite the remaining weakness. We have adapted the discussion as requested to include a statement on the need to validate the model in a physiological system such as organoids, primary cells or in vivo.

Dear Petr,

Congratulations on an excellent manuscript! I am very pleased to inform you that it has been accepted for publication in The EMBO Journal. Thank you for comprehensively addressing the initially raised referee criticisms and the editorial requests for corrections and changes.

If you have any questions, please do not hesitate to contact the Editorial Office. Thank you for your contribution to The EMBO Journal. Working with you has been a pleasure!

Best regards,

Ioannis
